## Registered report

psychology

metascience, research methods, credibility, transparency, replication, real-time procedures

**Author for correspondence:**
Zoltan Kekecs
e-mail: kekecs.zoltan@gmail.com

# Raising the value of research studies in psychological science by increasing the credibility of research reports: the transparent Psi project

Zoltan Kekecs[1,2], Bence Palfi[3], Barnabas Szaszi[1], Peter Szecsi[1], Mark Zrubka[4], Marton Kovacs[1], Bence E. Bakos[1], Denis Cousineau[5], Patrizio Tressoldi[6], Kathleen Schmidt[8,9], Massimo Grassi[7], Thomas Rhys Evans[10], Yuki Yamada[11], Jeremy K. Miller[13], Huanxu Liu[12], Fumiya Yonemitsu[14], Dmitrii Dubrov[15], Jan Philipp Röer[16], Marvin Becker[16], Roxane Schnepper[16], Atsunori Ariga[14], Patrícia Arriaga[17], Raquel Oliveira[17], Nele Põldver[18], Kairi Kreegipuu[18], Braeden Hall[9], Sera Wiechert[19,20], Bruno Verschuere[19], Kyra Girán[1] and Balazs Aczel[1]

[1]Institute of Psychology, ELTE, Eotvos Lorand University, Izabella u 46. 1064, Budapest, Hungary
[2]Department of Psychology, Lund University, Box 213, Lund 221 00, Sweden
[3]Department of Surgery and Cancer, Imperial College London, London, UK
[4]Department of Psychology, University of Amsterdam, P.O. Box 19268, 1000 GG Amsterdam, The Netherlands
[5]École de psychologie, University of Ottawa, 136, Jean-Jacques Lussier, Ontario, Canada, K1N 6N5
[6]Studium Patavinum, and [7]Dipartimento di Psicologia Generale, Università di Padova via Venezia 8, 35131 Padova, Italy
[8]Department of Psychology, Ashland University, Ashland, OH 44805, USA
[9]School of Psychological & Behavioral Sciences, Southern Illinois University, Carbondale, USA
[10]School of Human Sciences, University of Greenwich, UK
[11]Faculty of Arts and Science, and [12]Graduate School of Human-Environment Studies, Kyushu University, 744 Motooka, Nishi-ku, Fukuoka 819-0395, Japan
[13]Department of Psychology, Willamette University, 900 State Street, Salem, OR 97301, USA
[14]Faculty of Letters, Chuo University, Hachioji, Tokyo 192-0393, Japan
[15]National Research University Higher School of Economics, Russian Federation

[16]Department of Psychology and Psychotherapy, Witten/Herdecke University, Witten, Germany
[17]Iscte-University Institute of Lisbon, CIS_Iscte, Av. das Forças Armadas, 1649-026, Lisbon, Portugal
[18]University of Tartu Institute of Psychology, Estonia
[19]University of Amsterdam, Amsterdam, The Netherlands
[20]Hebrew University of Jerusalem, Jerusalem, Israel

ZK, 0000-0001-9247-9781; PT, 0000-0002-6404-0058; KS, 0000-0002-9946-5953; HL, 0000-0001-7078-5956;
FY, 0000-0001-8774-4499; DD, 0000-0001-8146-4197; PA, 0000-0001-5766-0489; NP, 0000-0001-7307-544X;
BA, 0000-0001-9364-4988

The low reproducibility rate in social sciences has produced hesitation among researchers in accepting published findings at their face value. Despite the advent of initiatives to increase transparency in research reporting, the field is still lacking tools to verify the credibility of research reports. In the present paper, we describe methodologies that let researchers craft highly credible research and allow their peers to verify this credibility. We demonstrate the application of these methods in a multi-laboratory replication of Bem's Experiment 1 (Bem 2011 *J. Pers. Soc. Psychol.* **100**, 407–425. (doi:10.1037/a0021524)) on extrasensory perception (ESP), which was co-designed by a consensus panel including both proponents and opponents of Bem's original hypothesis. In the study we applied direct data deposition in combination with born-open data and real-time research reports to extend transparency to protocol delivery and data collection. We also used piloting, checklists, laboratory logs and video-documented trial sessions to ascertain as-intended protocol delivery, and external research auditors to monitor research integrity. We found 49.89% successful guesses, while Bem reported 53.07% success rate, with the chance level being 50%. Thus, Bem's findings were not replicated in our study. In the paper, we discuss the implementation, feasibility and perceived usefulness of the credibility-enhancing methodologies used throughout the project.

# 1. Introduction

Trust is currently in short supply in the psychology research community. Low reproducibility of important findings, growing evidence for a systematic publication bias and high prevalence of questionable research practices have increased scepticism about the trustworthiness of research reports [1,2]. Without reliable methods to distinguish between trustworthy and untrustworthy studies, the generalized mistrust in the field can lead to a devaluation of unbiased research. In the present paper, we provide a new collection of methodological tools for designing credible studies and for making this credibility transparent and verifiable for other researchers. We will illustrate the use of these innovative tools through a case study, a fully transparent multi-laboratory replication of Bem's Experiment 1 [3], which was co-designed by a consensus panel including both proponents and opponents of the hypothesis driving the original study.

A generalized scepticism about research reports was demonstrated in a recent study evaluating the replicability of 21 social science experiments [4]. Camerer *et al.* asked researchers to predict the likelihood that each of the 21 effects would *not* replicate in a series of two highly powered replication attempts. On average, 43% of researchers from the field of psychology estimated the chance that a given finding would not replicate as 50% or higher. Importantly, this scepticism was not restricted to unreproducible findings. For findings that did replicate, 26% of researchers on average (range: 7%–59%) predicted a 50% or higher chance that they would not replicate. Respondents were informed that both replication attempts would have 90% power to detect an effect smaller than the one reported in the original study, so those who truly believed in the effect should have reported a 10% or smaller chance for non-replication. Such 'true believers' were quite rare, only reaching 30% on average (6%–54%) across the 13 successfully replicated effects.[1] These results highlight that even well-conducted studies with reproducible findings suffer from the general mistrust of research reports among psychology researchers.

After realizing that too much trust was placed in research reports in the past, the field began seeking methods to ascertain the reliability of reported findings. Replication is the gold standard verification tool in science [5,6]. However, time and resources are limited, preventing researchers from replicating every research finding we rely upon. These limitations can be especially problematic on the cutting edge of research where waiting for independent replications might run the risks of losing competitive advantage or of time-sensitive topics losing their relevance. Furthermore, replication attempts are not

---

[1]Data extracted from the open data of Camerer *et al.* [4] accessible at: https://osf.io/6cu54/.

immune to suspicions of bias and flawed protocol execution, so supporters and sceptics of a particular finding can find themselves in a deadlock, trusting one study over the other (e.g. [7]). Thus, replication is a reliable but costly method for verifying credibility, and spending resources on replications might not be seen as worthwhile if the results of the replication can be easily dismissed on the ground of suspected biases or protocol deviations that cannot be proven or disproven.

Another important development in the field has been the recognition that classic research reports do not hold sufficient information to judge their credibility. As a response, many started to advocate for greater transparency in research reporting [2,8–11]. Current transparency standards such as preregistration, registered reports and open data, code and materials cover the planning phase and the reporting phase of the research project. However, these standards do not cover the fidelity of protocol execution and the integrity of data collection and data management, so there is no guarantee that the project has been executed as planned or as reported in the final paper, and the trustworthiness of the data is not guaranteed either. Several types of biases can remain hidden in this blind spot, known as the transparency gap. For example, mistakes can cause deviations from predetermined study protocols, researcher biases can manifest when dealing with unforeseen events or during communicating with participants or managing data, and deliberate fraud can take place at any time during the execution of the protocol [12–14]. This discontinuation in transparency leaves room for doubt about the credibility of the study. Thus, further innovations are needed in order to enable researchers to show that their research was executed with the highest integrity and with appropriate safeguards against bias.

Credibility concerns caused by the gap in transparency between planning and final research product can be addressed by demonstrating high protocol fidelity—the degree to which the study was executed exactly as planned or reported. The present paper showcases a collection of tools enabling researchers to demonstrate fidelity. (i) Verifiable record of protocol executions can be provided by directly collecting data into a trusted third-party repository and by depositing laboratory logs. (ii) Stakeholders can gain real-time insight into the study flow by applying born-open data and automated real-time research reports. (iii) Confidence in protocol integrity can be further raised by demonstrating that steps were taken to minimize the risk of mistakes, unforeseen events and protocol deviations. This can be achieved by fine-tuning the study protocol based on extensive piloting and using systematic interventions to minimize bias and mistakes. (iv) Employing external auditors can also provide an added layer of assurance.

Some of these techniques have only recently been made possible by technical advances, such as direct data deposition, or born-open data, and real-time research reports, while others such as piloting, staff training or audit have been around for a while now in psychology or other disciplines. The novelty of the approach we advocate lies in the systematic use of all appropriate tools combined to patch up the transparency gap, and by this, providing a more complete assurance about the integrity of the research study than was possible before.

## 1.1. A case study—a multi-laboratory replication of Bem, 2011, experiment 1

In order to demonstrate the use of these techniques in a real-world scenario, we have implemented all of the above-described tools in a community-designed fully transparent multi-laboratory replication of Bem's Experiment 1 [3].

Recently, researchers have reported positive results in support of 'psi' phenomena in several publications [3,15–19], interpreting their results as evidence for human extrasensory perception (ESP). This interpretation has been criticized on several accounts by sceptical researchers, who proposed instead that the dominance of positive findings in parapsychology is due to questionable research practices, publication bias and lack of methodological rigour [10,20–23]. The scientific discourse surrounding parapsychology studies played an important role in initiating the reformist movement in psychological science [24,25]. Bem's 2011 paper reporting the findings of nine experiments targeting precognition had an especially high impact, sparking several commentaries and replication attempts.

In a recent meta-analysis [26] assessing the experimental paradigms used by Bem, Experiment 1 produced the highest effect size across the 14 studies, including 863 participants in total (standardized mean difference = 0.14 [0.08–0.21]). We chose to replicate this particular experiment as a case study to demonstrate the use of the credibility-enhancing methods for multiple reasons. First, credibility expectations are extremely high for a replication of this study. This is true for sceptics and proponents of ESP alike. Sceptics do not expect a positive effect because it does not fit the current mainstream scientific paradigm, while proponents do not expect a negative effect, since the finding is supported by multiple replications and meta-analytic findings; thus, both sides demand extremely reliable evidence to prove them wrong. Expectations of credibility are not only high for this particular study,

but also for parapsychology research studies in general, which has driven methodological innovation in this field. For example, an early prototype of the Registered Reports format has been implemented in the *European Journal of Parapsychology* as early as the 1970s [27]. The same extreme credibility expectations prompt valuable feedback about the adequacy and coverage of the credibility-enhancing methodologies throughout the life cycle of our project from stakeholders engaged in the community-based study design, registered report review, multi-laboratory study execution and editing of the final research report.

Secondly, the project allowed us to test a prediction of the main prevailing theory about the dominance of positive findings in parapsychology literature. As a main starting point of the reformist movement in psychology, sceptical researchers proposed that the dominance of positive findings in parapsychology is due to questionable research practices, publication bias and lack of methodological rigour. We will refer to this view as the 'pure bias theory' of parapsychological effects. This theory has very strong support in the scientific community, and it has profound implications for other fields of biomedical and social sciences. It implies that, with absolutely no true effects, researcher degrees of freedom and biases inherent in the publication system by themselves can generate a whole field-worth of positive findings, and levels of statistical significance that would be considered convincing in more mainstream fields. If the pure bias theory is right, then we need to reconsider how we assess evidential value and how we interpret accumulated scientific findings in all fields of research characterized by the same research and publication practices. However, in spite of its profound implications, this theory has not been formally tested before, mainly because of the lack of tools to ascertain the presence or absence of the suspected biases. The credibility-enhancing methods described in this paper in combination with previously established transparency best practices will be used to negate or reveal known sources of researcher and publication biases. Effective control of these biases allows us to test a prediction of the pure bias theory. Specifically, the pure bias theory would predict that after eliminating the possibility of researcher and publication biases, and lack of methodological rigour, no effect would remain in this paradigm. Of course, no single replication can address a research question about the literature as a whole. However, according to a recent meta-analysis [26], this paradigm produced one of the most robust effects in parapsychology, so testing the prediction of the pure bias theory in this paradigm was certainly a good start.

Finally, this study has strong historical relevance to the reformist movement in psychology. The debate surrounding what could have contributed to the positive results reported by Bem [3], and how these biases could be prevented, had a defining impact on the reforms introduced in psychological science in the past decade. Just as the original Bem study is regarded by sceptics as a prototypical example of the prevailing biases on the field, the current project provides a prototypical example, agreed upon by nearly 29 research experts in the field, of how methodological best practices can improve credibility and minimize the risk of bias.

In summary, this paper aims to:

(1) present methodological tools useful for demonstrating protocol fidelity and providing a case study to illustrate their implementation,
(2) share experiences about the reception and perceived adequacy of these tools by researchers on the field, gathered through the life cycle of the case study,
(3) assess the costs and benefits of implementing these methodological advances in a multi-laboratory project, and
(4) test the prediction derived from the pure-bias theory of parapsychological effects, that Bem's original findings would not replicate when controlling researcher and publication biases and mistakes.

# 2. Material and methods

The study was preregistered on the OSF. The preregistration can be found at the following link: https://osf.io/a6ew3.

## 2.1. Consensus design process

The study protocol was first developed by the lead authors, then the protocol was amended through a consensus design process. During this process, researchers involved in the discussion and replication of the original study were systematically identified and invited to participate in the study design. The panel of this consensus design process included 29 experts (15 proponents and 14 opponents of the ESP

interpretation of the original results). After two rounds of review, the panel reached consensus according to pre-specified criteria, agreeing that the protocol has a high methodological quality and is secure against questionable research practices. The details of the consensus design process are described in the 'Consensus Design Process' section of the Supplement.

## 2.2. Replication study protocol

### 2.2.1. Participants

Adult participants with no psychiatric illnesses who have not participated in the study before and who are not under the influence of drugs or alcohol at the time of the session were eligible to participate. These criteria were checked before the start of data collection by self-report. No participant could be excluded based on these criteria after data collection has started.

### 2.2.2. Procedures

The experimental paradigm closely matched the protocol of Bem's [3] Experiment 1. After receiving a briefing about the experiment and its goals from the experimenter, participants completed 36 trials in a laboratory setting, in each of which they were presented with two curtains on the computer screen, and had to guess which one hides a picture. If the participant chose the correct (target) curtain, the 'reward image' was revealed, otherwise, a uniform grey background was revealed. Importantly, the target side (left or right) was determined randomly by the computer after the participant's guess. This randomization was done with replacement, so in each trial, the target side was completely random. The reward image for each trial was also determined after the participant's guess by randomly selecting from a pool of 36 reward images without replacement. The pool contained 18 erotic and 18 non-erotic images, resulting in 18 'erotic trials' and 18 'non-erotic trials'. The randomization of the target side and the reward images were independent of each other. The outcome of the trial was whether the participant successfully guessed the target side. The erotic images were selected from the erotic subset of the NAPS image set [28], while the non-erotic images were from the IAPS [29].

Participants also completed two brief questionnaires: one about belief in and experiences with ESP, the other one related to sensation-seeking. We included these questionnaires to match the original protocol by Bem as closely as possible, because it is unclear whether exposure to these questionnaires would alter the outcome of the study. However, data from these questionnaires are not used in hypothesis testing. Just like in the original experiment, an image of the starry sky appeared and participants got relaxation instructions before the experimental trials began: 'For the next 3 min, we would like you to relax, watch the screen, and try to clear your mind of all other thoughts and images.' The same image appeared between each trial for 3 s, and participants were instructed to 'clear their minds for the next trial' during this time.

There were a few deviations from the protocol of the original Bem [3] experiment: multiple participants could be tested in the same experimental space at the same time; compensation of participants could differ between data collection sites (e.g. sites had the option to offer university credit for participation, monetary compensation or no compensation at all); and we used a different set of erotic images than in the original study for legal reasons. All of these deviations were accepted by the consensus panel.

At the time of joining the study, all experimenters and site principal investigators (PIs) completed the Australian Sheep-Goat Scale (ASGS) in English, assessing their belief in the paranormal and self-reported paranormal experiences [30]. These data were used for descriptive purposes.

## 2.3. Methods for demonstrating and monitoring protocol fidelity

In this section, we list the methodological tools used to ensure and demonstrate protocol fidelity in the present study, and the rationale for using them. Importantly, these techniques were not used in the original study. One of our goals in this project was to assess whether the findings of the original study would replicate with these added safeguards against bias and methods improving transparency. In this project, we implemented a large number of methodological tools that can be used to increase credibility and to ensure protocol fidelity, and we describe their usefulness below. We decided to use all of these techniques in combination in our particular project because we undertook a replication in a highly contentious topic. While some of these techniques may not be applicable to or necessary for

every research project, all of them are topic-neutral and can be adopted in most areas of psychological research.

### 2.3.1. Direct data deposition

Direct data deposition means that during the study, incoming data is immediately and directly routed to a trusted version-controlled third-party repository (GitHub). This method provides insight into the data flow of the study and contributes to the transparency of the study process. Due to version controlling, each state of the data file is stored, providing a permanent record of the history of the data that is unalterable by the researchers. Thus, all data manipulation (e.g. exclusion of outliers, creation of derived variables, etc.) becomes registered and remains reversible. Overall, the application of direct data deposition allows researchers to be open about data acquisition, the flow of research sessions and participants, data management and pre-processing decisions. Moreover, it can prevent data loss, and allows for a seamless transition into data sharing, in our case, continuous data sharing from the start of data collection (born-open data). Online studies sometimes store data in version-controlled third-party databases. We argue that this feature should be used not only for internal laboratory purposes but also to demonstrate data integrity for other researchers. Free public repositories such as GitHub or OSF are ideal for this purpose; available data can be easily shared with others, and the researchers and the repository owners are financially independent. For an earlier implementation of a form of direct data deposition, see [31].

### 2.3.2. Born-open data

Data that are made public as they are being collected are referred to as 'born-open' data [31], which has been listed among the best practice transparency guidelines [32]. Accordingly, our GitHub repository was (and still is) open access. The data appeared as tpp_live_results_from_[date].csv at https://github.com/gy0p4k/transparent-psi-results, with new data automatically clipped to the end of the appropriate data file as it was being deposited. This repository is also linked to the open materials on OSF. Born-open data provided real-time insights into the flow of data collection in our project. This way, anyone could follow the research sessions in roughly real time through the data, save a copy of the data at any stage of the project, or re-analyse the data at any time, providing further authenticity to our data and analysis results.

One difficulty with born-open data is that sharing data in real time can result in some participants' data becoming identifiable for those who know when a given person participated in the study. We overcome this issue by encrypting timestamps and by depositing data in batches of 200 rows at a time, containing data from about five participants, to maintain concealment of participants' personal identity.

Rouder [31] described how their research laboratory set up an automatic periodic (daily) push of the contents of a shared local drive containing research data collected at their laboratory to a public GitHub repository. Rouder labelled this process as creating born-open data. However, we believe it is useful to explicitly distinguish direct data deposition (meaning immediate and direct saving of data into a version-controlled third-party repository) from born-open data (i.e. the data is made public immediately as it is being collected). Direct data deposition is meaningful in its own right without data sharing because it provides a tool through which the integrity of the raw data can be demonstrated. In some situations, researchers may not want or be able to share data; nevertheless, they can still use direct data deposition to substantiate claims about their data integrity (e.g. that no data were deleted, modified or excluded during analysis). So here, we define born-open data a little differently from Rouder, referring only to the immediate data-sharing aspect without direct data deposition.

When combined with direct data deposition, born-open data does not provide much added bias-reduction compared with post-data-collection data sharing. However, as an extra benefit, it guarantees that the data will not stay in the file drawer, as data are shared continuously, and it allows for immediate secondary use of the data instead of delaying months or years until publication.

### 2.3.3. Real-time research report

Real-time research reports are automatized reports that are continuously updated as data flows in. The results of our study were displayed and updated in real time as the data were being accumulated via our ShinyApp at http://178.128.174.226:3838/TPP_follow_results/ (the R code of the Shiny App

is available here: https://github.com/kekecsz/shiny-server/tree/master/TPP_follow_results), and the report's GitHub page: https://github.com/marton-balazs-kovacs/transparent-psi-manuscript. By providing a real-time account of the research progress and the accumulation of evidence, a real-time report provides further transparency at the study execution stage. Some might argue that real-time reports are redundant when born-open data is also present. While a dedicated expert might be able to reproduce the same report from born-open data and a preregistered code and analysis plan, providing automatic processing of the data allows anyone, not only the industrious and statistically adept, to receive a summary report of the accumulated evidence so far. This accessibility makes real-time research reports ideal for making the project transparent for the public and the media, and also for facilitating citizen-science projects. Furthermore, this type of reporting can represent the constant accumulation of evidence and the updating of our beliefs due to this evidence better than the practice of publishing at the end of the study. This reporting format is also consistent with the framework of continuous evidence updating as opposed to drawing dichotomous, all-or-nothing type of conclusions. Variations of real-time research reports can be found in other disciplines, such as the Live Synoptic Survey Telescope project that provides real-time alerts of new discoveries, which mostly targets other researchers in the field, or the real-time collision feed of the ATLAS project, which mainly aims to engage the public [33,34]. The combination of direct data deposition and automated analysis and reports may also improve the integrity of studies using sequential stopping [35].

Note that born-open data and real-time research reports can reveal study outcomes to research staff and participants; thus, this technique should be avoided in projects where naive experimenters or participants are desired, or in these cases some additional safeguards may be required [36].

### 2.3.4. Laboratory logs

Laboratory logs are records of laboratory activities, events and metadata related to experimental sessions. In our study, experimenters prepared session logs for each research session. These logs included start date and time, number of participants present and documentation of relevant events and protocol deviations, and actions taken in response to these if any. We also recorded server logs containing information about scheduled and unscheduled server access and relevant unexpected events (e.g. server downtime or crashes during sessions). The session logs and server logs are shared together with the other open materials on OSF. Furthermore, we automatically collected metadata about each session, including date and time (hashed), client system information, laboratory ID, experimenter ID conducting the session, and ASGS score of the experimenter and site-PI at the laboratory, which are stored in the main data file. Laboratory logs provide important information about protocol fidelity, because they can inform the PIs, and ultimately the readers of the research report, of any deviation from the prescribed protocol and about contextual factors recorded in the metadata. Some of this process was automated in our study, such as the server logs and the metadata collection, but the session logs were written by the experimenters; thus, human factors can introduce bias in those records. Nevertheless, we believe that these logs contributed important information about protocol fidelity and increased research credibility. Furthermore, laboratory logs can assist in early identification of problems in protocol fidelity and can help with flagging data that were affected by unexpected events, mistakes or protocol deviations. This information can also be used to fine-tune the data collection protocol or the exclusion criteria after a feasibility or pilot study. See Schnell [37] and Rouder et al. [13] for further information on the possible contents and benefits of laboratory logs.

### 2.3.5. Manuals, checklists and video-verified training

Following Bem's original protocol, participants completed the experiment in the laboratory, where an experimenter provided pre- and post-experimental briefing and supervised the session. We used several interventions to ensure that the experimenters delivered the study protocol as intended, which was especially important in a multi-laboratory experiment with many experimenters involved. First of all, we provided a step-by-step manual for experimenters, and a checklist that they could use to ensure each instruction is followed. Furthermore, we verified that all of the experimenters are well-trained and capable of following the instructions in the manual. To achieve this, each experimenter performed a scripted trial session after their training. This session was video recorded and submitted to the principal investigators of the research project (Z.K. and B.A.) who evaluated the success of protocol delivery based on a standardized checklist. Experimenters were asked to continue training

and submit a new video until they demonstrated as-intended protocol delivery. The video recordings were reviewed by the external auditors as well, who included their evaluation of these recordings in their final report. These videos were not released to protect the identity of the experimenters, but the evaluation of these videos by the auditors is openly available in the final audit report. Furthermore, we shared one such recording on OSF from an experimenter who consented to have their video available. Sharing such recordings has the added benefit of improving exact replicability, since they provide rich contextual information regarding how the protocol was delivered [14]. Similar procedures are being used in other multi-laboratory projects to decrease between-laboratory variation in protocol execution, although utilizing all these three features (manuals, checklists and video-verified training) within one project is uncommon.

### 2.3.6. Feasibility and pilot studies

Before launching data collection, we tested the adequacy of the research protocol and software in a smaller-scale study before the actual full-scale research, conducted with 184 participants in two laboratories. Using experiences from this pilot study, we made minor changes to the instructions for experimenters and to the experimental software to address issues and inconsistencies reported by the participants and the experimenters. A document containing laboratory logs of the pilot study and the resulting amendments to the experimental protocol and materials is shared on OSF. A well-documented pilot study can improve protocol fidelity because the protocol used in the study is already tried and tested, which decreases the likelihood of unexpected events causing bias or introducing researcher degrees of freedom. By sharing information on the amendments to the protocol in response to lessons learned during the pilot study, we aimed to demonstrate the impact of the pilot study. For more details on pilot and feasibility studies, see van Teijlingen & Hundley [38] and Lancaster *et al.* [39]. Pilot studies are quite common in psychology research. However, they are often poorly reported, so their potential for improving study integrity is not fully realized.

### 2.3.7. External research audit

External audit refers to the delegation of the task of assessing certain aspects of research integrity to a trusted third party. An IT auditor and two research auditors independent of the laboratories involved in the study were also involved in the project. The IT auditor was responsible for the evaluation of the integrity of the software and data deposition pipeline used in the project. The research auditors were responsible for evaluating protocol delivery and data integrity. These external auditors published reports about the project after data collection ended. Information about the auditors, their tasks and responsibilities, and their reports is accessible via OSF (https://osf.io/fku6g/). The practice of external audit is common in interventional medicine research, especially pharmaceutical research. However, formal external audit is almost unheard of in psychological science. The closest thing in the field of psychology is the stage-2 registered report review, but whether and to what extent the scope of the stage-2 review includes a systematic audit is currently unknown. The total transparency approach used in this research provides an opportunity for anyone to verify the credibility of the findings, but reviewing all the open materials takes considerable effort and some expertise. Accordingly, some voices in the field have advocated for supplementing peer review with a formal research audit [40,41]. Delegating the task of assessing certain aspects of research integrity to a trusted third party provides an added layer of assurance to those who do not review the materials themselves. This approach also allows for materials that cannot be openly shared due to confidentiality (in our case, the recorded trial research sessions) to still be used to demonstrate protocol fidelity.

### 2.3.8. Preregistration and registered reports

After acceptance of the manuscript as a registered report, and before any data collection, we preregistered the study on OSF (https://osf.io/a6ew3). The benefits of preregistration and registered reports are discussed extensively elsewhere (e.g. [42–45]).

### 2.3.9. Open materials

Material necessary to replicate our study and our analysis are available via the Materials component of the project on OSF (https://osf.io/3e9rg/). This component includes experimental materials, instruction

manuals for experimenters and analysis code. Stimulus images used in the study can be obtained free of charge for researchers from the administrators of the NAPS and IAPS image sets.[2]

### 2.3.10. Tamper-evident technology for fraud protection

The experimental software we used was protected from external manipulation by being located on a cloud-based server instead of being run on local computers at experimental sites. The laboratories and participants interacted with the data collection software via a Web-browser. Each participant (irrespective of the laboratory where they completed the study) was measured using the same software running on a central server. Software code running on the server was version controlled and any changes were automatically tracked via git. The IT auditor was able to verify at any time that the software code was unaltered via assessing the tracked changes, and via comparing this to the instance of the code deposited in GitLab at the start of the project. The GitLab project was made public after data collection had finished (https://gitlab.com/gyorgypakozdi/psi). With the use of direct data deposition and born-open data, any manipulation of the data (e.g. modifying the server-side copy of the data) would have been clear and could be traced back at any time because of the audit trail kept by GitHub. Research auditors were also responsible for keeping track of data integrity. Fraudulent manipulation of the analysis was prevented by the preregistered and open analysis code. These measures in combination ensured that fraud attempts by manipulating the experimental software, data or analysis would be immediately visible.

## 2.4. Hypotheses

To determine whether Bem's original findings could be replicated when using the added methodologies controlling researcher and publication biases and mistakes, we contrasted the likelihood of our results under the following two models.

Model 1 (M1) assumed that humans' guesses about the future, randomly determined, position of a target have a success rate higher than chance when correct guesses are reinforced by erotic images.

Model 0 (M0) assumed that humans' guesses about the future, randomly determined, position of a target does not have a higher than chance success rate when correct guesses are reinforced by erotic images.

Hypotheses only refer to erotic trials, because the effect was only observed in these trials in the original experiment, so the hypothesis test was restricted to these trials (i.e. non-erotic images were kept to match the original study protocol, but data on these were not be used in the confirmatory analyses). Note that these were one-sided models, thus, statistical tests were equally one-sided. We used a one-tailed test because the goal of the analysis was to determine whether the overall higher than chance success rate reported in the previous studies [3,26] can be reproduced when known sources of bias are addressed by the methodologies mentioned above.

## 2.5. Analysis plan

We used four statistical tests simultaneously to contrast the two hypothesized models: a frequentist mixed-effects logistic regression, and three Bayesian proportion tests using different priors. We planned to only conclude support for a model if all four tests passed the predefined thresholds for supporting the same model; otherwise, we would conclude that the study did not conclusively support either of the two models. This hybrid multiple testing approach had the advantage of yielding robust and credible results for a wide range of readers with a variety of preferred statistical methods. We used a sequential analysis plan to conserve resources. The primary analyses were carried out when 37 836 (minimum sample size), 62 388, 86 958, 111 528 and 136 080 (maximum sample size) erotic trials tested in the study were reached. If all participants were to finish all 18 erotic trials, 2102, 3466, 4831, 6196 and 7560 participants would be included in each analysis, respectively. These analysis points were selected based on the simulations analysing the operational characteristics of the study (see below).

---

[2]We have deposited the images in a private OSF repository to preserve them for the future. If the original images become unavailable from the original sources (and provided that this would not involve violation of rights), we will share the images with other researchers upon request.

### 2.5.1. Stopping rules

If all four primary analysis tests were to support the same model (either M0 or M1) at any of the pre-specified analysis points, we planned to stop data collection and conclude that our study yielded strong support for that model. If this was not the case when reaching the pre-specified maximum sample size, we planned to stop data collection and conclude that the study did not yield conclusively strong evidence for either model. The conclusions drawn for each of these stopping rules were pre-specified during the consensus design. The above-mentioned stopping rules could only be triggered at the pre-specified sequential analysis points.

### 2.5.2. Mixed-effects logistic regression

We built an intercept-only mixed logistic regression model using the `glmer` function in the `lme4` package in R [46]. This allowed for a random intercept for participants to predict the outcome of the guess (success or failure) in the experimental trials. We computed the Wald confidence interval for the probability of success by computing the confidence interval around the point estimate of the odds ratio of success, and transformed it to probability using the logit to probability function. If the upper bound of the confidence interval (CIub) was lower than 0.51, this test would support M0. If the test did not support M0, we planned to check whether the lower bound of the confidence interval (CIlb) was greater than 0.5. If so, this test would support M1. Otherwise, we would conclude that the tests did not conclusively support either model. As a starting point, we would use a 99.5% confidence interval, and since we are using a sequential analysis plan, we planned to adjust the confidence interval to be wider with each test according to the Bonferroni correction. Because we conduct two mixed-effects logistic regression tests at each analysis point (CIub < 0.51 to determine support for M0 and CIlb > 0.5 to determine support for M1), the confidence intervals used at the five analysis points were planned to be 99.75%, 99.875%, 99.91667% and 99.9375% CIs, respectively.

### 2.5.3. Bayesian proportion test

Based on the total number of successful guesses during erotic trials, and the total number of erotic trials performed by all participants combined, we computed the Bayes factor (BF01), to show the change in the odds of the observed proportion of successes under M0 compared with M1 [47]. M0 assumed a success rate of 50% in the population, while M1 assumed a success rate higher than 50%. If Bayes factors (BF01) is lower than 0.04 (1/25), the test would support M1, while if it was higher than 25, the test would supports M0. Otherwise, we would conclude that the tests did not conclusively support either model.

To make our statistical inference robust to different analytical decisions regarding the priors, we computed three Bayes factors (BF01) based on three different prior assumptions about the probability of correct guesses under M1, denoted by three different beta priors (1. Uniform prior, 2. Knowledge-based prior, 3. Replication prior). In all of the Bayes factor calculations, we used a point-value model for M0 denoting that the chance of each successful guess was 50%. For M1 we specified three different beta priors on $p$, where $p$ was the probability of correct guesses. As we used one-tailed hypothesis testing, all of the prior distributions of M1 were restricted to values of $p$ between 0.5 and 1, that is, the interval (0.5, 1). Thus, the distributions were truncated at $p = 0.5$ and renormalized so that they integrated to 1.

(1) *Uniform prior:* In this calculation, we used a beta prior with the parameters alpha = 1 and beta = 1.
(2) *Knowledge-based prior:* This knowledge-based prior was originally proposed by Bem, Utts and Johnson [48], and thus, we refer to it as the BUJ prior. The prior was described as having a normal distribution with a mean at $d = 0$ (Cohen's $d$ effect size) and the 90th percentile at $d = 0.5$. To match this prior in the binomial framework, we used a beta distribution with alpha = 7 and beta = 7. This distribution has 90% of its probability mass distributed between 0.500 and 0.712, where $p = 0.712$ is equivalent to $d = 0.5$ effect size. (We used the formula 'logodds = $d \times pi/\sqrt{3}$' to convert $d$ to log odds ratio [49]. Then, log odds ratio was converted to probability using the inverse-logit formula: '$p = \exp(\text{logodds})/(1 + \exp(\text{logodds}))$'. Thus, the final equation for getting $p$ from $d$ was: '$p = \exp(d \times pi/\sqrt{3})/(1 + \exp(d \times pi/\sqrt{3}))$').
(3) *Replication prior:* We computed a replication Bayes factor (BF01) (e.g. [50]), where the prior was a beta distribution with alpha = 829 and beta = 733. These parameter values were based on data gathered in Bem's original Experiment 1: 53.1% success rate in 1560 erotic trials, meaning 828 successes and 732 failures.

### 2.5.4. Pooling data across participants

It is important to note that in the Bayesian proportion tests we used the total number of successful guesses on erotic trials, pooled across all participants, and the total number of erotic trials performed by all participants combined. This approach assumes that all data points are independent, and is necessary to guarantee that premature stopping strategies cannot introduce bias into our data. Pooling repeated measures data across participants is not common in psychology, because repeated observations from the same participant usually hold information about each other. Failure to account for this dependency can lead to inflated inferential error rates. The dependence is usually treated by collapsing data within participants (e.g. computing the average success rate for each person). However, this strategy is biased by premature stopping strategies. For example, a person could stop early when they have achieved an above 50% average success rate. This could be handled by restricting analyses to data coming from fully completed sessions, but then a person could stop before finishing the session when success rate is below 50%, this way 'removing' several unsuccessful guesses from analysis. Thus, simple within-participant averaging is not viable.

Using a mixed-effects logistic regression is a viable alternative, since the contribution of each participant's estimate to the population effect estimate is weighted by the number of trials each participant completed. However, the parameter estimates can be still slightly biased if premature stopping strategies are used.

An approach that offers complete protection against bias from premature stopping strategies is to analyse the proportion of successes achieved in all trials combined using a proportion test irrespective of which participant contributed the data (using data from every trial, even if the session was stopped prematurely). But, this approach is vulnerable to another kind of bias: if some people are systematically better at guessing the position of the target than others, inferential error rates can be inflated. Note, however, that systematic personal differences can only occur if there is an ESP effect. So the sensitivity to detect higher than chance success rate can decrease, and the probability to falsely accept M0 can increase. If there is no ESP, every guess has a 50% success chance and no systematic personal differences are possible, and thus, the chance for falsely accepting M1 (falsely showing support for the ESP model) is not affected.

As shown above, each approach has its caveats. This is the reason we applied the hybrid approach described above: our mixed effect logistic regression takes into account the possible dependence of data points within participants, while our Bayesian proportion tests protect against bias from premature stopping strategies. We only claim support for M0 or M1 if all primary analyses support the same model. This way, we increased the robustness of our inferences and protected against the biases inherent in each technique separately.

### 2.5.5. Robustness test

At the conclusion of the study, we carried out two alternative variants of the Bayesian proportion tests used in our primary analysis: (1) a frequentist proportion test and (2) a full-Bayesian hypothesis test using parameter estimation. The first one was requested by the consensus design panel because this was one of the original statistical approaches used in Bem's original report [3], while the second was suggested to incorporate the other currently popular Bayesian hypothesis testing method [51].

(1) As part of the frequentist proportion test approach, two proportion tests were conducted. The first test assessed whether we can reject the null model of 51% or higher successful guess chance against the alternative model of lower than 51% successful guess chance in the population. If the null model is rejected, this would be interpreted as support for M0. If the null model was not rejected, we planned to conduct another test to assess whether we could reject the null model of 50% or lower successful guess chance against the alternative model of greater than 50% successful guess chance in the population. If the null model is rejected, this would be interpreted as support for M1. A significance threshold of 0.005 was used in these tests [52].

(2) We also carried out a Bayesian parameter estimation of the main parameter value, chance of successful guesses, updating our belief about the parameter compared with data collected in the original study. Thus, we used the same beta prior as in the case of the replication Bayes factor described above (alpha = 829 and beta = 733). The updated distribution with the results of the current study (alpha = 829 + the observed number of correct guesses and beta = 733 + the observed number of incorrect guesses) yielded the posterior distribution. We report the mode and the 90%

highest density interval (HDI) of the posterior distribution. Furthermore, we used the region of practical equivalence (ROPE) method for hypothesis testing [51]. We have determined the ROPE using simulations so that we have a reasonably high chance to detect a very small effect size if M1 is true, while still being able to correctly detect no effect if M0 is true. Through this process, the ROPE was determined to be 0–50.6% chance of successful guesses. (This area is different from the threshold of equivalence used in the NHST) robustness test mentioned above because the interpretation of the results of these tests is different). We will calculate the posterior mass of the parameter that falls inside and outside of the ROPE. If more than 95% of the posterior falls outside the ROPE, we would consider that support for M1. If more than 95% of the posterior falls inside the ROPE, we would consider that evidence supporting M0. If neither of these criteria is fulfilled, the result of this test would be considered inconclusive.

The robustness of primary findings to different statistical approaches is commented on in the Discussion and Conclusion sections of the article, but the primary hypothesis test inference is not changed by the robustness test results.

### 2.5.6. Exploratory analysis

The Consensus design panel suggested including an exploratory analysis regarding the distribution of the correct guess rate of participants. In this analysis, we contrasted the empirical distribution of the observed successful guess rate in the study with the distribution of a random sample of 18 000 000 simulated observations (1 000 000 simulated participants) taken from a population with a homogeneous 50% successful guess rate (which we will call 'expected distribution'). We visualized these distributions as overlaid histograms, and we quantified the difference between the distributions using the Earth mover's distance (EMD; also known as the Wasserstein metric). This analysis might inform future research about potential personal differences if they exist. If ESP exists and it is an ability of the participants, people might differ in the extent of their ESP abilities. For example, there might be a subgroup of people who have strong abilities, and others with weak or no abilities, this might show up as a heavier right-hand side tail in the empirical distribution compared with the expected distribution of the observed successful guess rate. We only used data from participants who finished all 18 erotic trials in this analysis. The results of this and other exploratory analyses are discussed in the results and Discussion sections. However, they do not replace confirmatory analyses, and their findings do not influence the conclusions or the abstract of the paper, which were all predetermined in the stage-1 registered report for each possible study outcome.

## 2.6. Sample size determination

### 2.6.1. Expected effect size

Bem's original Experiment 1 yielded 53.1% correct guesses in 1650 trials involving erotic images. However, later meta-analyses suggested that the effect might be lower than this in the population (Bem *et al*. [26]; Bierman *et al*. [53]; Rouder & Morey [21]). Based on these data, the sample size and analysis plan was determined so that the primary analysis of the current study would support M1 if the true correct guess chance is 51% or higher in the population, or support M0 if the true correct guess chance is 50% or lower in the population, with a high probability for correct inference.

### 2.6.2. Operational characteristics

The operational characteristics (power and inferential error rates) of the design and analysis plan were analysed with two different simulation-based methods. In method 1, we simulated experiments by drawing random samples from populations where the true correct guess chance was set at either 45, 48, 49, 49.5, 49.8, 49.9, 50, 50.1, 50.2, 50.3, 50.4, 50.5, 50.6, 50.7, 50.8, 50.9, 51, 51.1, 51.2, 51.5, 52, 53 or 56%. Each of these scenarios was tested in 5000 simulated experiments. However, the populations where the true correct guess chance was 50% and 51% were of particular interest for the performance of our analysis plan, so we simulated 10 000 experiments with these scenarios to increase precision. In method 2, instead of simulating a fixed effect size, the true correct guess chance in the population was sampled randomly from a beta distribution with parameters alpha = 829 and beta = 733 (based on the findings of the original study) in each of 10 000 simulated experiments [54]. Using these methods we determined the probability of making correct, incorrect or inconclusive inferences. In all of the above-

mentioned simulated scenarios the guess chance was homogeneous in the population (each individual had the same guess chance). However, the performance of our model was also tested assuming personal differences in guess chance. In the scenario where average population guess chance was set to 51%, for each participant we computed a personal guess chance by adding a random number to this population average drawn from a normal distribution with a mean of 0 and a standard deviation of 0.15. This means that we simulated a population where people have different abilities to guess the location of the target correctly, with 10% of the simulated sample being extremely lucky (greater than 70% chance of success), and 10% of the population being extremely unlucky (less than 30% chance of success).

These simulations assuming no systematic individual differences yielded that the sampling and analysis strategy described above have a greater than 0.95 probability to correctly support M1 if the true correct guess chance is 51% or higher, or M0 if it is 50% or lower, and the probability of falsely supporting M1 and M0 is lower than 0.0002 and 0.001 respectively. Our strategy would be able to correctly support M1 with a decent accuracy even if the true correct guess chance was 50.7% in the population (correct inference rate = 88%). However, its sensitivity drops off if the effect is lower than this, and if the true correct guess chance in the population was 50.2% or lower, the study would falsely indicate strong support for M0 in more than half of the experiments. Thus, we need to be aware that extremely small effects might be unnoticed by our study. However, importantly the probability of false support for M1 always remains very low, with $p < 0.0002$. The simulations assuming large personal differences indicated a 0.9 probability to correctly support M1 if the true correct guess chance was at least 51%, and the probability of falsely supporting M0 still remained below 0.002 in this case. The analysis of operational characteristics and its results are detailed in the 'Analysis of operational analysis' section of the Supplement.

## 2.7. Treating missing data

Data from all completed erotic trials were included in the analysed data pool, regardless of the total number of trials completed by the participant. This is to prevent bias from premature stopping strategies or data otherwise missing systematically related to the performance of the individual (see above).

## 2.8. Inclusion of data in analysis

All data were entered into the data analysis that is collected during the main study (not the pilot study), except for data generated during system tests. (The experimenter ID(s) of the test account(s) were specified in the preregistered analysis code.) The pilot data were not combined with the data collected in the main experiment and were not used in the confirmatory analyses. No other data were excluded from analysis for any reason.

## 2.9. Methodological considerations specific to the study of extrasensory perception

There are specific methodological issues related to the possible existence of ESP effects. Some of these issues such as the 'psi experimenter effect', the 'checker effect' and 'psi fatigue' may have implications in this design, and the method of randomization could also impact the interpretation of the results, so they were extensively discussed during the consensus design process. These issues are discussed in the 'Additional methodological considerations' section in the Supplement.

# 3. Results

## 3.1. Pilot study

The main aim of the pilot study was to test the experimental procedures, and to discover unanticipated events. The pilot sessions were conducted at ELTE (Budapest, Hungary) and University of Ottawa (Ottawa, Canada) with university students who received course credit as compensation. The sessions took approximately 20 min. Valid data was contributed to the pilot study by 184 participants. All participants completed all presented trials.

We found the experimental procedure feasible and acceptable for the participants. There were no documented accounts of a person refusing participation. Some issues were noted during the pilot sessions with regard to the experimental software, ambiguous instructions to experimenters, and the experimenter training process. We made adjustments to the instructions, training documents and the software to address these issues. No changes were made to the experimental protocol itself. The laboratory notes of the pilot sessions and the actions taken based on the experiences of the pilot study are shared on OSF.

## 3.2. Main study

Data analysis was carried out based on the preregistered protocol and analysis plan as described above. The analysis code is available in the Materials via OSF: https://osf.io/jdukb.

### 3.2.1. Changes compared with the preregistered protocol

There were no substantial changes to the preregistered protocol. Two minor changes occurred. One change was a correction of an error in the analysis code calculating a descriptive statistic about side-preference during guesses and proportion of the target sides. This is a non-crucial descriptive statistic that is not mentioned in the analysis plan, just in the descriptive results of the study. This statistic has no relevance to any of the confirmatory or exploratory analyses. The second minor change was related to the COVID-19 pandemic which coincided with data collection. At some data collection sites, more than two people were not allowed in a room at the same time. Since the trial session video that was used for the verification of the training of experimenters was designed for three people: two mock participants and the experimenter, we created a new trial session script that allowed for creating this video with only two people: one mock participant and one experimenter. We only allowed the use of this modified trial session script if the laboratory notified us that COVID restrictions prevented them from recording with three people in the room. These deviations are unlikely to have had a substantial impact on the results of the study. For more details on these changes to the preregistered materials, see the change log on OSF: https://osf.io/45e82

### 3.2.2. Site report

Laboratories serving as data collection sites were recruited via twitter posts, listservs and studyswap from 12 June 2018 to 31 December 2021. In total, 37 laboratories signed up for participating in the replication, 10 of which completed the enrolment process and collected data. The characteristics of the collaborating laboratories contributing at least one datapoint are listed in table 1. Data collection started on 10 January 2020 and ended on 29 April 2022. Most site-leaders had a low ASGS score indicating that they had low belief in the paranormal. Scores ranged between 0 and 23 with a mean score of 4.85 (s.d. = 7.31), and a median of 1.5. The experimenters had a more balanced distribution of ASGS scores. Scores ranged between 0 and 23 with a mean score of 6.07 (s.d. = 6.48), and a median of 5.

### 3.2.3. Sample and study characteristics

In total, 2220 individuals participated in the study. Among these, 2207 participants started the session before the study stopping rule was triggered. An additional 13 participants started the session after the stopping rule was met, but their data were not included in the analysis. Of those who started the session before the study stopping rule was triggered, 92 (4.17%) dropped out before providing valid data for the primary analysis (i.e. they declined participation, were ineligible, or stopped before the first erotic trial). Valid data was contributed to the primary analysis by 2115 participants, completing a total number of 37 836 erotic trials. The age range[3] of most (92.62%) participants was 18–29 years; 67.52% of participants identified as women, and 32.39% identified as men. The average score on the ESP belief item was 3.46 (s.d. = 1.09), and the average score of the sensation-seeking items was 2.71 (s.d. = 0.76). Both scales ranged from 1 to 5, with lower values indicating lower belief in ESP and lower sensation-seeking. Participants chose the left-side curtain in 49.08% of the trials (meaning that there was a slight right-side bias in participant choices), while the target side was left in 49.88% of the trials.

---

[3]To protect personal identity of the participants, we only collected information on age range, and not exact age.

**Table 1.** Characteristics of the Collaborating Laboratories. Note: Total N and Total # of erotic trials represent participants and trials which were included in the main confirmatory analysis (participants with at least one valid erotic trial); ASGS: Australian Sheep Goat Scale, ASGS scores in the table are total scores calculated by summing the scores of the individual items. ASGS total scores range from 0 to 36 with higher scores indicating greater belief in ESP. Where a laboratory had multiple Site-PIs, the ASGS score is the mean of the PIs' total scores. 'Not disclosed' means that the laboratory members did not give permission for that data point to be disclosed in order to protect personal data such as the ASGS score of a certain individual or the identity of the experimenters on the trial session video recordings.

| institution | country | language | total N | total no. of erotic trials | no. of experimenters | site-PI name | site-PI ASGS | mean experimenter ASGS |
|---|---|---|---|---|---|---|---|---|
| ELTE | Hungary | Hungarian | 944 | 16 930 | 6 | Balázs Aczél | 2 | 1.3 |
| not disclosed | Japan | Japanese | 101 | 1818 | 2 | not disclosed | 6 | 0.5 |
| Southern Illinois University | USA | English | 205 | 3690 | 1 | Kathleen Schmidt | 0 | 7 |
| not disclosed | not disclosed | not disclosed | 53 | 907 | 2 | not disclosed | 0 | 8 |
| Willamette University | USA | English | 149 | 2668 | 3 | not disclosed | 1 | 6.6 |
| University of Amsterdam | The Netherlands | English | 101 | 1803 | 4 | not disclosed | 11 | 11.8 |
| University of Ottawa | Canada | English | 59 | 1062 | 3 | Denis Cousineau | 0 | 8.3 |
| University of Tartu | Estonia | Estonian | 106 | 1908 | 2 | not disclosed | 0.5 | 0.5 |
| Iscte-University Institute of Lisbon | Portugal | Portuguese | 106 | 1862 | 2 | Patrícia Arriaga | 5 | 2 |
| University of Padova | Italy | Italian | 291 | 5188 | 1 | Patrizio Tressoldi | 23 | 22 |

Twenty-five (1.18%) of the sessions were terminated after starting the first erotic trial but before reaching the last erotic trial. This resulted in 192 missing data points among erotic trials. Note that in accordance with the preregistered analysis plan, trials from both completed and terminated sessions are included in the primary analysis. Additionally, three sessions' data were only partially included in the confirmatory analysis, because the preregistered analysis point was at reaching 37 836 erotic trials, and these three sessions were still in progress when this trial number was reached. The sessions were completed as all other sessions, but data were only included until the 37 836th erotic trial. Thus, data from some trials in these sessions were not in the confirmatory analysis.

Data collection was stopped because all four primary analysis tests support the same model (M0) at the first pre-specified analysis points. As mentioned above, the raw data were continuously open and updated in real time during the project via GitHub: https://github.com/kekecsz/transparent-psi-results/tree/master/live_data. With each server restart a new data file was generated. For ease of reuse, we have combined the raw data files into a single file named 'final_combined_live_data_for_analysis.csv' and shared this via our OSF page: https://osf.io/24qkr. No other changes were made to the data file other than simply concatenating the raw data files (in the correct time order) into a single data file; thus, this data file is still raw (unprocessed).[4]

### 3.2.4. Audit results, protocol deviations and unexpected events

The research auditors were Stephen Baumgart (University of California, Santa Barbara), and Michelangelo Vianello (University of Padova). The IT auditor was Luca Semenzato (University of Padova). The auditors did not note any substantial threats to the integrity of the study or its findings. Only minor issues were noted, some of which we discuss in the Limitations section below. The reports of the auditors can be accessed via OSF in the 'audit reports' folder, while the CVs of the auditors are in the 'auditor search' folder: https://osf.io/fku6g/. The auditors are not authors of this paper. Potential conflicts of interest may exist: one of the research auditors and the IT auditor work at University of Padova, where we also had a collaborating laboratory. The IT auditor has joint publications with the lead researcher of the University of Padova site.

There were no notable unexpected events or protocol deviations reported at the research sites, or on the server hosting the experimental software. No protocol deviations were reported by the site-PIs throughout the project.

### 3.2.5. Primary analysis

Overall, we observed 49.89% successful guesses within a total of 37 836 erotic trials. As a comparison, Bem reported 53.07% successes among 1560 trials in his original experiment. All four tests used for the primary hypothesis testing supported M0, thus the stopping rule was triggered with our minimum sample size achieved. This result means that the primary analysis as a whole yielded strong evidence in support of M0 (see figures 1 and 2),[5] meaning that the probability of correct guesses in this task was no higher than chance. As shown in figure 2, using a 99.75% CI, we estimated that the chance of successful guesses in the population falls between 49.11% and 50.67% (computed using the final mixed logistic regression within the primary analysis).

### 3.2.6. Robustness test

The proportion test assessing whether the chance of successful guesses is lower than 51% was significant ($p < 0.001$), and the test assessing whether the chance of successful guesses is greater than 50% was not

---

[4]This data file contains all recorded data during the live sessions. Note that the confirmatory analysis was run on the first 37 836 valid erotic trials as preregistered. This dataset includes a very few trials run with test accounts. The test account IDs were preregistered, see the analysis code on how to exclude these.

[5]Note that Bayes factors are interpreted as the degree by which some prior beliefs about the relative odds of two models are to be updated. This means that the Bayes factors computed in our study should not be interpreted on their own, rather, they should be used to 'update' the readers' prior beliefs about the relative odds of the two models. For example, a person who found M0 to be a hundred times more likely to be true than M1 before the study, after observing a BF01 = 1/25 at the end of the study could update their beliefs accordingly, and still think that M0 is more likely, but only four times compared with M1. Because we do not know the prior odds of the readers, the interpretation of the Bayes factors in this paper are written assuming that the reader believed the two models to be equally likely before seeing our results. If this is not the case, the reader should update their beliefs accordingly.

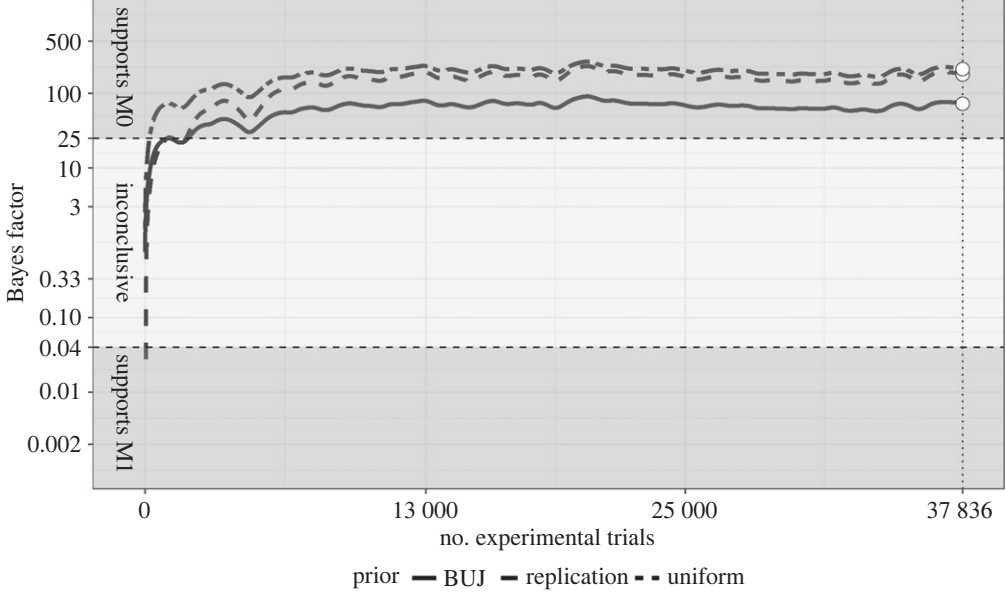

Bayes factor computed at each experimental trial

**Figure 1.** The figure shows cubic splines fit over Bayes factor values computed at each experimental trial. The solid, dashed and two-dashed lines represent the Bayes factors computed with the BUJ prior, replication prior and the uniform prior respectively. The dashed horizontal lines represent the predetermined thresholds for supporting M0 and M1. The dotted vertical line indicates the stopping points, where the Bayes factors were checked against the stopping criteria. The figure indicates that the Bayes factors using all three predetermined priors passed the BF01 > 25 threshold. This threshold has been passed relatively early during data collection, and the Bayes factors remained above the threshold throughout the rest of the study.

significant ($p = 0.665$). This result provides further support for M0, since we can reject the hypothesis that the probability of successful guesses in this task is higher than 51%.

The 90% highest density interval of the parameter was 49.57%–50.40%. The posterior distribution of the parameter chance of successful guesses is displayed in figure 2. More than 95% of the probability mass fell within the ROPE, which provided additional support for M0. Our results indicated that there is less than 1% (0.96%) probability that the chance of successful guesses in the population is higher than 50.6%. Thus, the findings of the primary analysis were supported by the robustness tests.

### 3.2.6.1. Exploratory analysis

Because both research auditors mentioned that there were some reports of Internet connectivity issues and software crashes in the laboratory notebooks, we performed the main analysis on a sample where we excluded all participants who did not have 18 erotic trials (those with unfinished experimental sessions). The parameter estimates, confidence intervals and Bayes factors barely changed in this analysis compared with the main confirmatory analysis, and the interpretation of the results is the same. Average success rate among those who completed all trials compared with those who stopped before completing all 18 erotic trials was 49.90% and 46.79%, respectively. However, only 25 participants stopped the session prematurely, so we cannot draw any conclusions from this comparison.

In the next set of exploratory analyses our goal was to explore a potential alternative explanation of the at-chance hit rate in the sample. There are some articles in the field of parapsychology which claim that the average hit rate at chance level in the sample as a whole is produced by a bimodal distribution of two distinct subgroups: unexpectedly lucky or talented individuals who consistently perform at higher than chance accuracy, and unexpectedly unlucky individuals who consistently perform at lower than chance accuracy [55]. This is often referred to as 'positive psi' and 'negative psi', and, since the performance is thought to be linked with belief in ESP, the consistent positive performers (and believers in psi) are called 'sheep', while the consistent negative performers (and ESP sceptics) are called 'goats'.

Figure 3 displays the distribution of successful guess rates of participants overlaid on the expected distribution of successful guess rates if M0 is true (the guess rate in the population is 50%). As per our preregistered analysis plan, we only included those participants in this exploratory analyses whose full set of 18 erotic trials were part of the main confirmatory analysis. Thus, we excluded participants with

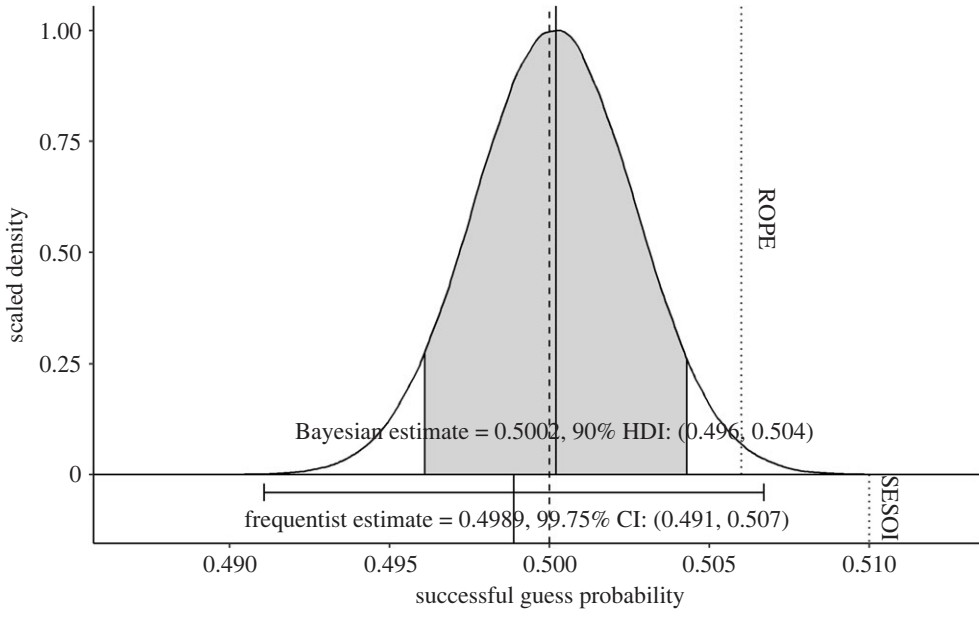

**Figure 2.** The figure shows the results of the mixed effect logistic regression (bottom) and the Bayesian parameter estimation robustness test (top). The density curve shows the posterior distribution derived from the Bayesian parameter estimation analysis with the 90% highest density interval (HDI) overlaid in grey. The horizontal error bar represents the 99.75% confidence interval (CI) derived from the mixed effect logistic regression in the primary analysis. Both of these are interval estimates for the probability of correct guesses in the population. The dashed vertical line represents 0.5 probability of correct guess chance, the dotted vertical line on the top represents the threshold of the region of practical equivalence (ROPE) used in the Bayesian parameter estimation (0.506), while the dotted vertical line on the bottom represents the threshold of the equivalence test (smallest effect size of interest, SESOI) used in the mixed effect logistic regression (0.51). The figure indicates that both the Bayesian parameter estimation and the frequentist mixed model support the null model, with the estimates very close to 50%, and falling well below 51% correct guess probability.

incomplete sessions and participants who were completing their session when the target trial number was reached (total sample size = 2087). The difference between the theoretical and the observed distributions was EMD = 0.037. The visual inspection shows no substantial deviation between the two histograms, which does not indicate uneven distribution of guess chance.

We explored the sheep-goat hypothesis further in a set of *post hoc* exploratory analyses by calculating the correlation between the performance of the individuals in their odd and their even experimental trials. If there are individuals who consistently guess below or above chance level, there should be a positive correlation between the odd and even trial performance. The correlation was $r = 0.026$ (95% CI: −0.017, 0.069), which is very close to 0 and does not seem to support the sheep-goat hypothesis. To investigate the possibility of experimenter sheep-goat effects on performance, we also examined the relationship between the participant's performance (successful guess rate) and the ASGS score of the experimenter present during the session. We built a linear mixed model predicting the successful guess rate of the participant with the ASGS total score of the experimenter as a fixed effect predictor and a random intercept of the experimenter ID. The same analysis was done separately with the site-PI's ASGS total score as the fixed predictor and site-PI ID as a random intercept. The parameter estimates corresponding to the effect of the ASGS score were very close to zero in both of these analyses, providing no support for the sheep-goat hypothesis (experimenter ASGS score effect estimate: −0.0003 [95% CI: −0.001, 0.001]; site-PI ASGS score effect estimate: −0.0002 [95% CI: −0.001, 0.001]).

## 4. Discussion of the replication study

In this study we replicated the protocol of Bem [3] Experiment 1 utilizing methodological innovations to improve transparency and credibility. The protocol was amended and approved by a group of experts in a consensus design process. Data collection was carried out in 10 laboratories from nine different

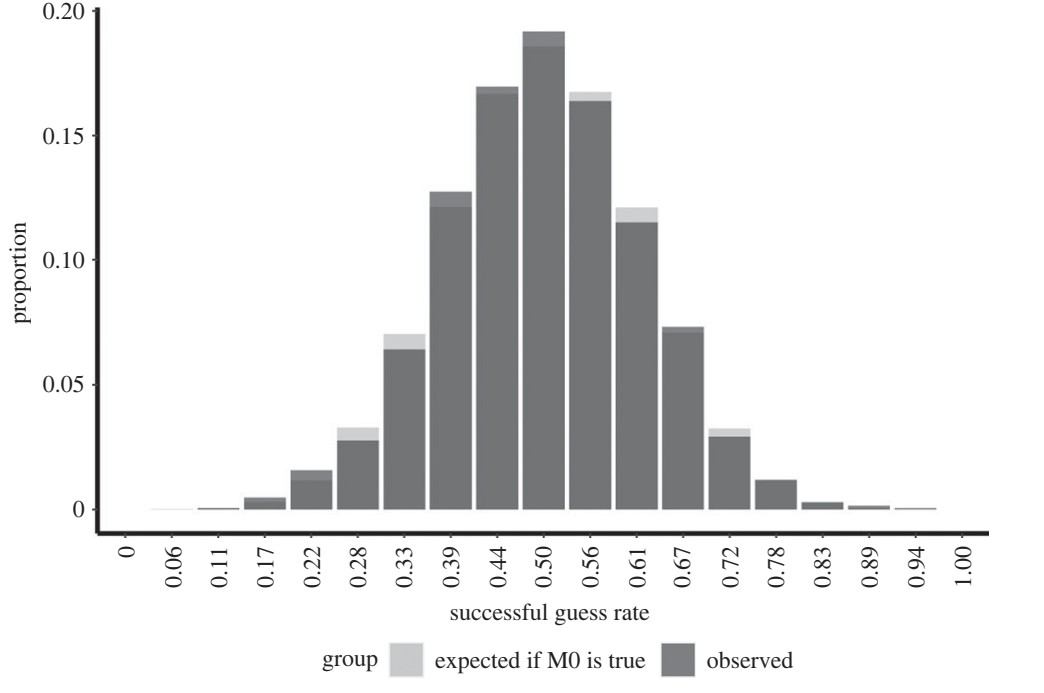

expected and the actual observed correct guess rate of participants

**Figure 3.** The figure displays the expected (light grey) and the actual observed (dark grey) correct guess rate of participants. The distribution of the expected guess rate was determined by drawing 18 000 000 samples (1 000 000 simulated participants) from a simulated population where the probability of successful guesses was set to 0.5, and the success rates in this sample were calculated. The figure does not show a substantial deviation between the expected and the observed distribution of successful guess rates, which lends further support to the hypothesis that participants guessed at chance (50%) accuracy in this task, and that there are no subgroups of unexpectedly lucky or talented (sheep) or unexpectedly unlucky (goats) participants in the population.

countries, during which 2115 participants provided valid data for analysis. The aims of this study were twofold; first, the project aimed to demonstrate the implementation and assess the usefulness of a set of credibility-enhancing tools. Second, we aimed to test the prediction of the 'pure bias theory' of parapsychology via the replication of Bem's 2011 Experiment 1. In this section we discuss the findings of the replication study; we discuss our experiences related to the methodological tools in the final section of the paper.

Multiple individual studies and meta-analyses report positive findings in parapsychology [16,26]. Most parapsychology researchers interpret these findings as evidence for the existence of ESP (also known as psi, or anomalous cognition). We call this the ESP model of parapsychology, and according to these recent meta-analyses, the model predicts that participants should have a higher than 50% chance of correctly guessing the target side in the paradigm used in Bem's 2011 Experiment 1. On the other hand, most sceptical researchers think that these positive findings are the result of biases, errors and mistakes associated with methodological inadequacies and the publication system. We call this the pure bias theory of parapsychology. This theory predicts that after eliminating the possibility of researcher and publication biases, and lack of methodological rigour, the previously shown positive findings in this paradigm would not replicate. In the replication study, we aimed to investigate which of these two models would be more successful at predicting the outcome. Since we found a probability of successful guesses very close to 50%, our study provides strong support for the prediction of the pure bias theory, and contradicts the prediction of the ESP theory.

One objection that the proponents of the ESP theory might voice is that this is only one study against many which—overall—support the existence of ESP. The recent meta-analysis by Bem *et al.* [26] listed 14 studies that investigated precognition reinforced by a positive reward, 11 of which produced a positive effect size, and reported an overall positive effect across all the studies. Nevertheless, our study's conclusions were based on 2115 participants; this sample was more than 20 times as large as that in the original experiment and over twice as large as the total number of participants in the 14 studies included in the meta-analysis combined. We are aware of two additional replications of Bem's 2011

Experiment 1 that were conducted since the 2016 meta-analysis. One was published as an honours thesis ($n = 106$) [56], and the other as a PhD thesis ($n = 208$) [57], and neither of them found a significantly higher than chance success rate. None of these 16 previous studies involved the level of methodological rigour and transparency that was applied in this project. The only other preregistered replication study that we are aware of was conducted by Wagenmakers and colleagues [58], who found support for the null model, similar to our experiment. However, that experiment was conducted independently of Daryl Bem, who later criticized the protocol for using an inadequate stimulus set. Conversely, our study protocol was based on a consensus of a panel of experts, which involved Daryl Bem as well as other notable researchers who have contributed to the debate surrounding this original study (for more details on the composition of the panel, recruitment and eligibility for the panel, see the 'Consensus Design Process' section of the Supplement). All in all, the evidentiary weight of our findings should be relatively high compared with previous replication attempts.

We explored alternative explanations that could be compatible with the ESP theory even when the observed successful guess rate is near 50%. As explained previously, the sheep-goat theory argues that the normal population consists of subgroups, one of which includes people who are consistently lucky or talented in ESP (also called sheep), and another which includes people who are unlucky or who produce negative ESP effects (also called goats; Storm & Tressoldi [55]). If the 'positive psi effect' produced by the sheep is roughly equal to the 'negative psi effect' produced by the goats, these effects would cancel each other out in the pooled data. Our exploratory analysis did not reveal any indication of such subgroups of participants in our data. Since the sheep-goat theory associates positive and negative psi effects with belief in psi and scepticism [55], we also explored the potential association of belief in psi and performance in the experimental task. We did not find any indication of such an association in our data.

Even though the predictions of the pure bias theory were confirmed over the prediction of the ESP theory, our results cannot rule out the existence of ESP itself, and the goal of the project was not to prove or disprove its existence. In fact, some theories of ESP are compatible with our finding (e.g. [59]). What we can conclude is that the original finding by Bem in this experiment is likely to be simply an artefact, and that this paradigm is unlikely to yield evidence of ESP if it does exist. This conclusion is all the more important because the study design that we replicated was the one that yielded the highest effect size in the 2016 meta-analysis. The fact that this effect was unreproducible, even with the input of Daryl Bem and more than a dozen other parapsychological researchers during protocol planning, should make readers cautious regarding that and other similar meta-analytical findings, which are mainly drawn from studies that are conducted without preregistration or using other best practices in experimental research. Due to its controversial claims, ESP research should probably apply the highest possible standards to reduce methodological bias and error, and to limit researcher and analyst degrees of freedom. However, our recommendations for increasing credibility do not only apply to ESP research, but to biomedical and social sciences in general. We should raise the standards of credible original research, and increase the standards for including studies in meta-analyses.

## 4.1. Limitations of the replication study

There were some deviations from the original Bem [3] protocol, most of which were made necessary given the large sample size requirement to achieve sufficiently low inferential error rates, such as testing multiple participants at the same time, and differences in the compensation of participants. We also used a different set of erotic images as reward stimuli. We see these deviations as theoretically irrelevant to the purpose of the experiment, which is also verified by the fact that the consensus design panel approved these deviations.

Individual differences cannot be effectively uncovered by this experimental design because of the low number of trials per participant. Our exploratory analysis did not indicate the possibility of such differences, but this question should be investigated with a different experimental design where more data is gathered from each individual. Furthermore, if the true guess chance is higher than 50%, there may be some moderating factors that are not controlled in this study design, such as belief in ESP. While we did not include belief in ESP in the main confirmatory models, we ran *post hoc* exploratory analyses on the potential influence of this parameter on performance, and there was no indication of any effect.

Just like in the original study, we only provided two options for sex and four options for sexual orientation, because the erotic images were not validated in other populations. We are aware that this

range of options is not sensitive to the full diversity in gender identity and sexual orientations. For some participants who did not belong to these categories, the presented images might have been less likely to evoke sexual arousal, although the impact of this on our results is probably small.

The IT audit report noted that the server did not correctly log the authorized login accesses. Specifically, the log configuration on the server was left on default settings, which caused the log file to be overwritten every few days. This was not intended and was only noticed at the end of data collection. This made it impossible to verify that the server access log kept by the PI is accurate and full. Nevertheless, the IT auditor noted that due to the redundancies in verifying the integrity of the experimental software and data, it is still very unlikely that there would have been any undetected modifications in the experimental software and the data.

Both research auditors noted that there were some reports of software crashes and Internet connectivity issues reported during the experiments. Although these could have contributed some noise to the data when re-analysing the data by excluding all non-finished experimental sessions, the results remain almost identical and the interpretation of the results is the same as in the main confirmatory analysis.

# 5. Conclusion of the replication study

In this section we provide a brief conclusion of the replication study results. Stating this conclusion is important because the conclusion below was pre-written and approved during the consensus design process (other versions of this conclusion were also created for each possible outcome of the study; these can be seen in the stage-1 registered report).

This study was designed to have greater than 0.95 probability of correctly detecting a very small correlation between human guesses and future random events (51% or higher correct guess rate) if that model is true and greater than 0.95 probability of correctly supporting the null model if it is true. The study methodology included public preregistration of the planned analysis and all associated methodological decisions that could affect the study outcome, formally documented software audit, measures to prevent and detect fraud and researcher biases, and making the data and study information publicly available with full transparency.

The data were more consistent with the model assuming that humans' guesses about the future, randomly determined, position of a target do not have a higher than chance success rate, rather than the model assuming that they do. We have observed 49.89% successful guesses within 37 836 trials. Observing this percentage of successful guesses is 72 times more likely if the guesses are successful at random than if they have a better than chance success rate. Taken at face value, the data provide strong evidence that in this experimental set-up the probability of successfully guessing later computer-generated random events is not higher than chance level contrary to what was previously reported by Bem [3] and others (for a collection of studies using the same paradigm see Bem *et al*. [26]).

The findings of this study are not consistent with the predictions of the ESP model in this particular paradigm. The methodology of the present study reasonably addressed all alternative explanations stemming from deficiencies in modal research practice [60] that we were able to identify, with extensive input from other researchers. The failure to replicate previous positive findings with this strict methodology indicates that it is likely that the overall positive effect in the literature might be the result of recognized methodological biases rather than ESP. However, the occurrence of ESP effects could depend on some unrecognized moderating variables that were not adequately controlled in this study, or ESP could be very rare or extremely small, and thus undetectable with this study design. Nevertheless, even if ESP would exist, our findings strongly indicate that this particular paradigm, used in the way we did, is unlikely to yield evidence for its existence. The results proved to be robust to different statistical approaches, increasing our confidence in our inference. Our parameter estimation indicates that there is a more than 99% probability that the chance of successful guesses in the population is lower than 50.6%.

# 6. Discussion of the methodological tools

The primary aim of this project was to develop a set of tools that would allow researchers to demonstrate and verify the credibility of research findings. Here, we report our experiences in implementing and utilizing these methods in the transparent Psi project.

## 6.1. Consensus design process

The consensus design process was utilized to ensure the robustness of our study design and to improve the acceptability of our findings among the stakeholders in the field. Completing the consensus design process took roughly nine months, including everything from the planning and preparation of the process to finalizing the protocol based on the process results and the debriefing of the panel members. This is probably too large a time-commitment for a typical research study where serious *post hoc* criticism or opposition is not expected. On the other hand, such a process could be essential to the acceptance of the results in cases where the hypotheses or study design are controversial in a field with adversarial research groups or theoretical camps. The actual usefulness of the consensus design process used in the TPP will only be assessable a few years after the publication of this paper; nevertheless, we intend to examine the effects with a survey sent out to the original panel members. We are in the process of writing up a step-by-step guide for implementing the consensus procedure used in this project [61].

## 6.2. Data handling

Thanks to using direct data deposition, the history of our data throughout the research life cycle was preserved and demonstrates that it remains in its raw form ever since it was gathered. This feature can easily dispel any doubts about the integrity of our data. Furthermore, thanks to born-open data, and real-time research reports, researchers who were interested in the study could follow the data accumulation and the development of the findings continuously during the project. This accessibility can provide further assurance to the continuous integrity of the data gathered in this study. One additional benefit of the real-time research report was to the core research team. We could use this real-time updating report to keep track of data accumulation in the study very easily without having to download and re-analyse the data every time, and to keep an eye on the reliability of the performance of the data collection program via the number of dropouts/discontinued sessions. Thus, this tool not only improves research credibility, it also serves as a good 'quality of life tool' to help principal investigators monitor their project.

A drawback of this data management approach is that its set-up requires programming expertise. We wrote a step-by-step guide on how to implement direct data deposition from Google Drive to GitHub, which could be used by researchers if they use Google Forms or other software that saves data to a folder that is synced with Google Drive. This guide was written in a way that a person with no prior programming knowledge can implement it. https://github.com/kekecsz/Direct_data_deposition_ guide/blob/master/Guide%20to%20set%20up%20Direct%20Data%20Deposition%20from%20GDrive% 20to%20GitHub.md Furthermore, we have open materials available on how to set up a simple real-time research report based on data collected through Google Forms in a shiny app here: https://github.com/ marton-balazs-kovacs/going_real_time. Our team is working on further solutions that would increase the accessibility of these data management tools to a wider group of researchers.

Real-time data sharing and real-time reporting might lead to personal identification of data. In cases where this is problematic, we suggest including safeguards to prevent this. For example, in our study, we only pushed data to GitHub after every 200 rows in our dataset (roughly after every five participants), and we hashed the time-stamp of completion. In our project, this was deemed sufficient safeguard to prevent personal identification of data. However, other safeguards may be necessary in projects with different study designs. Furthermore, real-time data sharing and real-time reporting may also influence research participants and researchers participating in the project. For example, it might lead to breaking of blinding of group allocation, it might lead to premature stopping of data collection by researchers due to lost motivation [36], or changes in behaviour of researchers delivering interventions. Researchers who want to implement such born-open methods need to consider these and other potential consequences, and need to implement safeguards to mitigate bias, for example through maintaining blinding in the data and report as well, using automation, defining comprehensive stopping rules prior to data collection, and so forth. There is also a risk that people who are following the study results via the real-time research reports will draw conclusions prematurely from interim results. In our study, we tried to mitigate this risk by placing a warning on the real-time report site saying, 'Result not yet final! Data presented on this page represent the current trend calculated from the data. The results should not be over-interpreted! Random variations may cause the data to cross the decision thresholds. Statistical decisions will only be drawn at the pre-

specified stopping points. The next stopping point will be at reaching X trials.' Researchers implementing real-time research reports might consider including similar warnings.

Importantly, direct data deposition works without born-open data and real-time reports, and allows for the verification of data integrity even if data is only shared after data collection was completed, or with the addition of a predefined automated analysis pipeline, even if the raw data is not shared at all. So in some cases where real-time data sharing or data sharing in general is problematic, direct data deposition can still provide benefits to credibility.

## 6.3. Laboratory logs

We used a laboratory notebook to monitor the fidelity of the research protocol. Since keeping laboratory notes is a practice that is familiar to most researchers, we will not describe it in detail, but we will mention some experiences with using them in our project. Instead of opting for a simple free-text laboratory log, we used a Google Form which included a list of pre-specified questions the experimenters had to answer immediately after each research session. The form contained questions about the research sessions such as date, time, number of participants who signed up and number of participants who actually showed up for the sessions. Additionally, we included the experimenter checklist in this form, which allowed the experimenters to note if all the steps specified in the protocol were delivered, or if not, which steps were missed. We also asked the experimenters to report any notable events such as software crashes and participants finishing sessions prematurely, and had some free-text fields where details could be provided about these or other unexpected events and about the actions taken in response to them. Furthermore, site-PIs were asked to submit reports of unexpected or important events that were not noted in these notebooks (but no such reports were submitted). This laboratory notebook allowed the core research team and also the research auditors to assess overall fidelity of protocol delivery in the study. Additionally, it allowed for accurate monitoring of data collection progress at collaborating laboratories.

Such a laboratory notebook is of course susceptible to human error and human biases, so even though it can reveal important information, it should not be taken as an objective indicator of protocol fidelity. As noted by Rouder *et al.* [13], as much of the notebook should be automated as possible.

Laboratory notebooks containing notes of each research sessions can be accessed on OSF via https://osf.io/myjsw. A pdf version of the laboratory notebook form that was used to fill out the notebook is accessible at: https://osf.io/kmqpr.

## 6.4. External research audit

Even though some researchers have called for the use of research audit in psychology, the adoption of these techniques has been minimal thus far [40,41]. There are multiple reasons for this, chief among them being that the invitation of such systematic and thorough scrutiny is not yet in the culture of social science research. This context limits knowledge about the availability and need for research audit, funding for such services, training routes that would prepare a researcher to be an auditor, and availability of such services in general. While we did contact audit firms about completing the research audit, retaining their services did not fit within our budget constraints, probably due to contract rates based on pharmaceutical and medical device trials with large budgets. As an alternative option, we asked the consensus design panel members to suggest researchers who may be good auditors for our project. We reached out to these individuals and asked them to submit an application to be an auditor in our project. This recruitment method for auditors is suboptimal, since it can be biased, and can lead to potential conflicts of interest among the auditors. Our case further demonstrates the lack of availability of this service and the lack of appropriate funding for this service in psychological science.

There may be a perception among researchers that undergoing a research audit can only produce negative consequences. In the best case scenario, the auditors find no problems and have nothing to report. On the other hand if they do find inadequacies, those problems might decrease the impact of the paper or even invalidate it completely. In our view, there are several benefits to openly inviting such systematic investigation into the integrity of a research project, and receiving an approving review in the process. One of these benefits involves an assurance to the researchers involved in the project and the funders of the project about the integrity and validity of the approach used in the study, which far surpasses that of traditional peer review. Similarly, it can also serve as a powerful signal of credibility to the readers. As opposed to a classical peer review or simply using open

materials, the readers know which materials were checked, they know who checked them, and what they found, even if no issues were reported. Thus, the use of such formal, transparent and systematic research audit could be a worthwhile investment for improving the credibility and reproducibility of research in social sciences. The role of funding bodies is critical to increasing the adoption of this approach.

## 6.5. Tamper-evident technology for fraud protection

We have used a combination of automation and continuous version control to create a set-up where it is verifiable that our findings are derived from the raw data that were produced through completing the experiments pre-specified in our research protocol. Specifically, our data were recorded through a custom-made experimental software, which updated its own source code from the git repository's remote master branch every 5 min. This way, the data could not come from any other source than the latest version of the software on the git repository, which keeps an audit trail (version history) recording any changes in the software during data collection. As mentioned previously, the raw data collected by the software was immediately pushed to a git repository via the direct deposition process, thereby making apparent any alterations or deletions. Finally, the analysis code used for the confirmatory analysis was preregistered on OSF (and also made available on GitHub), ensuring that the analysis results were independent of human factors. In fact, even the conclusions of the study were pre-specified, adding another layer of safeguard so that the interpretation of the results is also automatic. Together, our methods provided an unbroken pipeline from data collection to data interpretation contributing to a high degree of credibility of the findings.

While this system made fraud virtually impossible or clearly detectable in our particular study, it has multiple caveats. Setting up this custom-made program required programming skills, and we hired a professional programmer to do it. Also, verification of data and software integrity takes time and requires some expertise. Furthermore, in our particular system, the verifiability was not extended to the participant-side. In theory, the whole study could have been completed by a single person, or a software agent. The verification that real participants took part in the study would take considerable investigative effort. In our study, exploiting this limitation would not have been possible in a way to produce excess positive results. In the experiment, people were basically guessing the outcome of a virtual coin flip before the coin is flipped. If ESP does not exist, it does not matter whether this guess is done by a single person, or 2000, or a software, the outcome of the coin flip would still be unpredictable. However, this set-up unfortunately leaves a few options for fraud in the negative direction. If someone wants to hide the existence of the effect, they could instruct participants to answer randomly, or according to a preset table of left or right answers, or they could take people out of the equation altogether, and write a program that would complete the experiment with pre-specified or random responses. This was not the case in our study, but our set-up does not allow us to prove this easily, which is a limitation of our current approach. We are working on a solution which extends the usability and verifiability of this solution to the average researcher, and which would extend to the verifiability of participants as well. Ideally, researchers could utilize a user-friendly experimental software system that has all of these features built in and running seamlessly in the background, so that using these credibility-enhancing solutions do not require any extra effort to implement.

## 6.6. Comprehensive documentation

We placed special attention to creating a straightforward folder structure for storing all of the study materials on the projects' OSF page (https://osf.io/3e9rg/) and to provide a detailed documentation of them. Moreover, we created analysis code that is clearly commented, allowing for the easy understanding of the purpose of the different code chunks. By these measures our aim was to enhance the understandability of the shared materials to increase the possibility of error-detection during project execution and facilitate their reuse after publication.

## 6.7. Other interventions to improve credibility and minimize mistakes and bias from human factors

In this project, we also used a pilot study, preregistration and open materials. The application of these tools are essential in research, but they have been discussed extensively elsewhere, so we will not discuss them in detail here.

The use of a manual for experimenters was crucial in our project, since we had to ensure that the protocol is executed in the same standardized manner in multiple research sites across nine different countries. Having a detailed manual was seen as the most efficient option to achieve this fidelity. In a single site study, it may be possible to forgo using manuals with simply direct training of the research staff or executing the research protocol by the PI. Nevertheless, even in such studies, having an experimenter manual can aid in minimizing mistakes and preparing for unusual events, and vastly increase the reproducibility of the study if made openly available. Perhaps the only time when a manual is unnecessary is in fully automated studies (e.g. survey studies with Web-based recruitment). In these types of studies, sharing a detailed research protocol, including the details of the recruitment process, and the experimental software or survey itself may be sufficient to allow for a very close replication. Just like manuals, checklists for experimenters are commonly used tools that can help in ensuring and maintaining standardized performance of research staff during the course of a research project, and there are very few reasons to omit them.

Less common is the use of video-based verification of training of experimenters. The video-recorded trial session allows the investigator to check preparedness of the research staff for data collection, but in principle, personally observing a mock research session could provide the same oversight. The main added benefit is that a recording also makes it possible to demonstrate this preparedness (and the rigorous training process) to people external to the project. In total, 47 trial session videos were submitted to the lead investigators of our project, out of which 20 were rejected. More than half of the experimenters' first trial session videos were rejected. The most common reason for rejection of the trial session videos was the apparent risk of the video unmasking participant sexual orientation.[6] Another leading cause for rejection of trial videos was downplaying the importance of personal performance in the experiment, providing incorrect or confusing description of the experimental task to participants, or providing incorrect interpretation of the results or the end-of-study feedback screen for participants. Minor deviations from protocol were also common, like forgetting to put the 'experiment in progress' sign on the door, or not telling participants how to indicate that they are finished with the experiment. This high rate of unsuccessful videos was a surprise to us. We expected to see fewer failed attempts because of the detailed experimenter manual, checklist and trial session instructions. We made clarifications in the manual and the checklist after seeing the first few rejected videos to help reduce the prevalence of these issues.

In hindsight, the video-verified training was a great tool in this multi-site project to prevent unprepared experimenters from delivering the protocol and to standardize the level of preparedness for protocol delivery across the data collection sites. Language barriers may have also contributed to the high number of first rejections. We asked experimenters to either do the trial session video in English, or if they did it in a non-English language to provide an English transcript. Experimenters may have chosen to record in English to avoid the hurdle of having to transcribe and translate the video. For future projects we advise lead investigators to ask for videos specifically in the language that the experiment will be delivered in (with translated transcript) without the option to submit the video in another language, to avoid this issue.

We also conducted a code review, during which an external person who was not involved in the project previously, Pietro Rizzo, tested the analytical reproducibility of the findings by running the analysis code and cross-checking the results with the data reported in the manuscript. Furthermore, he also checked the comments in the analysis code and made suggestions for improvements.

## 6.8. Cost-benefit analysis

Up to this point, we have focused on discussing the benefits and limitations of the credibility-enhancing tools. In this section, we will enrich this discussion by evaluating the costs and the perceived usefulness of these techniques. We hope this analysis will make it easier for the reader to decide which of these techniques to implement in their own research or institution.

Our study was supported by Bial Foundation (grant no. 122/16) via a grant of 43 000 EUR. Roughly 30 000 EUR of this budget was spent on salaries of the coordinating research team, roughly 7000 EUR

---

[6]In this project participants indicated their sexual orientation in the experimental software so that they could get erotic images that are appropriate for them. This means that an onlooker could uncover the sexual orientation of the participant by seeing the types of erotic images they get in the experiment. Experimenters were instructed to take multiple precautionary measures to prevent this, but many times the trial session video still displayed some risk of unmasking participant sexual orientation. In these cases, the video was rejected and the experimenters were instructed specifically on how to avoid this.

was spent on contracts with a software developer and the three research auditors, and the rest covered other costs such as conference and publication. However, this grant only covered a portion of the total costs associated with the project. The project was made possible by generous support of volunteer work and explicit or implicit subsidies from the institutions of the researchers who were involved in the project; thus, estimating the total dollar cost of the research is difficult, especially given regional differences in labour costs. Instead, we provide an estimate of the work hours that are associated with implementing and carrying out each highlighted credibility-enhancing method so readers can understand the amount of labour it requires.

In table 2, we provide a non-comprehensive summary of the benefits associated with each technique, and estimates for two scenarios: how many work hours would be devoted to the specific methods (i) if researchers were to implement these techniques in an average single-site experimental psychology study, or (ii) if researchers were to implement the same techniques in a larger scale, multi-site experimental psychology study. We used this approach because the costs of some of the methods are dependent on the scale of the study. For example, laboratory logs have a time cost for each research session since the experimenter has to manually complete the log, the video-verified training depends on the number of experimenters involved in the study, and so forth. For the single-site study, we calculated with eight consensus design panel members, two experimenters running the study sessions, 100 data collection sessions, one research auditor, and one IT auditor. For the multi-site study, we calculated with values close to that of the present study: 30 consensus design panel members, 30 experimenters, 1000 data collection sessions, two research auditors, and one IT auditor. We did not log work hours spent on these methods in our project, so these estimates are based on hindsight estimates. Also, note that the time estimates do not account for the learning process of implementing these techniques for the first time or mistakes and restarts that might be associated with the first-time implementation of any sophisticated system. The time estimates do not include the time spent on finding the right programmer and the auditors either. Tables S4 and S5 in the Supplement give more details about the calculations behind the estimated work hour costs.

These estimates indicate that by far the most expensive methodological tool in the toolkit is the consensus design procedure, which involves several hundred hours of work divided across assistants, coordinating researchers and panel members. It is also apparent from the time estimates that the automatized tools, such as direct data deposition, born-open data, real-time research reports and tamper-evident software, scale very well, since they mainly require one-time implementation. Thus, their cost-benefit ratio might be more favourable for large-scale projects compared with the other tools in the toolkit.

We were also interested in the perceived usefulness of these research tools by the researchers who were exposed to them during the study. Therefore, at the end of the study, we ran an internal survey of members of the data-collection laboratories excluding researchers at ELTE where the coordinating research team resides. Fourteen responses were submitted, nine respondents were site-PIs, three were experimenters, one was both a site-PI and an experimenter, and one had another role (not specified). We asked the respondents how useful they would rate each of the credibility-enhancing methodological tools showcased in our study (possible ratings: useless, limited usefulness, somewhat useful, very useful), and how likely they see themselves using these techniques and methods in future projects (possible ratings: very unlikely, somewhat unlikely, somewhat likely, very likely). In this survey, laboratory logs, manuals for experimenters, checklists, preregistration and open materials were rated 'very useful' by more than 70% (10 or more out of 14) respondents. Most of the tools were rated 'very useful' by the majority of respondents, except for direct data deposition and tamper-evident software, but even these were rated as 'somewhat useful' or 'very useful' by more than 70%. One respondent indicated that both real-time research reports and training verification video recordings were useless. No other 'useless' ratings were observed. More than 70% of the respondents indicated that they are 'very likely' to use experimenter checklists, preregistration and open materials in future projects. On the other hand, for born-open data, real-time research reports, external research audit and tamper-evidence software, at least 50% of the respondents said they are 'somewhat unlikely' or 'very unlikely' to use them in the future. This is arguably a limited and biased sample, but this survey can still give some insight into the perceived value of these methodological tools by researchers who were recently exposed to all of them. For more details, see tables 3 and 4.

# 7. Conclusion about the methodological tools

Based on our experiences with implementation and use of these tools in practice and the results of the cost-analysis and our internal survey, we can sort the methodological tools into three tiers. Tier A:

**Table 2.** Cost-benefit analysis of the credibility-enhancing methods. Note: for the single-site study, we calculated with eight consensus design panel members, two experimenters running the study sessions, 100 data collection sessions, one research auditor, and one IT auditor. For the multi-site study, we calculated with 30 consensus design panel members, 30 experimenters, 1000 data collection sessions, two research auditors, and one IT auditor.

| research tool | benefits | cost in work hours | | | | | |
| --- | --- | --- | --- | --- | --- | --- | --- |
| | | single-site project | | | multi-site project | | |
| | | assistant | researcher | programmer | assistant | researcher | programmer |
| consensus design | protection against *post hoc* criticism, robustness of the scientific product, increases acceptability among stakeholders | 160 | 224 | 0 | 320 | 560 | 0 |
| direct data deposition | demonstrates data integrity (with or without data sharing) | 2 | 1 | 4 | 2 | 1 | 4 |
| born-open data | transparency during the data collection process, earlier reusability, demonstrates data integrity | 3 | 3 | 8 | 3 | 5 | 6 |
| real-time research report | easy monitoring of study progress, transparency during the data collection process for a larger audience, demonstrates data integrity and consistency of conclusions over time | 2 | 8 | 8 | 2 | 8 | 8 |
| laboratory logs | easy monitoring of study progress, assess fidelity of protocol delivery, provide meta-data on the session level | 8 | 7 | 10 | 34 | 7 | 10 |
| manual for experimenters | increased fidelity of protocol delivery | 6 | 20 | 0 | 90 | 20 | 0 |
| checklist for experimenters | increased fidelity of protocol delivery | 1.5 | 3 | 0 | 16.5 | 3 | 0 |
| training verified by video recording | demonstrates fidelity of protocol delivery, can aid reproducibility | 2 | 8 | 0 | 45 | 26 | 0 |
| external research audit | assesses and demonstrates integrity of data collection and study procedures | 4 | 22 | 19 | 4 | 40 | 19 |
| preregistration | transparency about research and analysis plan | 4 | 48 | 0 | 4 | 48 | 0 |
| open materials | transparency about research materials, increased reproducibility and reusability | 4 | 8 | 6 | 4 | 8 | 6 |
| tamper-evident software | demonstrates fidelity of protocol delivery | 0 | 0 | 1 | 0 | 0 | 1 |

**Table 3.** Internal survey results—usefulness of methodological tools.

| research tools | very useful | somewhat useful | limited usefulness | useless |
|---|---|---|---|---|
| consensus design | 9 | 5 | 1 | 0 |
| direct data deposition | 7 | 6 | 2 | 0 |
| born-open data | 8 | 5 | 2 | 0 |
| real-time research report | 9 | 3 | 2 | 1 |
| laboratory logs | 11 | 3 | 1 | 0 |
| manual for experimenters | 12 | 2 | 1 | 0 |
| checklist for experimenters | 13 | 1 | 1 | 0 |
| training verified by video recording | 7 | 5 | 2 | 1 |
| external research audit | 7 | 6 | 2 | 0 |
| preregistration | 12 | 2 | 1 | 0 |
| open materials | 14 | 1 | 0 | 0 |
| tamper-evident software | 5 | 9 | 1 | 0 |

**Table 4.** Internal survey results—likelihood of future use of methodological tools.

| research tools | very likely | somewhat likely | somewhat unlikely | very unlikely |
|---|---|---|---|---|
| consensus design | 4 | 9 | 2 | 0 |
| direct data deposition | 3 | 8 | 3 | 1 |
| born-open data | 5 | 3 | 6 | 1 |
| real-time research report | 3 | 4 | 7 | 1 |
| laboratory logs | 9 | 2 | 3 | 1 |
| manual for experimenters | 10 | 4 | 1 | 0 |
| checklist for experimenters | 12 | 3 | 0 | 0 |
| training verified by video recording | 7 | 4 | 2 | 2 |
| external research audit | 2 | 5 | 7 | 1 |
| preregistration | 13 | 1 | 0 | 1 |
| open materials | 15 | 0 | 0 | 0 |
| tamper-evident software | 0 | 7 | 6 | 2 |

laboratory logs, manual for experimenters, checklists, preregistration and open materials are very useful for most researchers and almost always worth the invested time and effort. Tier B includes consensus design and direct data deposition. They are generally useful, but require significant investment. The consensus design requires a lot of time and can delay the start of research projects compared with traditional research trajectories. Nevertheless, they can still be worthwhile in research areas where the field is divided, for controversial research questions, or for expensive projects where the researchers want to mitigate the risk of the results being dismissed or invalidated by *post hoc* criticism. Setting up direct data deposition is not very time consuming, but it currently requires the involvement of a programmer in the research team and transitioning the data collection from a readily available commercial data collection program (or paper and pencil data collection) to a custom written program. If a willing programmer is available, this work is manageable; however, many traditional research laboratories may not find direct data deposition viable, especially for small-scale projects with no or limited funding. Direct data deposition could be made much more accessible with the development of a software handling this process or the integration of this feature into the commonly used data collection platforms; such advances would make this method viable for most research

studies. Tier C: born-open data, real-time research report, training verification video, external research audit and tamper-evident software all have their limitations as previously discussed that make them less broadly applicable to research projects. They may be still worth implementing in high-profile and/or controversial studies which require an increased level of verifiable credibility. Improvements and innovations in these methods might increase their viability in the future.

Ethics. Testing was conducted with the approval of Eötvös Loránd University Faculty of Education and Psychology Research Ethics Committee, with ethical permission 2018/13. In addition, each data collection site had ethics permission from their local ethics authority. The research was conducted in accordance with the Declaration of Helsinki.

Data accessibility. The authors declare that all data, code and materials supporting the paper are available from the Open Science Framework: https://osf.io/3e9rg/. The data collection software code is available via GitLab: https://gitlab.com/gyorgypakozdi/psi. The transparency checklist report [62] is available in the Supplement.

The data are provided in electronic supplementary material [63].

Authors' contributions. Z.K.: conceptualization, data curation, formal analysis, funding acquisition, investigation, methodology, project administration, resources, software, supervision, validation, visualization, writing—original draft, writing—review and editing; B.P.: conceptualization, funding acquisition, resources, writing—original draft, writing—review and editing; B.S.: conceptualization, funding acquisition, resources, writing—original draft, writing—review and editing; P.S.: investigation, resources, writing—review and editing; M.Z.: investigation, resources, writing—review and editing; M.K.: conceptualization, formal analysis, resources, software, visualization, writing—original draft, writing—review and editing; B.E.B.: investigation, resources, writing—review and editing; D.C.: project administration, resources, writing—review and editing; P.T.: project administration, resources, writing—review and editing; K.S.: investigation, project administration, supervision, writing—review and editing; M.G.: project administration, resources, writing—review and editing; T.R.E.: project administration, resources, writing—review and editing; Y.Y.: funding acquisition, investigation, project administration, resources, supervision, writing—original draft, writing—review and editing; J.K.M.: investigation, project administration, resources, writing—review and editing; H.L.: data curation, investigation, resources, writing—original draft, writing—review and editing; F.Y.: data curation, funding acquisition, resources, writing—original draft, writing—review and editing; D.D.: funding acquisition, investigation, writing—review and editing; J.P.R.: investigation, project administration, writing—review and editing; M.B.: investigation, project administration, writing—review and editing; R.S.: investigation, project administration, writing—review and editing; A.A.: investigation, resources, supervision; P.A.: investigation, project administration, resources, writing—review and editing; R.O.: investigation, project administration, resources, writing—review and editing; N.P.: investigation, project administration, resources, writing—review and editing; K.K.: investigation, resources, writing—review and editing; B.H.: investigation, project administration, writing—review and editing; S.W.: investigation, project administration, writing—review and editing; B.V.: investigation, project administration, writing—review and editing; K.G.: investigation, project administration, writing—review and editing; B.A.: conceptualization, funding acquisition, investigation, methodology, project administration, resources, supervision, validation, writing—original draft, writing—review and editing.

Conflict of interest declaration. We have no competing interests.

Funding. The project was funded by the Bial Foundation via grant no. 122/16. D.D. was supported within the framework of the Basic Research Program at the National Research University Higher School of Economics (HSE University) and a subsidy by the Russian Academic Excellence Project '5–100'.

Acknowledgements. We are grateful for the consensus design panel members for their contribution in designing the research protocol (including Daryl Bem, Dick Bierman, Robbie C.M. van Aert, Denis Cousineau, Michael Duggan, Renaud Evrard, Christopher French, Nicolas Gauvrit, Ted Goertzel, Moritz Heene, Jim Kennedy, Daniel Lakens, Alexander Ly, Maxim Milyavsky, Sam Schwarzkopf, Björn Sjödén, Anna Stone, Eugene Subbotsky, Patrizio Tressoldi, Marcel van Assen, David Vernon, Eric-Jan Wagenmakers, and those members who did not give permission to disclose their participation in the panel). We are also grateful for the external auditors of the project: Stephen Baumgart, Michelangelo Vianello, and Luca Semenzato, and for the contribution of Dana Arnold in the project. Furthermore, we are very thankful of the contribution of Simon Beaudry, Pari-Gole Noorishad; Alyssa LeGuerrier from University of Ottawa, and other experimenters who contributed to data collection in this project. We also thank Richard Morey for his assistance in developing some of the custom functions used in our analysis code, and Pietro Rizzo for conducting the code review at the end of the project. Furthermore, we are grateful for the work and helpful suggestions of the editor and the reviewers of the manuscript.

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
