## [Peer Review File · Royal Society Open Science]

Review History

RSOS-191375.R0 (Original submission)

Review form: Reviewer 1

Do you have any ethical concerns with this paper?

No

Recommendation?

Major revision

Comments to the Author(s)

Manuscript ID: RSOS-191375

Different terms are used in the literature and the current manuscript – psi, precognition, ESP – in this review I will simply use “ESP” to refer to the phenomenon under scrutiny.

Below I have made remarks under the six headings (marked with *stars*) that the journal requested that I comment on.

The significance of the research question(s)

Without a clear definition of “significance” and what criteria I’m supposed to use to evaluate it - I’m not sure what is meant by this heading so I won’t comment.

The logic, rationale, and plausibility of the proposed hypotheses

It might not be customary for reviewers to explicitly state their own beliefs about the phenomenon under scrutiny in a review. However, I think that might be relevant in this case due to the controversial nature of the phenomenon. I have a very strong belief that ESP does not exist. There is not only an absence of substantive theory or plausible mechanism – the purported effect defies the (comparatively well-established) laws of physics and common sense (how could casinos and ESP co-exist, for example).

Nevertheless, I am a subscriber to “Cromwell’s rule” - one should not categorically rule out anything, for this makes it impossible to learn (see Lindley, 1991, Making Decisions).

Consequently, I would be prepared to change my mind that ESP does not exist, its just that the evidence required for me to reach a favourable state of belief would be overwhelming.

Practically, this means that however well designed and executed the proposed study is, a favourable outcome for the ESP hypothesis would not be sufficient to convince me that the phenomenon was real (though it would move my belief incrementally in that direction). A more likely reaction from skeptics to a positive finding will be to question what other mechanisms may have given rise to the finding. This would be an appropriate Lakatosian defense (Meehl, 1990) given that the theory being defended (established laws of physics) have considerable “money in the bank” (a strong track record of empirical support).

I expect that my skepticism towards ESP is shared by many scientists (though the only empirical support for that statement I am aware of is Wagner & Monnet, 1979, cited in Bem, 2011), but my open-mindedness might not be. Some reactions to the publication of Bem (2011) suggested that it should have been desk-rejected because of the low likelihood of the topic under scrutiny (i.e., regardless of any methodological concerns). I am aware that similar discussions about the present manuscript have taken place and the fact I am being asked to comment on the plausibility of the proposed hypothesis suggests that this is a relevant criterion in the editor’s accept/reject decision making process. Consequently, I feel it is necessary to (briefly) make the case for why this manuscript *should not be* rejected on the grounds that the phenomenon under scrutiny is highly implausible to many scientists. Here are three reasons:

(1) It is possible (though probably rare) that phenomena that are a priori implausible, subsequently turn out to have some degree of accuracy. In the present case, this naturally seems highly unlikely, but it is possible. Thus, if ESP is real, rigorous scientific scrutiny of the kind proposed here will be necessary to establish its existence and improve our understanding of it.

(2) For implausible phenomena that are in fact not real, exposing them to robust empirical scrutiny may help to convince those who believe the phenomena are actually plausible. In the present case, I would speculate that the number of people who believe in ESP (some of whom may be in a position to substantially influence the lives of others) is not insignificant, and the actual or potential negative impact of them holding that belief (if it is false) could be quite serious. Thus, if ESP is not real, rigorous scientific scrutiny of the kind proposed here may help to convince some that their beliefs are unfounded.

(3) Finally, scientific methods do not give us direct access to the truth of the world around us. They are a set of tools that help us overcome the limitations of our own cognition and sensory apparatus. It is therefore possible that even if ESP is not real, evidence consistent with its reality is obtained using scientific methods. In such a scenario, the integrity of those methods would be call

into question and efforts could be made to improve them. This was of course the dominant reaction of the scientific community to the Bem (2011) study. It also seems appropriate (see the points above about Lakatosian defence; Meehl, 1990).

There are several specific objections that I think could be raised against the study that are reasonable and should be considered. On balance, I don't feel they outweigh the arguments made above, but are worthy of consideration. The objections are that the study is potentially (1) inefficient (poor use of resources); and/or (2) dangerous (due to the negative consequences of a false positive).

(1) It may be inefficient to use resources testing an implausible hypothesis when they could be used for greater benefit elsewhere. Relevant resources include participants' time, researchers' time, funding, editors and reviewers time, financial aspects etc. It seems reasonable that decision making is relegated to the relevant parties i.e., participants can decide if its worth their time (and are also compensated), the researchers have obviously decided its worth their time, funders have decided its worth their financial investment, and peer review is underway. A systematic efficiency assessment of such projects (those evaluating low plausibility hypotheses) might be useful but is beyond the scope of this specific case.

(2) The study is potentially dangerous due to the negative consequences of a false positive. Even if ESP is not real, a positive effect could be obtained because of chance statistical variation or some unanticipated experimental confound or bias. More than reasonable effort has been made to avoid this possibility, but the implications of a false positive should still be considered. If such a situation arises, I imagine that many within the scientific community will rigorously scrutinize the finding and evaluate it in its proper context (i.e., consider the low prior plausibility). But outside of scientific argumentation, the implications of such a result being published in a respected journal like *RSOS* could have serious negative repercussions. The authors have been (admirably) balanced in their write up of the article, but I do wonder whether the scenario of a positive result needs additional commentary, particularly in the abstract. Such commentary would reinforce that the finding needs to be evaluated in context i.e., that an individual with a highly skeptical prior should continue to think that the phenomenon is highly implausible, even given this positive result. It unclear if the authors would add this to their partly drafted discussion section, but there is currently little discussion of the fact that a positive finding would be contrary to well established physical laws – to such an extent that (probably for most scientists) the most likely explanation of a positive result would be a undetected methodological artefact, not ESP. The editor might also want to consider the journals response to a positive finding – for example, accompanying the article with an editorial (as with Judd & Gawronski, 2011) and/or peer commentary (as with Wagenmakers et al. 2011) intended to provide appropriate context. Any press release would also need to be very carefully worded to convey that, despite a positive finding, ESP remains vanishingly unlikely. There is something of an 'appeal to numbers' and 'appeal to authority' about what I am suggesting, which would probably be inappropriate in purely scientific argumentation, but may be warranted when we consider that the general public might rely on such cues to evaluate the credibility of the finding.

Interestingly, testing the ESP hypothesis is actually framed as being somewhat secondary to the main goal of the paper – which is stated to be a case demonstration of several research tools that can improve transparency during study execution. This is a bit odd, because the results of the study will not tell us much about the effectiveness of the credibility tools – that is not the hypothesis under scrutiny. I expect this unusual framing arose from the collaboration of skeptics and non-skeptics. It doesn't seem overly problematic; but it is a little confusing to follow the line of argument when its framed this way.

The authors describe how there has been much attention to improving the transparency of study planning and reporting of studies, but less focus on study execution, a situation they refer to as the 'transparency gap'. The authors make a good case for why this study in particular could benefit from enhanced credibility though a little more elaboration on why these specific tools

might address specific concerns with ESP researcher might be helpful (e.g., there seems to be a suggestion that enhanced protection against fraud is necessary, but it is really explained why).

[NB – it may be worth noting that parapsychology research has a historical precedent for driving methodological reforms. For example, it seems to have been the impetus for an early prototype of the Registered Reports model (see Wiseman et al. 2019).]

The soundness and feasibility of the methodology and analysis pipeline (including statistical power analysis where applicable)

The attention to detail in this protocol is outstanding. Feasibility has been demonstrated empirically in a substantial pilot study conducted across two labs.

The statistical analysis plan seems appropriate and very thorough. The distinction between exploratory and confirmatory analysis is well articulated. Robustness checks are employed, and an interesting hypothesis will be explored in exploratory analyses but treated separately from the primary hypothesis / analysis. Unfortunately, due to my own knowledge limitations, I'd say my scrutiny of this aspect of the protocol is fairly superficial and I'd recommend additional scrutiny by a statistical expert (particularly an expert in Bayesian statistics) if that is not already the case.

I may have misunderstood, but there didn't seem to be a Bayesian analysis that takes into account a skeptic's v. low prior probability (see e.g., Wagenmakers et al. 2011). I also think it could be made clearer that even if there is statistical support for M1, this cannot necessarily be directly interpreted as evidence for ESP. Here's a comment from Jeff Rouder on Andrew Gelman's blog to that effect: "Perhaps this is too simplistic, but I think it's important to separate truth and Bayes factors. I believe I can learn from comparing models without any of them be true, and, in that vein, use Bayes factor as a comparison metric. If I am judicious and wise, the BF comparison of select models may point me toward the next step in understanding the phenomenon at hand." There are of course many other explanations for findings consistent with the M1 model other than ESP. Also, as argued by Fiedler & Krueger (2013), to assume a positive effect in this paradigm is evidence of ESP is to conflate the explanandum (the event to be explained) with the explanans (the argument used for the explanation). This doesn't seem to be addressed in sufficient detail in the current draft of the discussion (though perhaps this doesn't need addressing until stage 2 review).

Regarding the pre-study questionnaires about belief in ESP and sensation seeking – it seems sub-optimal to take these measures before, rather than after, the core experiment. Pre-study delivery will presumably increase the risk of demand characteristics – specifically, participants working out what the study hypothesis is and changing their behavior. Admittedly, it is not clear to me exactly how their behavior would change in a way that might affect the results of the study. For example, assuming psi is not real, even if participants know the researchers are studying psi, this should not alter their performance on the task because there is no deliberate strategy they can adopt to achieve anything other than chance performance. Nevertheless, if there is no good reason to conduct these measures before the experiment, my recommendation would be to perform them afterwards as a principle of good experimental design. The same goes for any hypothesis-blind experimenters. [NB – it's actually not clear to me if the participants and the experiments are supposed to be hypothesis blind or not, perhaps this could be made more explicit? Sorry if I've missed it]

Whether the clarity and degree of methodological detail would be sufficient to replicate exactly the proposed experimental procedures and analysis pipeline

This is likely to be the most transparent research project I've ever reviewed. The methods are clearly explained and the authors have said they will share relevant materials, such as instructions to experimenters. Additionally, they will provide at least one video of the experiment being performed which will aid any replication efforts.

A few comments/suggestions:

(1) “Stimulus images used in the study can be obtained free of charge for researchers from the administrators of the NAPS and IAPS image sets.” – this is not ideal from a reproducibility perspective because the long-term preservation of and access to these materials relies on a third party. If the system maintaining these image sets were no longer operating in 10 years time for instance, then it may no longer be possible to conduct a high fidelity replication of the study. I understand that there may be rights issues that prevent this, but would it be possible for the researchers to share these materials directly themselves?

(2) “These issues are discussed in the ‘Additional methodological considerations’ section in the Supplement.” – I don’t know where to obtain the supplement so I have not evaluated any part of it.

(3) The analysis code is super clear and well annotated. It has been shared on the OSF and already successfully tested on the pilot data. We can even see it operating for ourselves on the provided website. This is beyond excellent. One suggestion would be to take steps to ensure that the software environment in which the analysis is performed can also be preserved. There are several tools available that can help with this, for example Binder or Code Ocean. I’ve found Code Ocean quite straightforward to use.

Whether the authors provide a sufficiently clear and detailed description of the methods to prevent undisclosed flexibility in the experimental procedures or analysis pipeline

The descriptions in the paper seem very clear to me. Execution of the protocol is going to be filmed and audited by independent researchers. Assuming those procedures are robust. its hard to see how meaningful protocol deviations could occur and remain undisclosed.

The authors are eagerly constraining their own analytic flexibility by providing the exact code they will use to perform the analysis – that’s about as good as it gets.

Whether the authors have considered sufficient outcome-neutral conditions (e.g. positive controls) for ensuring that the results obtained are able to test the stated hypotheses

It is not clear what such a positive control would be in the present design as there is no intervention.

Miscellaneous minor comments

“replication is a safe but costly method for verifying credibility” – I’m not sure what is meant here by “safe”, could you clarify?

“The credibility enhancing methods described in this paper in combination with previously established transparency best practices can effectively negate or reveal all known sources of researcher and publication bias.” – that is a very strong statement and it does not seem warranted. It is very difficult to measure bias and demonstrate it has been reduced or negated. We know that existing measures, such as pre-registration, are far from 100% effective (e.g., Claesen et al. <https://psyarxiv.com/d8wex>). Some of the current measures (e.g., direct data deposition) have not been tested (as far as I know).

“Adult participants with no psychiatric illness who have not participated in the study before and who are not under the influence of drugs or alcohol at the time of the session will be eligible to participate.” – perhaps state that these are inclusion criteria so they cannot be later misused as post-hoc exclusion criteria.

“After receiving a briefing about the experiment and its goals” – who provides this briefing? Relevant whether it’s a hypothesis-blind experiment or hypothesis-aware PI, for example.

“not all participants will get the same compensation to facilitate accrual” – The meaning of this sentence, and the general procedures for compensating participants, was unclear to me.

Regarding direct data deposition (DDD) and born open data – is it theoretically possible for the code to be modified to allow a form of ‘man in the middle attack’ – for example, temporarily holding the data and enabling manipulation prior to deposition. This of course seems extremely unlikely – but the whole reason for thinking DDD is necessary is the potential for fraudulent interaction with the data – thus it seems reasonable to consider just how robust this system would be against a determined bad actor.

“Any manipulation of the data is clear and can be backtracked at any time because of the audit trail kept by GitHub” – this implies that manipulation of the data would be possible, but traceable. Could the authors elaborate on how the manipulation could occur?

A potential opportunity for bias – can the experimenters observe participants’ performance? If so, this could influence their completion of the lab logs which could lead to exclusions. Experimenters should ideally not be aware of the hypothesis and unable to monitor participants performance.

Perhaps an explicit statement that the pilot data is not being combined with the main data is warranted?

“The failure to replicate/successful replication of previous positive findings with this strict methodology indicates that it is likely/unlikely that the overall positive effect in the literature might be/would be only the result of recognized methodological biases rather than ESP/biases.” – I don’t think this statement is accurate for skeptics with a low prior belief in ESP. If the study does show an effect consistent with M1, then a skeptic with a very low prior belief will still not be convinced by this single study. The possibility of some unanticipated/unrecognized methodological artefact would remain a more likely explanation (for some).

Review form: Reviewer 2 (Brice Beffara-Bret)

Do you have any ethical concerns with this paper?

No

Recommendation?

Accept with minor revision

Comments to the Author(s)

All my comment are in the attached file (see Appendix A).

Review form: Reviewer 3

Do you have any ethical concerns with this paper?

Yes

Recommendation?

Major revision

Comments to the Author(s)

Please see the attached PDF for full comments (Appendix B).

Review form: Reviewer 4

Do you have any ethical concerns with this paper?

No

Recommendation?

Accept in principle

Comments to the Author(s)

Overall assessment:

Thank you for the opportunity to review the research proposal for the 'transparent Psi project'. I was very excited to read it. It is a carefully planned proposal that hits a soft spot at the right time – the debate around ESP following Bem's 2011 article and the subsequent replication attempts. The proposed research plan offers an impressive, extensive battery of tests and safeguards that will allow to unequivocally settle the debate. I am also a bit skeptic. My skepticism has nothing to do with the ability of the research team to execute the ambitious plan; on the contrary, I am convinced that the plan is adequate and the research team has the capacity to execute it. Rather, it is about the cost-benefit of the proposed safeguards and feasibility of implementation in future studies. I wonder if the time, effort, human labour, and dollar costs of such a study can be invested regularly in scientific investigations. My guess is that this type of concerted effort might be limited to a (very) small number of critical studies that attempt to resolve an ongoing dispute. Overall, this looks like a terrific project and I strongly recommend it goes ahead. I would like to see practical, smaller scale and cost-effective solutions to the credibility problem.

Significance:

Very high, given both the scientific and the public interest in ESP, and the ongoing controversy surrounding Bem's original study (2011) and subsequent replication or analysis attempts (eg Wagenmakers and colleagues, 2011, 2015).

Logic and rationale:

The hypothesis are broadly based on the original study, with M1 and M0 indicating support for better-than-random guessing, vs random. As such, they are adequate.

Methodology:

The proposed methods are sound and clearly explained. The proposal outlines a number of key features that serve to ensure the credibility of the findings. These include deposition of protocols and testing logs, born open data, other bias reduction, and increased transparency via third party observers. Each 'tool' has its own merit, and together they form an almost formidable wall to protect the integrity of the study (notwithstanding my concerns about the costs required for the implementation).

Other considerations:

- The proposal highlights the advantages of real time report, with which I wholeheartedly agree. In a nut shell, data is immediately deposited to a publicly open repository and the results of the study will be analysed as the data is accumulated (not at the end of data collection). This offers greater transparency. A potential risk of the real time report though is that spurious effects could look significant at some arbitrary point in time and members of the public or media might this snapshot in time to draw conclusions. Are there appropriate mechanisms to minimise that kind of risk?
- A potential risk of the born-open data is that sharing data in real time that compromise that anonymity of the participants. The research team had taken a number of measures (encrypted timestamps, group release) to minimise this risk and I have found the overall strategy satisfactory.

- The research team had offered a number of exit points for data collection, based on predetermined numbers of trials and subjects. Perhaps they would like to consider Bayesian Optional Stopping (e.g., Rouder, 2014, PB&R) or if not explain why it is not adequate.

Decision letter (RSOS-191375.R0)

03-Oct-2019

Dear Dr Kekecs,

The Editors assigned to your Stage 1 Registered Report ("Raising the value of research studies in psychological science by increasing the credibility of research reports: The Transparent Psi Project.") have now received comments from reviewers. We would like you to revise your paper in accordance with the referee and editors suggestions which can be found below. Please note this decision does not guarantee eventual acceptance.

When submitting your revised manuscript, you must respond to the comments made by the referees and upload a file "Response to Referees" in "Section 2 - File Upload". Please use this to document how you have responded to the comments, and the adjustments you have made. In order to expedite the processing of the revised manuscript, please be as specific as possible in your response.

Kind regards,
Andrew Dunn
Royal Society Open Science
openscience@royalsociety.org

on behalf of Professor Chris Chambers (Registered Reports Editor, Royal Society Open Science)
openscience@royalsociety.org

Associate Editor Comments to Author (Professor Chris Chambers):

Associate Editor: 1

Comments to the Author:

Four expert reviewers have now assessed the Stage 1 manuscript. Before summarising the reviews and the key points raised, I want to note for the record my own reasoning in handling this manuscript and deciding to send it for review. I do this especially in light of the fact that, if eventually accepted and published at Stage 2 with results, the open review policy of RSOS will lead to the publication of the reviews and my editorial action letters, as well as the potential publication of an editorial and/or commentaries. Given the controversial subject matter, the content of this discussion is therefore likely to fall, quite rightly, under public scrutiny.

I want to begin by making clear that upon receiving a presubmission enquiry from the authors

about this potential RR, my instinct was to decline a full submission. This stemmed from my belief that the phenomenon under investigation in this work (psi) is as close to impossible as anything that might be conceived in science, contradicting established physical laws (which I believe must constrain any evidence that could be obtained in psychological science), and therefore belonging firmly in the realm of pseudoscience. Moreover, it is my firm belief that IF any positive evidence for psi were to be observed in such a study, however rigorous, it (a) would almost certainly have arisen due to some form of experimental error, known or unknown, and (b) could risk significantly misleading the public about the likely existence of psi -- a point, as we will see, that was also raised by Reviewer 1.

However, I am also mindful that the role of a journal editor – and indeed a scientist more generally – is to remain skeptical not only about the truth/existence of unlikely theories or phenomena, but to remain skeptical about our own skepticism and not allow our biases to limit the advancement of science. I therefore consulted with editorial colleagues at the journal, and publicly (see <https://twitter.com/chrisdc77/status/1156946007839690754>) to determine whether my thoughts on this matter were well aligned with those of the scientific community.

As it happens, they were not. The chief editor of RSOS – a physical scientist – noted that if we, as editors, were sufficiently convinced by the rigour of the method then there would be nothing to fear by considering the study further, and potentially something to gain (a point well made in light of the framing of the proposal to demonstrate a new and more rigorous approach to conducting psychological science). In the same vein, the popular choice among the community on social media was that it should not be in the editor's prerogative to allow their prior beliefs as to the likely existence of a phenomenon to hinder research from being undertaken, which is of course an especially salient issue for assessing a Stage 1 RR, because a RR, unlike a regular article, provides the opportunity for the review process to hinder (though not outright block) certain research questions from being asked in the first place.

I did not fully agree with the arguments against my original position (and views were mixed), but the response gave me pause and on this basis I decided to send the manuscript out for in-depth review. This does not change my belief that psi almost certainly does not exist, and I admit I have taken the decision to review this paper with lingering apprehension given potential consequences for the reputation of the journal (and potentially the Royal Society) but, far more importantly, for misleading the public, IF evidence that could be construed in favour of psi were to be produced. None of these views are intended to bias or influence the reviewers or authors in their assessments or revisions, and I will do my best to put my beliefs aside in rendering any editorial decisions or recommendations. My position (and biases) are simply noted here as matter of record in the event that the manuscript progresses to Stage 2 acceptance and these reviews are published.

Turning to the reviews themselves, the assessments are mainly positive but there were four major issues identified by multiple reviewers. The first issue, raised by Reviewer 1 and 3, is whether (and how) the study will meet its desired aim of testing the credibility of the ultra-transparent research tools and methods being proposed when there is no comparison to regular methods. The second issue, raised by Reviewers 1 and 2 is whether the authors have taken sufficient steps to eliminate experimenter bias and ensure adequate blinding. The third point, raised by Reviewers 3 and 4 is whether the authors should consider undertaking a systematic cost/benefit analysis of the proposed new methods, and whether they believe this approach is necessary across psychological science more widely (if so, why; if not, why not?). On this third point I believe it would be reasonable to reserve this for the Discussion at Stage 2, perhaps with some foreshadowing in the Stage 1 manuscript. And fourth, Reviewers 3 and 4 raise concerns about participant confidentiality and research ethics, with Reviewer 3 in particular questioning whether the proposal complies with the requirements of GDPR.

In addition to these comments, the reviewers also raise concerns about the risk of misleading readers and the public in event of a positive result (Rev 1, much in line with own views above),

whether the “transparency gap” is a particular problem for psi (is it intended as a guard against fraud?), and whether a negative finding would in fact rule out psi (see Rev 3). Despite the very thorough methodological approach, the reviewers also note a number of areas where additional detail or justification is required, for instance in considering whether the Bayesian analyses sufficiently take into account a skeptic’s prior (Rev 1) and questioning the basis of several aspects of the Bayesian analyses (Rev 2). Reviewer 1 also asks whether the steps taken to ensure computational reproducibility and public availability of the study materials are sufficient.

From an editorial perspective, I consider all of these points to be readily addressable in a revision. I therefore invite the authors to consider the reviews and revise the manuscript accordingly.

Comments to Author:

Reviewer: 1

Comments to the Author(s)

Manuscript ID: RSOS-191375

Different terms are used in the literature and the current manuscript – psi, precognition, ESP – in this review I will simply use “ESP” to refer to the phenomenon under scrutiny.

Below I have made remarks under the six headings (marked with *stars*) that the journal requested that I comment on.

The significance of the research question(s)

Without a clear definition of “significance” and what criteria I’m supposed to use to evaluate it - I’m not sure what is meant by this heading so I won’t comment.

The logic, rationale, and plausibility of the proposed hypotheses

It might not be customary for reviewers to explicitly state their own beliefs about the phenomenon under scrutiny in a review. However, I think that might be relevant in this case due to the controversial nature of the phenomenon. I have a very strong belief that ESP does not exist. There is not only an absence of substantive theory or plausible mechanism – the purported effect defies the (comparatively well-established) laws of physics and common sense (how could casinos and ESP co-exist, for example).

Nevertheless, I am a subscriber to “Cromwell’s rule” - one should not categorically rule out anything, for this makes it impossible to learn (see Lindley, 1991, *Making Decisions*). Consequently, I would be prepared to change my mind that ESP does not exist, its just that the evidence required for me to reach a favourable state of belief would be overwhelming. Practically, this means that however well designed and executed the proposed study is, a favourable outcome for the ESP hypothesis would not be sufficient to convince me that the phenomenon was real (though it would move my belief incrementally in that direction). A more likely reaction from skeptics to a positive finding will be to question what other mechanisms may have given rise to the finding. This would be an appropriate Lakatosian defense (Meehl, 1990) given that the theory being defended (established laws of physics) have considerable “money in the bank” (a strong track record of empirical support).

I expect that my skepticism towards ESP is shared by many scientists (though the only empirical support for that statement I am aware of is Wagner & Monnet, 1979, cited in Bem, 2011), but my open-mindedness might not be. Some reactions to the publication of Bem (2011) suggested that it should have been desk-rejected because of the low likelihood of the topic under scrutiny (i.e., regardless of any methodological concerns). I am aware that similar discussions about the present manuscript have taken place and the fact I am being asked to comment on the plausibility of the proposed hypothesis suggests that this is a relevant criterion in the editor’s accept/reject decision making process. Consequently, I feel it is necessary to (briefly) make the case for why this manuscript *should not be* rejected on the grounds that the phenomenon under scrutiny is highly implausible to many scientists. Here are three reasons:

(1) It is possible (though probably rare) that phenomena that are a priori implausible,

subsequently turn out to have some degree of accuracy. In the present case, this naturally seems highly unlikely, but it is possible. Thus, if ESP is real, rigorous scientific scrutiny of the kind proposed here will be necessary to establish its existence and improve our understanding of it.

(2) For implausible phenomena that are in fact not real, exposing them to robust empirical scrutiny may help to convince those who believe the phenomena are actually plausible. In the present case, I would speculate that the number of people who believe in ESP (some of whom may be in a position to substantially influence the lives of others) is not insignificant, and the actual or potential negative impact of them holding that belief (if it is false) could be quite serious. Thus, if ESP is not real, rigorous scientific scrutiny of the kind proposed here may help to convince some that their beliefs are unfounded.

(3) Finally, scientific methods do not give us direct access to the truth of the world around us. They are a set of tools that help us overcome the limitations of our own cognition and sensory apparatus. It is therefore possible that even if ESP is not real, evidence consistent with its reality is obtained using scientific methods. In such a scenario, the integrity of those methods would be called into question and efforts could be made to improve them. This was of course the dominant reaction of the scientific community to the Bem (2011) study. It also seems appropriate (see the points above about Lakatosian defence; Meehl, 1990).

There are several specific objections that I think could be raised against the study that are reasonable and should be considered. On balance, I don't feel they outweigh the arguments made above, but are worthy of consideration. The objections are that the study is potentially (1) inefficient (poor use of resources); and/or (2) dangerous (due to the negative consequences of a false positive).

(1) It may be inefficient to use resources testing an implausible hypothesis when they could be used for greater benefit elsewhere. Relevant resources include participants' time, researchers' time, funding, editors and reviewers time, financial aspects etc. It seems reasonable that decision making is relegated to the relevant parties i.e., participants can decide if it's worth their time (and are also compensated), the researchers have obviously decided it's worth their time, funders have decided it's worth their financial investment, and peer review is underway. A systematic efficiency assessment of such projects (those evaluating low plausibility hypotheses) might be useful but is beyond the scope of this specific case.

(2) The study is potentially dangerous due to the negative consequences of a false positive. Even if ESP is not real, a positive effect could be obtained because of chance statistical variation or some unanticipated experimental confound or bias. More than reasonable effort has been made to avoid this possibility, but the implications of a false positive should still be considered. If such a situation arises, I imagine that many within the scientific community will rigorously scrutinize the finding and evaluate it in its proper context (i.e., consider the low prior plausibility). But outside of scientific argumentation, the implications of such a result being published in a respected journal like *RSOS* could have serious negative repercussions. The authors have been (admirably) balanced in their write up of the article, but I do wonder whether the scenario of a positive result needs additional commentary, particularly in the abstract. Such commentary would reinforce that the finding needs to be evaluated in context i.e., that an individual with a highly skeptical prior should continue to think that the phenomenon is highly implausible, even given this positive result. It unclear if the authors would add this to their partly drafted discussion section, but there is currently little discussion of the fact that a positive finding would be contrary to well established physical laws – to such an extent that (probably for most scientists) the most likely explanation of a positive result would be a undetected methodological artefact, not ESP. The editor might also want to consider the journal's response to a positive finding – for example, accompanying the article with an editorial (as with Judd & Gawronski, 2011) and/or peer commentary (as with Wagenmakers et al. 2011) intended to provide appropriate context. Any press release would also need to be very carefully worded to convey that, despite a positive finding, ESP remains vanishingly unlikely. There is something of an

‘appeal to numbers’ and ‘appeal to authority’ about what I am suggesting, which would probably be inappropriate in purely scientific argumentation, but may be warranted when we consider that the general public might rely on such cues to evaluate the credibility of the finding.

Interestingly, testing the ESP hypothesis is actually framed as being somewhat secondary to the main goal of the paper – which is stated to be a case demonstration of several research tools that can improve transparency during study execution. This is a bit odd, because the results of the study will not tell us much about the effectiveness of the credibility tools – that is not the hypothesis under scrutiny. I expect this unusual framing arose from the collaboration of skeptics and non-skeptics. It doesn’t seem overly problematic; but it is a little confusing to follow the line of argument when its framed this way.

The authors describe how there has been much attention to improving the transparency of study planning and reporting of studies, but less focus on study execution, a situation they refer to as the ‘transparency gap’. The authors make a good case for why this study in particular could benefit from enhanced credibility though a little more elaboration on why these specific tools might address specific concerns with ESP researcher might be helpful (e.g., there seems to be a suggestion that enhanced protection against fraud is necessary, but it is really explained why).

[NB – it may be worth noting that parapsychology research has a historical precedent for driving methodological reforms. For example, it seems to have been the impetus for an early prototype of the Registered Reports model (see Wiseman et al. 2019).]

The soundness and feasibility of the methodology and analysis pipeline (including statistical power analysis where applicable)

The attention to detail in this protocol is outstanding. Feasibility has been demonstrated empirically in a substantial pilot study conducted across two labs.

The statistical analysis plan seems appropriate and very thorough. The distinction between exploratory and confirmatory analysis is well articulated. Robustness checks are employed, and an interesting hypothesis will be explored in exploratory analyses but treated separately from the primary hypothesis / analysis. Unfortunately, due to my own knowledge limitations, I’d say my scrutiny of this aspect of the protocol is fairly superficial and I’d recommend additional scrutiny by a statistical expert (particularly an expert in Bayesian statistics) if that is not already the case.

I may have misunderstood, but there didn’t seem to be a Bayesian analysis that takes into account a skeptic’s v. low prior probability (see e.g., Wagenmakers et al. 2011). I also think it could be made clearer that even if there is statistical support for M1, this cannot necessarily be directly interpreted as evidence for ESP. Here’s a comment from Jeff Rouder on Andrew Gelman’s blog to that effect: “Perhaps this is too simplistic, but I think it’s important to separate truth and Bayes factors. I believe I can learn from comparing models without any of them be true, and, in that vein, use Bayes factor as a comparison metric. If I am judicious and wise, the BF comparison of select models may point me toward the next step in understanding the phenomenon at hand.” There are of course many other explanations for findings consistent with the M1 model other than ESP. Also, as argued by Fiedler & Krueger (2013), to assume a positive effect in this paradigm is evidence of ESP is to conflate the explanandum (the event to be explained) with the explanans (the argument used for the explanation). This doesn’t seem to be addressed in sufficient detail in the current draft of the discussion (though perhaps this doesn’t need addressing until stage 2 review).

Regarding the pre-study questionnaires about belief in ESP and sensation seeking – it seems sub-optimal to take these measures before, rather than after, the core experiment. Pre-study delivery will presumably increase the risk of demand characteristics – specifically, participants working out what the study hypothesis is and changing their behavior. Admittedly, it is not clear to me exactly how their behavior would change in a way that might affect the results of the study. For

example, assuming psi is not real, even if participants know the researchers are studying psi, this should not alter their performance on the task because there is no deliberate strategy they can adopt to achieve anything other than chance performance. Nevertheless, if there is no good reason to conduct these measures before the experiment, my recommendation would be to perform them afterwards as a principle of good experimental design. The same goes for any hypothesis-blind experimenters. [NB – its actually not clear to me if the participants and the experiments are supposed to be hypothesis blind or not, perhaps this could be made more explicit? Sorry if I've missed it]

Whether the clarity and degree of methodological detail would be sufficient to replicate exactly the proposed experimental procedures and analysis pipeline

This is likely to be the most transparent research project I've ever reviewed. The methods are clearly explained and the authors have said they will share relevant materials, such as instructions to experimenters. Additionally, they will provide at least one video of the experiment being performed which will aid any replication efforts.

A few comments/suggestions:

(1) "Stimulus images used in the study can be obtained free of charge for researchers from the administrators of the NAPS and IAPS image sets." – this is not ideal from a reproducibility perspective because the long-term preservation of and access to these materials relies on a third party. If the system maintaining these image sets were no longer operating in 10 years time for instance, then it may no longer be possible to conduct a high fidelity replication of the study. I understand that there may be rights issues that prevent this, but would it be possible for the researchers to share these materials directly themselves?

(2) "These issues are discussed in the 'Additional methodological considerations' section in the Supplement." – I don't know where to obtain the supplement so I have not evaluated any part of it.

(3) The analysis code is super clear and well annotated. It has been shared on the OSF and already successfully tested on the pilot data. We can even see it operating for ourselves on the provided website. This is beyond excellent. One suggestion would be to take steps to ensure that the software environment in which the analysis is performed can also be preserved. There are several tools available that can help with this, for example Binder or Code Ocean. I've found Code Ocean quite straightforward to use.

Whether the authors provide a sufficiently clear and detailed description of the methods to prevent undisclosed flexibility in the experimental procedures or analysis pipeline

The descriptions in the paper seem very clear to me. Execution of the protocol is going to be filmed and audited by independent researchers. Assuming those procedures are robust. its hard to see how meaningful protocol deviations could occur and remain undisclosed.

The authors are eagerly constraining their own analytic flexibility by providing the exact code they will use to perform the analysis – that's about as good as it gets.

Whether the authors have considered sufficient outcome-neutral conditions (e.g. positive controls) for ensuring that the results obtained are able to test the stated hypotheses

It is not clear what such a positive control would be in the present design as there is no intervention.

Miscellaneous minor comments

"replication is a safe but costly method for verifying credibility" – I'm not sure what is meant here by "safe", could you clarify?

"The credibility enhancing methods described in this paper in combination with

previously established transparency best practices can effectively negate or reveal all known sources of researcher and publication bias.” – that is a very strong statement and it does not seem warranted. It is very difficult to measure bias and demonstrate it has been reduced or negated. We know that existing measures, such as pre-registration, are far from 100% effective (e.g., Claesen et al. <https://psyarxiv.com/d8wex>). Some of the current measures (e.g., direct data deposition) have not been tested (as far as I know).

“Adult participants with no psychiatric illness who have not participated in the study before and who are not under the influence of drugs or alcohol at the time of the session will be eligible to participate.” – perhaps state that these are inclusion criteria so they cannot be later misused as post-hoc exclusion criteria.

“After receiving a briefing about the experiment and its goals” – who provides this briefing? Relevant whether it’s a hypothesis-blind experiment or hypothesis-aware PI, for example.

“not all participants will get the same compensation to facilitate accrual” – The meaning of this sentence, and the general procedures for compensating participants, was unclear to me.

Regarding direct data deposition (DDD) and born open data – is it theoretically possible for the code to be modified to allow a form of ‘man in the middle attack’ – for example, temporarily holding the data and enabling manipulation prior to deposition. This of course seems extremely unlikely – but the whole reason for thinking DDD is necessary is the potential for fraudulent interaction with the data – thus it seems reasonable to consider just how robust this system would be against a determined bad actor.

“Any manipulation of the data is clear and can be backtracked at any time because of the audit trail kept by GitHub” – this implies that manipulation of the data would be possible, but traceable. Could the authors elaborate on how the manipulation could occur?

A potential opportunity for bias – can the experimenters observe participants’ performance? If so, this could influence their completion of the lab logs which could lead to exclusions. Experimenters should ideally not be aware of the hypothesis and unable to monitor participants performance.

Perhaps an explicit statement that the pilot data is not being combined with the main data is warranted?

“The failure to replicate/successful replication of previous positive findings with this strict methodology indicates that it is likely/unlikely that the overall positive effect in the literature might be/would be only the result of recognized methodological biases rather than ESP/biases.” – I don’t think this statement is accurate for skeptics with a low prior belief in ESP. If the study does show an effect consistent with M1, then a skeptic with a very low prior belief will still not be convinced by this single study. The possibility of some unanticipated/unrecognized methodological artefact would remain a more likely explanation (for some).

Reviewer: 2

Comments to the Author(s)

All my comment are in the attached file.

Reviewer: 3

Comments to the Author(s)

Please see the attached PDF for full comments.

Reviewer: 4

Comments to the Author(s)

Overall assessment:

Thank you for the opportunity to review the research proposal for the ‘transparent Psi project’. I was very excited to read it. It is a carefully planned proposal that hits a soft spot at the right time – the debate around ESP following Bem’s 2011 article and the subsequent replication attempts. The proposed research plan offers an impressive, extensive battery of tests and safeguards that will allow to unequivocally settle the debate. I am also a bit skeptic. My skepticism has nothing to do with the ability of the research team to execute the ambitious plan; on the contrary, I am convinced that the plan is adequate and the research team has the capacity to execute it. Rather, it is about the cost-benefit of the proposed safeguards and feasibility of implementation in future studies. I wonder if the time, effort, human labour, and dollar costs of such a study can be invested regularly in scientific investigations. My guess is that this type of concerted effort might be limited to a (very) small number of critical studies that attempt to resolve an ongoing dispute. Overall, this looks like a terrific project and I strongly recommend it goes ahead. I would like to see practical, smaller scale and cost-effective solutions to the credibility problem.

Significance:

Very high, given both the scientific and the public interest in ESP, and the ongoing controversy surrounding Bem’s original study (2011) and subsequent replication or analysis attempts (eg Wagenmakers and colleagues, 2011, 2015).

Logic and rationale:

The hypothesis are broadly based on the original study, with M1 and M0 indicating support for better-than-random guessing, vs random. As such, they are adequate.

Methodology:

The proposed methods are sound and clearly explained. The proposal outlines a number of key features that serve to ensure the credibility of the findings. These include deposition of protocols and testing logs, born open data, other bias reduction, and increased transparency via third party observers. Each ‘tool’ has its own merit, and together they form an almost formidable wall to protect the integrity of the study (notwithstanding my concerns about the costs required for the implementation).

Other considerations:

- The proposal highlights the advantages of real time report, with which I wholeheartedly agree. In a nut shell, data is immediately deposited to a publicly open repository and the results of the study will be analysed as the data is accumulated (not at the end of data collection). This offers greater transparency. A potential risk of the real time report though is that spurious effects could look significant at some arbitrary point in time and members of the public or media might this snapshot in time to draw conclusions. Are there appropriate mechanisms to minimise that kind of risk?
- A potential risk of the born-open data is that sharing data in real time that compromise that anonymity of the participants. The research team had taken a number of measures (encrypted timestamps, group release) to minimise this risk and I have found the overall strategy satisfactory.
- The research team had offered a number of exit points for data collection, based on predetermined numbers of trials and subjects. Perhaps they would like to consider Bayesian Optional Stopping (e.g., Rouder, 2014, PB&R) or if not explain why it is not adequate.

Author's Response to Decision Letter for (RSOS-191375.R0)

See Appendix C.

RSOS-191375.R1 (Revision)

Review form: Reviewer 1

Do you have any ethical concerns with this paper?

No

Recommendation?

Accept in principle

Comments to the Author(s)

I am satisfied that my comments on the previous version of the manuscript have been adequately addressed and I wish the authors all the best with the execution of their project.

Review form: Reviewer 2 (Brice Beffara-Bret)

Do you have any ethical concerns with this paper?

No

Recommendation?

Accept in principle

Comments to the Author(s)

Comments are in the attached file (Appendix D).

Review form: Reviewer 4

Do you have any ethical concerns with this paper?

No

Recommendation?

Accept in principle

Comments to the Author(s)

I reviewed the original submission. I will therefore not reiterate each point, but rather address the key concerns I had, and whether they were attended properly, in my views

I had two major concerns, and a few more minor points. The first major concern was the substantial investment required for the study, and hence the question of cost-benefit. The second major concern was about the risk of misinterpretation of the results by the public, especially because the data and results will be publicly open as the data is collected. Thus, the public, media, or both, may form partial or even false opinion that is based on this 'snapshot in time', without awaiting the full gamut of results at the conclusion of data collection.

I find the response of the authors to both issues satisfactory. For the former, they intend to include cost-benefit analysis in the Discussion section at Stage 2. This was now added to the aims of the project. For the latter, the authors propose to include clear warnings about the nature of the data, and warn users against use prior to the pre-specified stopping points.

I had the privilege of also reading the feedback of the Editor and other Reviewers, as well as the authors' response. I will leave it to the Editor and other Reviewers to assess those points, but my cursory observation suggests the authors seem to have been thinking carefully on ways to mitigate potential risks and pitfalls.

Decision letter (RSOS-191375.R1)

07-Jan-2020

Dear Dr Kekecs

On behalf of the Editor, I am pleased to inform you that your Manuscript RSOS-191375.R1 entitled "Raising the value of research studies in psychological science by increasing the credibility of research reports: The Transparent Psi Project" has been accepted in principle for publication in Royal Society Open Science. The reviewers' and editors' comments are included at the end of this email.

You may now progress to Stage 2 and complete the study as approved. Before commencing data collection we ask that you:

- 1) Update the journal office as to the anticipated completion date of your study.
- 2) Register your approved protocol on the Open Science Framework (<https://osf.io/>) or other recognised repository, either publicly or privately under embargo until submission of the Stage 2 manuscript. Please note that a time-stamped, independent registration of the protocol is mandatory under journal policy, and manuscripts that do not conform to this requirement cannot be considered at Stage 2. The protocol should be registered unchanged from its current approved state, with the time-stamp preceding implementation of the approved study design.

Following completion of your study, we invite you to resubmit your paper for peer review as a Stage 2 Registered Report. Please note that your manuscript can still be rejected for publication at Stage 2 if the Editors consider any of the following conditions to be met:

- The results were unable to test the authors' proposed hypotheses by failing to meet the approved outcome-neutral criteria.
- The authors altered the Introduction, rationale, or hypotheses, as approved in the Stage 1 submission.
- The authors failed to adhere closely to the registered experimental procedures. Please note that any deviations from the approved experimental procedures must be communicated to the editor immediately for approval, and prior to the completion of data collection. Failure to do so can result in revocation of in-principle acceptance and rejection at Stage 2 (see complete guidelines for further information).
- Any post-hoc (unregistered) analyses were either unjustified, insufficiently caveated, or overly dominant in shaping the authors' conclusions.
- The authors' conclusions were not justified given the data obtained.

We encourage you to read the complete guidelines for authors concerning Stage 2 submissions at <https://royalsocietypublishing.org/rsos/registered-reports#ReviewerGuideRegRep>. Please especially note the requirements for data sharing, reporting the URL of the independently registered protocol, and that withdrawing your manuscript will result in publication of a Withdrawn Registration.

Please note that Royal Society Open Science will introduce article processing charges for all new

submissions received from 1 January 2018. Registered Reports submitted and accepted after this date will ONLY be subject to a charge if they subsequently progress to and are accepted as Stage 2 Registered Reports. If your manuscript is submitted and accepted for publication after 1 January 2018 (i.e. as a full Stage 2 Registered Report), you will be asked to pay the article processing charge, unless you request a waiver and this is approved by Royal Society Publishing. You can find out more about the charges at <https://royalsocietypublishing.org/rsos/charges>. Should you have any queries, please contact openscience@royalsociety.org.

Once again, thank you for submitting your manuscript to Royal Society Open Science and we look forward to receiving your Stage 2 submission. If you have any questions at all, please do not hesitate to get in touch. We look forward to hearing from you shortly with the anticipated submission date for your Stage 2 manuscript.

Kind regards,
Lianne Parkhouse
Royal Society Open Science
openscience@royalsociety.org

on behalf of Professor Chris Chambers (Registered Reports Editor, Royal Society Open Science)
openscience@royalsociety.org

Associate Editor Comments to Author (Professor Chris Chambers):
Three of the four reviewers assessed the revised manuscript and all have responded positively (see reviews below).

Reviewers' comments to Author:

Reviewer: 1

Comments to the Author(s)

I am satisfied that my comments on the previous version of the manuscript have been adequately addressed and I wish the authors all the best with the execution of their project.

Reviewer: 2

Comments to the Author(s)

Comments are in the attached file.

Reviewer: 4

Comments to the Author(s)

I reviewed the original submission. I will therefore not reiterate each point, but rather address the key concerns I had, and whether they were attended properly, in my views

I had two major concerns, and a few more minor points. The first major concern was the substantial investment required for the study, and hence the question of cost-benefit. The second major concern was about the risk of misinterpretation of the results by the public, especially because the data and results will be publicly open as the data is collected. Thus, the public, media, or both, may form partial or even false opinion that is based on this 'snapshot in time', without awaiting the full gamut of results at the conclusion of data collection.

I find the response of the authors to both issues satisfactory. For the former, they intend to include cost-benefit analysis in the Discussion section at Stage 2. This was now added to the aims of the project. For the latter, the authors propose to include clear warnings about the nature of the data, and warn users against use prior to the pre-specified stopping points.

I had the privilege of also reading the feedback of the Editor and other Reviewers, as well as the authors' response. I will leave it to the Editor and other Reviewers to assess those points, but my cursory observation suggests the authors seem to have been thinking carefully on ways to mitigate potential risks and pitfalls.

Author's Response to Decision Letter for (RSOS-191375.R1)

See Appendix E.

RSOS-191375.R2 (Revision)

Review form: Reviewer 1

Is the manuscript scientifically sound in its present form?

Yes

Are the interpretations and conclusions justified by the results?

Yes

Is the language acceptable?

Yes

Do you have any ethical concerns with this paper?

No

Have you any concerns about statistical analyses in this paper?

No

Recommendation?

Accept with minor revision

Comments to the Author(s)

I appreciate the opportunity to provide a Stage 2 review for this Registered Report. I've read compared the Stage 1 and Stage 2 manuscripts and examined the changes/additions in detail. The discussion is informative, concise, and balanced. I have no concerns with the conduct, reporting, or interpretation of the study, and my review is therefore brief.

I have one minor comment/question:

Perhaps I missed the explanation, but in Table 1, why are some details (e.g., the institution or PI name) about participating laboratories not disclosed? Perhaps the reason could be provided in the table caption.

Below I have responded directly to the journal's questions for reviewers:

1. Whether the data are able to test the authors' proposed hypotheses by passing the approved outcome-neutral criteria (such as absence of floor and ceiling effects or success of positive controls or other quality checks). Failure to pass these conditions may lead to manuscript rejection.

Yes.

2. Whether the Introduction, rationale and stated hypotheses are the same as the approved Stage1 submission (required).

Yes.

3. Whether the authors adhered precisely to the registered experimental procedures.

Essentially yes. The authors report a few minor deviations from the registered protocol, which all seem completely appropriate to me.

4. Where applicable, whether any unregistered exploratory statistical analyses are justified, methodologically sound, and informative.

Yes.

5. Whether the authors' conclusions are justified given the data. Please note that editorial decisions will not be based on the perceived importance, novelty, or conclusiveness of the results.

Yes.

As a final word: this collaboration between researchers from both sides of a scientific debate, working together through a consensus design process is highly commendable. Moreover, I believe this is the most rigorous and transparent study I've ever read. For a group of researchers to voluntarily opt for this level of protection against their own biases is inspiring. A wonderful demonstration that the scientific spirit is alive and well. I congratulate the authors on their achievement and look forward to seeing this work published.

Review form: Reviewer 2 (Brice Beffara)

Is the manuscript scientifically sound in its present form?

Yes

Are the interpretations and conclusions justified by the results?

Yes

Is the language acceptable?

Yes

Do you have any ethical concerns with this paper?

No

Have you any concerns about statistical analyses in this paper?

No

Recommendation?

Accept with minor revision

Comments to the Author(s)

My comments are in the attached file (see Appendix F).

Review form: Reviewer 3

Is the manuscript scientifically sound in its present form?

Yes

Are the interpretations and conclusions justified by the results?

Yes

Is the language acceptable?

Yes

Do you have any ethical concerns with this paper?

No

Have you any concerns about statistical analyses in this paper?

No

Recommendation?

Accept with minor revision

Comments to the Author(s)

This paper is absolutely beautiful. It is the most comprehensive demonstration of 'open' and robust research practices yet, and deserves its place in the literature. The authors show 'how it is done' in a very convincing way. I applaud the authors for not only taking into account methodological rigor, but also research ethics and privacy. All in all, although one could (rightfully) argue that some of the measures the authors have taken are a bit too much of a good thing, the present project represents IMO the new gold standard for replications of high stake studies.

It also is the final nail in the coffin of the Bem paradigm. As such, this paper marks a very significant point in experimental parapsychology - after 11 years of discussion on the Bem findings, I would argue that even the most staunch believer will have to conclude that the alleged effect does not exist as a mechanistically replicable phenomenon. The importance of this cannot be overestimated - as the authors state, the paper does not disprove the psi hypothesis, but does provide very strong evidence against the psi-as-signal hypothesis.

The only minor point I would make is that the observed result, namely a null-effect for psi in a fully open setting, is completely in line with an alternative for the psi-as-signal hypothesis, namely the 'psi-as-correlation' hypothesis. In particular Walter Von Lucadou has argued that psi effects 'disappear' if one attempts to make direct measurements (or use psi to send signals from the future to the present, as in this experiment), and may only be manifested in anomalous correlation patterns within the data. Although I understand the skeptical tone in the present paper given the extremely convincing null result, it would be at least fair to address and acknowledge that even in the psi community, the ESP hypothesis is not uncontroversial.

Review form: Reviewer 4**Is the manuscript scientifically sound in its present form?**

Yes

Are the interpretations and conclusions justified by the results?

Yes

Is the language acceptable?

Yes

Do you have any ethical concerns with this paper?

No

Have you any concerns about statistical analyses in this paper?

No

Recommendation?

Accept with minor revision

Comments to the Author(s)

See attached pdf (Appendix G).

Decision letter (RSOS-191375.R2)

Dear Dr Kekecs:

On behalf of the Editor, I am pleased to inform you that your Stage 2 Registered Report RSOS-191375.R2 entitled "Raising the value of research studies in psychological science by increasing the credibility of research reports: The Transparent Psi Project." has been deemed suitable for publication in Royal Society Open Science subject to minor revision in accordance with the referee suggestions. Please find the referees' comments at the end of this email.

The reviewers and Subject Editor have recommended publication, but also suggest some minor revisions to your manuscript. We invite you to respond to the comments and revise your manuscript. Below the referees' and Editors' comments (where applicable) we provide additional requirements. Final acceptance of your manuscript is dependent on these requirements being met. We provide guidance below to help you prepare your revision.

Please submit your revised manuscript and required files (see below) no later than 14 days from today's (ie 05-Dec-2022) date. Note: the ScholarOne system will 'lock' if submission of the revision is attempted 14 or more days after the deadline. If you do not think you will be able to meet this deadline please contact the editorial office immediately.

on behalf of Professor Chris Chambers
(Registered Reports Editor, Royal Society Open Science)
openscience@royalsociety.org

Associate Editor Comments to Author (Professor Chris Chambers):

The four reviewers from Stage 1 kindly returned to evaluate your Stage 2 manuscript, and I have also read it carefully myself. As I expected based on my reading, the reviews are very positive -- perhaps the most consistently glowing I have seen in the last ten years. As you know (and as I laid out in my very first action letter over 3 years ago), I approached your submission with some hesistance as editor given the topic area and my own biases about the existence of ESP. However, after consultation, I decided at the time that my initial instincts to desk reject your submission were wrong, and at the point of Stage 1 IPA I was convinced that I had made the correct decision to encourage you to proceed. Now seeing the completed Stage 2 manuscript, I am even more convinced that this is quite a special piece of work. It is difficult to estimate the impact of any one contribution to the science, but I believe your article may kickstart a transformation in the way RRs are used to tackle the most important and/or controversial questions in any field. For this, I want to congratulate the entire team for this remarkable accomplishment.

The reviews are very positive, but there are nevertheless some useful nuggets to consider in Stage 2 revision. One in particular that resonated with me was the comment by Reviewer 4 to beef up the cost/benefit analysis in the Discussion. Most of the other suggestions are minor. In revising, please avoid any further changes to the parts of the manuscript unless doing so is necessary to correct an error of fact or typo/grammatical error (Nb. this instruction overrules all reviewer requests that might suggest doing otherwise).

I anticipate being able to accept your revised Stage 2 manuscript without in-depth further review.

Comments to Author:

Reviewer: 1

Comments to the Author(s)

I appreciate the opportunity to provide a Stage 2 review for this Registered Report. I've read compared the Stage 1 and Stage 2 manuscripts and examined the changes/additions in detail. The discussion is informative, concise, and balanced. I have no concerns with the conduct, reporting, or interpretation of the study, and my review is therefore brief.

I have one minor comment/question:

Perhaps I missed the explanation, but in Table 1, why are some details (e.g., the institution or PI name) about participating laboratories not disclosed? Perhaps the reason could be provided in the table caption.

Below I have responded directly to the journal's questions for reviewers:

1. Whether the data are able to test the authors' proposed hypotheses by passing the approved outcome-neutral criteria (such as absence of floor and ceiling effects or success of positive controls or other quality checks). Failure to pass these conditions may lead to manuscript rejection.

Yes.

2. Whether the Introduction, rationale and stated hypotheses are the same as the approved Stage 1 submission (required).

Yes.

3. Whether the authors adhered precisely to the registered experimental procedures.

Essentially yes. The authors report a few minor deviations from the registered protocol, which all seem completely appropriate to me.

4. Where applicable, whether any unregistered exploratory statistical analyses are justified, methodologically sound, and informative.

Yes.

5. Whether the authors' conclusions are justified given the data. Please note that editorial decisions will not be based on the perceived importance, novelty, or conclusiveness of the results.

Yes.

As a final word: this collaboration between researchers from both sides of a scientific debate, working together through a consensus design process is highly commendable. Moreover, I believe this is the most rigorous and transparent study I've ever read. For a group of researchers to voluntarily opt for this level of protection against their own biases is inspiring. A wonderful demonstration that the scientific spirit is alive and well. I congratulate the authors on their achievement and look forward to seeing this work published.

Reviewer: 3

Comments to the Author(s)

This paper is absolutely beautiful. It is the most comprehensive demonstration of 'open' and robust research practices yet, and deserves its place in the literature. The authors show 'how it is done' in a very convincing way. I applaud the authors for not only taking into account methodological rigor, but also research ethics and privacy. All in all, although one could (rightfully) argue that some of the measures the authors have taken are a bit too much of a good

thing, the present project represents IMO the new gold standard for replications of high stake studies.

It also is the final nail in the coffin of the Bem paradigm. As such, this paper marks a very significant point in experimental parapsychology - after 11 years of discussion on the Bem findings, I would argue that even the most staunch believer will have to conclude that the alleged effect does not exist as a mechanistically replicable phenomenon. The importance of this cannot be overestimated - as the authors state, the paper does not disprove the psi hypothesis, but does provide very strong evidence against the psi-as-signal hypothesis.

The only minor point I would make is that the observed result, namely a null-effect for psi in a fully open setting, is completely in line with an alternative for the psi-as-signal hypothesis, namely the 'psi-as-correlation' hypothesis. In particular Walter Von Lucadou has argued that psi effects 'disappear' if one attempts to make direct measurements (or use psi to send signals from the future to the present, as in this experiment), and may only be manifested in anomalous correlation patterns within the data. Although I understand the skeptical tone in the present paper given the extremely convincing null result, it would be at least fair to address and acknowledge that even in the psi community, the ESP hypothesis is not uncontroversial.

Reviewer: 2

Comments to the Author(s)

My comments are in the attached file (review and comment on Kekecs et al. stage 2 RR .pdf)

Reviewer: 4

Comments to the Author(s)

see attached pdf (review Royal Society_RSOS-191375.R2.pdf)

===PREPARING YOUR MANUSCRIPT===

one version should clearly identify all the changes that have been made (for instance, in coloured highlight, in bold text, or tracked changes);

If you have been asked to revise the written English in your submission as a condition of publication, you must do so, and you are expected to provide evidence that you have received language editing support. The journal would prefer that you use a professional language editing service and provide a certificate of editing, but a signed letter from a colleague who is a proficient user of English is acceptable. Note the journal has arranged a number of discounts for authors

using professional language editing services
(<https://royalsociety.org/journals/authors/benefits/language-editing/>).

===PREPARING YOUR REVISION IN SCHOLARONE===

-- If you are requesting an article processing charge waiver, you must select the relevant waiver option (if requesting a discretionary waiver, the form should have been uploaded, see 'File upload' above).

-- If you have uploaded any electronic supplementary (ESM) files, please ensure you follow the guidance at <https://royalsociety.org/journals/authors/author-guidelines/#supplementary-material> to include a suitable title and informative caption. An example of appropriate titling and captioning may be found at https://figshare.com/articles/Table_S2_from_Is_there_a_trade-off_between_peak_performance_and_performance_breadth_across_temperatures_for_aerobic_scoope_in_teleost_fishes_/3843624.

Author's Response to Decision Letter for (RSOS-191375.R2)

See Appendix H.

Decision letter (RSOS-191375.R3)

Dear Dr Kekecs:

I am pleased to inform you that your manuscript entitled "Raising the value of research studies in psychological science by increasing the credibility of research reports: The Transparent Psi Project." is now accepted for publication in Royal Society Open Science.

Please remember to make any data sets or code libraries 'live' prior to publication, and update any links as needed when you receive a proof to check - for instance, from a private 'for review' URL to a publicly accessible 'for publication' URL. It is also good practice to add data sets, code and other digital materials to your reference list.

If it is not already available in your OSF deposition, it is also recommended to make the code/software in your Gitlab repository accessible via a repository that provides depositions with DOIs.

Royal Society Open Science is a fully open access journal. A payment may be due before your article is published. Our partner Copyright Clearance Center's RightsLink for Scientific Communications will contact the corresponding author about your open access options from the email domain @copyright.com (if you have any queries regarding fees, please see <https://royalsocietypublishing.org/rsos/charges> or contact authorfees@royalsociety.org).

on behalf of Professor Chris Chambers (Subject Editor).

Follow Royal Society Publishing on Twitter: @RSocPublishing
Follow Royal Society Publishing on Facebook:
<https://www.facebook.com/RoyalSocietyPublishing/>
Read Royal Society Publishing's blog:
<https://royalsociety.org/blog/blogsearchpage/?category=Publishing>

Appendix A

This is a review and comment on Kekecs et al. stage 1 RR entitled “Raising the value of research studies in psychological science by increasing the credibility of research reports: The Transparent Psi Project.”

As I understand the project, two main aims motivate the research team :

1- Showing a concrete way to carry out a highly rigorous protocol based on the recent progress in methodology.

2- Using this protocol to test a controversial hypothesis in psychology

The significance of the research question(s):

This project is of high importance for the field. It is crucial to provide examples on how to manage a scientific project with all the tools supposed to increase transparency and scientific quality. I also appreciate the idea of hypothesis testing applied to Bem’s work because Bem’s findings triggered a lot of questions in our field but also in the general population concerning the rigor of psychology and science in general. Hence, I can say that this work really deserves to be carried out.

The logic, rationale, and plausibility of the proposed hypotheses

The two main aims are clearly linked and this is a great idea to use a methodological interest to deal with the test of a (highly) controversial hypothesis. The plausibility of the hypothesis is one of the questions of the project because of the “meta-science” aspect of this work. All in all, the introduction is very clear and perfectly shows the relevance of the work to be done.

The soundness and feasibility of the methodology and analysis pipeline (including statistical power analysis where applicable):

There is no doubt for me that the project is feasible and that the methodology is adapted to the aim of the authors. The research team did everything to ensure the quality of the protocol and the scientific rigor proposed here is impressive.

Whether the clarity and degree of methodological detail would be sufficient to replicate exactly the proposed experimental procedures and analysis pipeline:

There is no doubt that this procedure could be replicated. The precision of information in the manuscript and online is far above the classical reports in the field. Often, the authors even propose several ways to ensure the precision of their descriptions (e.g. videos of experimenters giving the instructions).

Whether the authors provide a sufficiently clear and detailed description of the methods to prevent undisclosed flexibility in the experimental procedures or analysis pipeline:

Degrees of freedom are very limited because of the clear details provided in the methods and analysis section. Moreover, This project benefits from a committee composed of “pro ESP” and “against ESP” which limits flexibility even more drastically.

Whether the authors have considered sufficient outcome-neutral conditions (e.g. positive controls) for ensuring that the results obtained are able to test the stated hypotheses:

The authors use the original protocol for the replication. The protocol is well suited to test the hypothesis. There is a “control condition” with neutral images but this condition will not be used in data analysis. I understand that the authors want to keep this condition in order to match the original protocol. However I would like to know more about their motivation to keep this condition out of the analysis. Indeed, we could also consider the comparison between the “erotic” and the “neutral” condition as relevant. The “neutral condition” could serve as a choice “baseline” representing the “no effect condition”.

I also have some others comments :

- 1) I do not understand why primary analysis are performed with Bayes Factors and then Bayesian parameter estimation is used for Robustness check. The procedures proposed by Kruschke (2018) could be used in the primary analysis for both parameter estimation and hypothesis testing.
- 2) I hope I did not miss anything in the manuscript but why 51%? What is the logical justification of this threshold?
- 3) Why $BF = 25$ for decision making? What is more it is likely that more observations will be needed to reach $BF = 0.04$ than 25 (e.g. see Schönbrodt, Wagenmakers, Zehetleitner, & Perugini, 2017). Is the protocol “fair” and evenly designed for both hypotheses ?
- 4) Why do you choose to report the 90% HDI and then to see if 95% of the posterior falls inside or outside the rope ?
- 5) Why do you choose not to blind experimenters? You take a lot of precautions in your protocol. I agree that the influence of experimenters may not be the largest source of bias but still, don't you think that blinding the experimenters could improve your protocol ?

Overall I find this project of high in interest and I want to underscore the meticulous job done by the research team. I look forward to their response to my comments.

Minor comments :

Page 17, Line 19: I think that there could be a punctuation problem in the following sentence “*We will build an intercept only mixed logistic regression model (using the glmer function in the*

lme4 package in R, Bates et al., 2019). The lme4 package. R package version, 2(1), 74.) allowing for a random intercept for participants, to predict the outcome (success or failure) of the guess in experimental trials.”

Page 21, Line 46: missing parenthesis: “(Bem et al., 2016; Bierman, Spottiswoode, & Bijl, 2016; Rouder & Morey, 2011”

References

Kruschke, J. K. (2018). Rejecting or Accepting Parameter Values in Bayesian Estimation. *Advances in Methods and Practices in Psychological Science*, 1(2), 270–280. <https://doi.org/10.1177/2515245918771304>

Schönbrodt, F. D., Wagenmakers, E.-J., Zehetleitner, M., & Perugini, M. (2017). Sequential hypothesis testing with Bayes factors: Efficiently testing mean differences. *Psychological Methods*, 22(2), 322–339. <https://doi.org/10.1037/met0000061>

Appendix B

Review of *“Raising the value of research studies in psychological science by increasing the credibility of research reports: The Transparent Psi Project.”*

Declaration of conflicts of interest: *I have been a participant in the first round of the Delphi process as laid down in the paper, but have withdrawn from the project because of a lack of time.*

The authors present a replication effort of the notorious studies by Bem (2011), in which it was claimed that research participants can ‘feel the future’. Bem’s original work was widely criticized because sceptics believed the effects to be impossible, the statistical analyses to be unsound, and, perhaps most importantly, because several sceptics failed to replicate the original results. Bem’s paper is seen as one of the main catalysts in the so-called replication crisis in psychology. Since 2011, many psychologists have called for reforms in the way psychological science is carried out. The authors of the present paper have implemented several of these methods (and then some more) in order to test the hypothesis that positive results in the parapsychological literature are due to bad research practices and methodological issues.

The authors of the present paper have employed an impressive range of safeguards to rule out any alternative explanations that have to do with the practical aspects of research, research design, and data handling should the project result in a positive finding, i.e. a finding that would be supportive of ESP.

Given the massive scope of the project, there are several levels at which this paper needs to be evaluated. I have structured my review according to the levels I personally find most relevant.

In short, I believe this project to have enormous potential - to my knowledge, it is the most comprehensive implementation of transparency and openness practices in psychological science to date. As such, regardless of theoretical importance, about which one may argue, this project deserves a place in the literature.

Relevance to field of parapsychology

The present paper aims to make a theoretical contribution to the field of parapsychology by testing the hypothesis that positive results in the literature are due to bad research practices. Although I believe the project can make a theoretical contribution to parapsychology, I do not believe the project can give an answer to the broad main question asked here, namely whether positive findings in the parapsychological literature are due to bad research practices.

During the Delphi process, only one single experiment has been considered, and an experiment that has several problematic issues with regard to methodology. The number of trials per participant is rather small, to name just one example. Moreover, in several replication studies, including a large project presented at this year’s convention of the Parapsychological Association, it has been shown that the sought-after effect is very likely not to replicate - regardless of the meta-analysis presented by the authors.

Obviously, the most likely reason sceptics will present for this is that these effects cannot exist, but even within parapsychology there are several theories that claim that ESP-effects disappear the moment they can reliably be used as a signal, or when 'organizational closure' (whatever that may mean) of a system is insufficient. The 'born open' nature of the project would directly conflict with the idea that anomalous phenomena only arise within rather isolated systems, and are only expressed as 'excess correlations' that are sometimes appear to violate the normal laws of physics (cf. Von Lucadou e.a.) If one takes this position, one would even explicitly predict a negative finding for this project. Although this idea may seem very alien and weird to mainstream psychologists, it is getting traction within the parapsychological community - not so much because it provides a convenient escape from explaining why ESP-effects are so hard to replicate, but because there is increasing empirical support using novel experimental paradigms and analysis techniques for this idea.

In sum, the issue here is that only a positive finding would be a theoretically meaningful result for the field of parapsychology. It would lend credibility to the psi hypothesis, and as an additional bonus, would make short work of the above mentioned 'model of pragmatic information'. However, in the much more likely event of a negative finding, there are two explanations - the mainstream explanation (psi does not exist), but also the explanation that psi cannot be measured in this particular manner, whether it is because of the paradigm, or because of the open nature of the project. In either case, the authors cannot make the claim that this project is able to answer their main research question, namely whether positive findings in the parapsychological literature are due to bad research practices. The project should at least be expanded with the above mentioned novel methods (they're called correlation matrix experiments - I'm quite sure the authors are familiar with such experiments) in order to increase the theoretical value of the study. Alternatively, at the very least the authors might want to consider to ask each involved experimenter and lab lead to (privately) note their expectations for their own contributions and for the project as a whole, to learn more about experimenter effects. In particular, when combined with a parallel replication without the all the stringent measures named in the paper (see below), this would make a very interesting and compelling theoretical contribution.

Relevance to methodology

Within this project, many of the innovations and reforms that have been proposed in the literature to improve the quality and reliability of the research process come together. I can only applaud the enormous effort the authors have made, and am deeply impressed with the result. In particular the shiny apps are absolutely fantastic and an absolute breakthrough in how to communicate science to a broader audience. This alone would warrant a publication on science communication.

That said, the experiences of the authors with this process are to me the most interesting part of this study - I feel these should be the main focus of the paper. A structured - quantitative and qualitative - evaluation of the entire process, with all relevant parties is indispensable. A very

interesting and perhaps even necessary addition to such an evaluation would be a parallel project in which the transparency measures such as born open data are not implemented, save the independent audit at the end. This would allow for a direct empirical test of the effect and effectiveness of using open practices for increasing data and process quality. Suppose that the audit indeed finds deviations from protocol and/or irregularities in the data which would alter the conclusions of the parallel project, this would lend strong support to the idea that open practices are a necessity for reliable science.

This is a very important point, as many of the perceived benefits of increased transparency in science are based on assumptions. Although these assumptions are very plausible, I would strongly welcome a proper empirical evaluation, in particular when combined with an assessment of experimenter expectations and belief (see above).

In the light of this previous point, another important part of the evaluation should be a detailed cost-benefit analysis of all the measures the authors have implemented. It is vitally important to realize that setting up a Delphi process, implementing parallel replications and a 'born open' data infrastructure, and an independent data and procedure audit are not only very time consuming, but also very expensive, and potentially out of reach for many researchers, let alone that many paradigms may not be implemented in such a way. Finally, a lot of flexibility is lost as a result of incorporating a Delphi process or requiring parallel multilab replications. Of course this is a trade-off: the situation pre-replication crisis was one of too much flexibility for researchers. It is a very interesting and highly relevant question where to put the line between flexibility and accountability. Overall, I must admit that on a personal level I feel the measures implemented in this project to be 'too much of a good thing', and from my personal experience as Director of Research and Data Infrastructure, I believe many researchers feel the same way. I would be very interested in hearing the authors' opinion on the matter whether the stringent measures in this project are *truly* necessary to guarantee scientific integrity or that they see the present project primarily as a proof of concept of several transparency reforms suggested by the open science community. This is not a fully empirical matter, obviously, but the authors of this paper and their co-workers are in the unique position to give a first-hand insight into how operating within very strict conditions works out. If not in this paper, I would strongly encourage the authors to give an account of their experiences beyond what we typically read in research papers.

As a side note, I would strongly recommend the editor of this journal to consider a call for opinion papers on this project, in particular to spark a debate within the community about the necessity for reforms towards openness and accountability, and how far such reforms should go.

In sum, I feel there is a great potential in this paper: regardless of the theoretical contribution, of which I have some reservations with regard to its value, these authors are pioneers in working within the most stringent conditions of transparency. A document of their experiences is extremely valuable for the scientific community.

Legal, and ethical issues

With regard to the measures implemented in the project, I do have several concerns about the ethical and legal aspects, in particular in the light of the European privacy legislation (GDPR). Given that under most ethics codes, compliance with legal regulations is required I have grouped ethics and legal together here. With regard to the ethics of the experimental procedure, I have absolutely no problems. However, the paper does not give sufficient information to evaluate if all legal regulations with regard to data protection have been met. I would like to stress here that this is not the authors' fault - there are many inclarities regarding the GDPR, and in particular how the GDPR affects open science. However, given the pioneering role of the authors I am afraid this point needs to be brought up.

The goal of the GDPR is to give citizens (data subjects) more control over how their personal data is used. The GDPR has derogations for research, but researchers should be precise and very careful in documenting how they protect the legal rights of their subjects.

Personal data has a broad definition under the GDPR - any data that may be traced back to an individual is personal data, even if a person is not directly identifiable. The authors have clearly recognized this point - for example, they even take care that participants are not identifiable by means of a time stamp. Of course, if it is fundamentally impossible to re-identify participants, data is anonymous and the GDPR does not apply. However, in order to claim that data is truly anonymous, one should run an analysis on parameters such as dispersion and separability of variables and records, which will be very difficult within the born-open model, as the dataset continuously changes. From a legal perspective, the born-open dataset as presented here will by definition be a pseudonymized dataset.

In order to operate transparently within the GDPR, pseudonymization of persona data prior to publicly sharing is not sufficient. In particular, explicit informed consent is required in which it is very clearly explained that data will be made public, what the risks are to the data subject (I can imagine these are very limited in this particular case), and what measures have been taken to protect the data subject's identity, how the data will be used, and that the data subject voids her/his rights on data removal, data correction, etc. The data subject needs to explicitly agree on these points. It would be very helpful if the authors could share their informed consent forms so other researchers who want to use the 'born open' model could use these to write their own.

Another legal issue arises with the external audit. The auditing party will need access to personal data, not just of research participants, but also of the researchers. Have the authors documented this process, and how have they arranged the necessary processing agreements and obtained informed consent?

In sum, the most important question I have for the authors at this point is whether a data protection impact assessment (DPIA) has been carried out, as required by the GDPR for a project as this (following the guidelines from the Article 29 Working Party on data protection

impact assessment, see http://ec.europa.eu/newsroom/document.cfm?doc_id=47711) During a DPIA, data protection issues are systematically charted and evaluated from a multi-stakeholder perspective. The resulting document can be used to demonstrate compliance with the GDPR. It is critically important to realize that a DPIA does not result in a 'yes' or 'no' with regard to GDPR compliance, but rather to find the optimal way to carry out a research project in such a way that data subjects' fundamental rights to data autonomy are safeguarded, and where this is not possible, this is clearly communicated to the data subjects.

If a DPIA has been carried out, this document would be extremely valuable for the open science community, and warrant a publication on its own, as other researchers who want to implement practices can refer to this document. If no DPIA has been carried out, can the authors specify why a DPIA has not been deemed necessary, and whether this was decided by an ethics committee, or a data protection officer of the host institute?

Appendix C

Response to comments

The effort put into reviewing this Stage 1 submission is nothing short of extraordinary. We are extremely grateful for the hard work of the editor and the reviewers on reviewing our manuscript and for the thoughtful comments. Below we list the comments and our responses and actions taken in response to each.

Responses to the associate editor:

Comment #1

Four expert reviewers have now assessed the Stage 1 manuscript. Before summarising the reviews and the key points raised, I want to note for the record my own reasoning in handling this manuscript and deciding to send it for review. I do this especially in light of the fact that, if eventually accepted and published at Stage 2 with results, the open review policy of RSOS will lead to the publication of the reviews and my editorial action letters, as well as the potential publication of an editorial and/or commentaries. Given the controversial subject matter, the content of this discussion is therefore likely to fall, quite rightly, under public scrutiny.

I want to begin by making clear that upon receiving a presubmission enquiry from the authors about this potential RR, my instinct was to decline a full submission. This stemmed from my belief that the phenomenon under investigation in this work (psi) is as close to impossible as anything that might be conceived in science, contradicting established physical laws (which I believe must constrain any evidence that could be obtained in psychological science), and therefore belonging firmly in the realm of pseudoscience. Moreover, it is my firm belief that IF any positive evidence for psi were to be observed in such a study, however rigorous, it (a) would almost certainly have arisen due to some form of experimental error, known or unknown, and (b) could risk significantly misleading the public about the likely existence of psi -- a point, as we will see, that was also raised by Reviewer 1.

However, I am also mindful that the role of a journal editor – and indeed a scientist more generally – is to remain skeptical not only about the truth/existence of unlikely theories or phenomena, but to remain skeptical about our own skepticism and not allow our biases to limit the advancement of science. I therefore consulted with editorial colleagues at the journal, and publicly (see <https://twitter.com/chrisdc77/status/1156946007839690754>) to determine whether my thoughts on this matter were well aligned with those of the scientific community.

As it happens, they were not. The chief editor of RSOS – a physical scientist – noted that if we, as editors, were sufficiently convinced by the rigour of the method then there would be nothing to fear by considering the study further, and potentially something to gain (a point well made in light of the framing of the proposal to demonstrate a new and more rigorous approach to conducting psychological science). In the same vein, the popular choice among the community on social media was that it should not be in

the editor's prerogative to allow their prior beliefs as to the likely existence of a phenomenon to hinder research from being undertaken, which is of course an especially salient issue for assessing a Stage 1 RR, because a RR, unlike a regular article, provides the opportunity for the review process to hinder (though not outright block) certain research questions from being asked in the first place.

I did not fully agree with the arguments against my original position (and views were mixed), but the response gave me pause and on this basis I decided to send the manuscript out for in-depth review. This does not change my belief that psi almost certainly does not exist, and I admit I have taken the decision to review this paper with lingering apprehension given potential consequences for the reputation of the journal (and potentially the Royal Society) but, far more importantly, for misleading the public, IF evidence that could be construed in favour of psi were to be produced. None of these views are intended to bias or influence the reviewers or authors in their assessments or revisions, and I will do my best to put my beliefs aside in rendering any editorial decisions or recommendations. My position (and biases) are simply noted here as matter of record in the event that the manuscript progresses to Stage 2 acceptance and these reviews are published.

Turning to the reviews themselves, the assessments are mainly positive but there were four major issues identified by multiple reviewers. The first issue, raised by Reviewer 1 and 3, is whether (and how) the study will meet its desired aim of testing the credibility of the ultra-transparent research tools and methods being proposed when there is no comparison to regular methods. The second issue, raised by Reviewers 1 and 2 is whether the authors have taken sufficient steps to eliminate experimenter bias and ensure adequate blinding. The third point, raised by Reviewers 3 and 4 is whether the authors should consider undertaking a systematic cost/benefit analysis of the proposed new methods, and whether they believe this approach is necessary across psychological science more widely (if so, why; if not, why not?). On this third point I believe it would be reasonable to reserve this for the Discussion at Stage 2, perhaps with some foreshadowing in the Stage 1 manuscript. And fourth, Reviewers 3 and 4 raise concerns about participant confidentiality and research ethics, with Reviewer 3 in particular questioning whether the proposal complies with the requirements of GDPR.

In addition to these comments, the reviewers also raise concerns about the risk of misleading readers and the public in event of a positive result (Rev 1, much in line with own views above), whether the "transparency gap" is a particular problem for psi (is it intended as a guard against fraud?), and whether a negative finding would in fact rule out psi (see Rev 3). Despite the very thorough methodological approach, the reviewers also note a number of areas where additional detail or justification is required, for instance in considering whether the Bayesian analyses sufficiently take into account a skeptic's prior (Rev 1) and questioning the basis of several aspects of the Bayesian analyses (Rev 2). Reviewer 1 also asks whether the steps taken to ensure computational reproducibility and public availability of the study materials are sufficient.

From an editorial perspective, I consider all of these points to be readily addressable in a revision. I therefore invite the authors to consider the reviews and revise the manuscript accordingly.

Response #1

We realize that a study related to parapsychology in any form is risky to publish by any journal, and the mere fact that our manuscript was sent out for review by a highly respected journal, such as RSOS, speaks to the outstanding scientific values that govern decision making and editor recruitment choices at the journal. Editors have great power as gatekeepers of science and it would have been very easy to reject this paper, as the editor's first instinct suggested. Instead, he went into great lengths to give a fair and unbiased treatment for this manuscript. We find the behavior of this extremely commendable.

Response to reviewer 1's comments

Comment #2

Different terms are used in the literature and the current manuscript – psi, precognition, ESP – in this review I will simply use “ESP” to refer to the phenomenon under scrutiny.

Below I have made remarks under the six headings (marked with *stars*) that the journal requested that I comment on.

The significance of the research question(s)

Without a clear definition of “significance” and what criteria I'm supposed to use to evaluate it - I'm not sure what is meant by this heading so I won't comment.

The logic, rationale, and plausibility of the proposed hypotheses

It might not be customary for reviewers to explicitly state their own beliefs about the phenomenon under scrutiny in a review. However, I think that might be relevant in this case due to the controversial nature of the phenomenon. I have a very strong belief that ESP does not exist. There is not only an absence of substantive theory or plausible mechanism – the purported effect defies the (comparatively well-established) laws of physics and common sense (how could casinos and ESP co-exist, for example).

Nevertheless, I am a subscriber to “Cromwell's rule” - one should not categorically rule out anything, for this makes it impossible to learn (see Lindley, 1991, Making Decisions). Consequently, I would be prepared to change my mind that ESP does not exist, its just that the evidence required for me to reach a favourable state of belief would be overwhelming. Practically, this means that however well designed and executed the proposed study is, a favourable outcome for the ESP hypothesis would not be sufficient to

convince me that the phenomenon was real (though it would move my belief incrementally in that direction). A more likely reaction from skeptics to a positive finding will be to question what other mechanisms may have given rise to the finding. This would be an appropriate Lakatosian defense (Meehl, 1990) given that the theory being defended (established laws of physics) have considerable “money in the bank” (a strong track record of empirical support).

I expect that my skepticism towards ESP is shared by many scientists (though the only empirical support for that statement I am aware of is Wagner & Monnet, 1979, cited in Bem, 2011), but my open-mindedness might not be. Some reactions to the publication of Bem (2011) suggested that it should have been desk-rejected because of the low likelihood of the topic under scrutiny (i.e., regardless of any methodological concerns). I am aware that similar discussions about the present manuscript have taken place and the fact I am being asked to comment on the plausibility of the proposed hypothesis suggests that this is a relevant criterion in the editor’s accept/reject decision making process. Consequently, I feel it is necessary to (briefly) make the case for why this manuscript *should not be* rejected on the grounds that the phenomenon under scrutiny is highly implausible to many scientists. Here are three reasons:

Response #2:

We would like to express our sincere gratitude to the reviewer for putting this section together in support of the publication of our manuscript. We fully agree with all three points.

Comment #3:

(1) It is possible (though probably rare) that phenomena that are a priori implausible, subsequently turn out to have some degree of accuracy. In the present case, this naturally seems highly unlikely, but it is possible. Thus, if ESP is real, rigorous scientific scrutiny of the kind proposed here will be necessary to establish its existence and improve our understanding of it.

(2) For implausible phenomena that are in fact not real, exposing them to robust empirical scrutiny may help to convince those who believe the phenomena are actually plausible. In the present case, I would speculate that the number of people who believe in ESP (some of whom may be in a position to substantially influence the lives of others) is not insignificant, and the actual or potential negative impact of them holding that belief (if it is false) could be quite serious. Thus, if ESP is not real, rigorous scientific scrutiny of the kind proposed here may help to convince some that their beliefs are unfounded.

(3) Finally, scientific methods do not give us direct access to the truth of the world around us. They are a set of tools that help us overcome the limitations of our own cognition and sensory apparatus. It is therefore possible that even if ESP is not real, evidence consistent with its reality is obtained using scientific methods. In such a scenario, the integrity of those methods would be called into question and efforts could be made to improve them. This was of course the dominant reaction of the scientific

community to the Bem (2011) study. It also seems appropriate (see the points above about Lakatosian defence; Meehl, 1990).

There are several specific objections that I think could be raised against the study that are reasonable and should be considered. On balance, I don't feel they outweigh the arguments made above, but are worthy of consideration. The objections are that the study is potentially (1) inefficient (poor use of resources); and/or (2) dangerous (due to the negative consequences of a false positive).

(1) It may be inefficient to use resources testing an implausible hypothesis when they could be used for greater benefit elsewhere. Relevant resources include participants' time, researchers' time, funding, editors and reviewers time, financial aspects etc. It seems reasonable that decision making is relegated to the relevant parties i.e., participants can decide if its worth their time (and are also compensated), the researchers have obviously decided its worth their time, funders have decided its worth their financial investment, and peer review is underway. A systematic efficiency assessment of such projects (those evaluating low plausibility hypotheses) might be useful but is beyond the scope of this specific case.

(2) The study is potentially dangerous due to the negative consequences of a false positive. Even if ESP is not real, a positive effect could be obtained because of chance statistical variation or some unanticipated experimental confound or bias. More than reasonable effort has been made to avoid this possibility, but the implications of a false positive should still be considered. If such a situation arises, I imagine that many within the scientific community will rigorously scrutinize the finding and evaluate it in its proper context (i.e., consider the low prior plausibility). But outside of scientific argumentation, the implications of such a result being published in a respected journal like *RSOS* could have serious negative repercussions. The authors have been (admirably) balanced in their write up of the article, but I do wonder whether the scenario of a positive result needs additional commentary, particularly in the abstract. Such commentary would reinforce that the finding needs to be evaluated in context i.e., that an individual with a highly skeptical prior should continue to think that the phenomenon is highly implausible, even given this positive result. It unclear if the authors would add this to their partly drafted discussion section, but there is currently little discussion of the fact that a positive finding would be contrary to well established physical laws – to such an extent that (probably for most scientists) the most likely explanation of a positive result would be a undetected methodological artefact, not ESP. The editor might also want to consider the journals response to a positive finding – for example, accompanying the article with an editorial (as with Judd & Gawronski, 2011) and/or peer commentary (as with Wagenmakers et al. 2011) intended to provide appropriate context. Any press release would also need to be very carefully worded to convey that, despite a positive finding, ESP remains vanishingly unlikely. There is something of an 'appeal to numbers' and 'appeal to authority' about what I am suggesting, which would probably be inappropriate in purely scientific argumentation, but may be warranted when we consider that the general public might rely on such cues to evaluate the credibility of the finding.

Response #3:

We agree with the reviewer's assessment of the above-mentioned critical issues.

We would like to add the following 'plain word summary' section to the abstract (or some other place seen fitting by the editor):

In case of a negative result:

“This project aimed to demonstrate the use of research methods that could improve the reliability of scientific findings in psychological science. Using rigorous methodology, we could not replicate the positive findings of Bem's 2011 Experiment 1. This finding does not confirm, nor contradict the existence of ESP in general, and this was not the point of our study. Instead, the results tell us that (1) it is likely that the original experiment was biased by methodological flaws, and (2) it is improbable that the paradigm used in the original study would be useful in detecting ESP effects if they exist.”

In case of a positive result:

“This project aimed to demonstrate the use of research methods that could improve the reliability of scientific findings in psychological science. Using rigorous methodology we could replicate the positive findings of Bem's 2011 Experiment 1. This finding does not confirm, nor contradict the existence of ESP in general, and this was not the point of our study. Instead, the results tell us that (1) it is unlikely that the positive findings of the original experiment can be explained only by the currently known methodological biases, and (2) more studies are warranted to investigate the causes for the positive effect. We do not know yet what these causes are, but it is important to note, that neither our study, nor the original study provide any evidence that these causes would be “paranormal”. Thus, it is still safe to assume that the effects at play are within the boundaries of known physics, psychology, and research methodology.”

Comment #4:

Interestingly, testing the ESP hypothesis is actually framed as being somewhat secondary to the main goal of the paper – which is stated to be a case demonstration of several research tools that can improve transparency during study execution. This is a bit odd, because the results of the study will not tell us much about the effectiveness of the credibility tools – that is not the hypothesis under scrutiny. I expect this unusual framing arose from the collaboration of skeptics and non-skeptics. It doesn't seem overly problematic; but it is a little confusing to follow the line of argument when its framed this way.

Response #4:

The primary aim of the project is to develop methods through which the credibility and rigour of research reports can be improved in a way that is verifiable for others. The replication study is used to demonstrate the use of these techniques in a case where expectations of trustworthiness and rigour are extreme. We made slight modifications to the text to make this more clear. We have reviewed the text to make sure that we do not say anywhere that we are testing or measuring the effectiveness of these methodological tools.

Comment #5:

The authors describe how there has been much attention to improving the transparency of study planning and reporting of studies, but less focus on study execution, a situation they refer to as the ‘transparency gap’. The authors make a good case for why this study in particular could benefit from enhanced credibility though a little more elaboration on why these specific tools might address specific concerns with ESP researcher might be helpful (e.g., there seems to be a suggestion that enhanced protection against fraud is necessary, but it is really explained why).

Response #5:

We do not believe that any of the main methodological tools described in the manuscript would be ESP-research specific. We state multiple times that expectations of rigour and transparency are very high towards parapsychological research studies. This necessitates a comprehensive approach to covering the transparency gap in order to improve credibility, but we cannot point at any one method that would be essential for ESP research but not for other types of studies. We now make this clear in the first paragraph of the “Methods for demonstrating protocol fidelity” section:

“... In this project, we showcase a large number of methodological tools that can be used to increase credibility and to ensure protocol fidelity, and we describe their usefulness below. We decided to use all of these techniques in combination in our particular project because we are undertaking a replication in a highly contentious topic. Even though not all of these techniques may be necessary or a good fit for all research projects, all of them are topic-neutral, and can be adopted in most areas of psychological research.”

Comment #6:

[NB – it may be worth noting that parapsychology research has a historical precedent for driving methodological reforms. For example, it seems to have been the impetus for an early prototype of the Registered Reports model (see Wiseman et al. 2019).]

Response #6:

This is an excellent point. We now included this note in the manuscript in the introduction section.

Comment #7:

The soundness and feasibility of the methodology and analysis pipeline (including statistical power analysis where applicable)

The attention to detail in this protocol is outstanding. Feasibility has been demonstrated empirically in a substantial pilot study conducted across two labs.

The statistical analysis plan seems appropriate and very thorough. The distinction between exploratory and confirmatory analysis is well articulated. Robustness checks are employed, and an interesting

hypothesis will be explored in exploratory analyses but treated separately from the primary hypothesis / analysis. Unfortunately, due to my own knowledge limitations, I'd say my scrutiny of this aspect of the protocol is fairly superficial and I'd recommend additional scrutiny by a statistical expert (particularly an expert in Bayesian statistics) if that is not already the case.

I may have misunderstood, but there didn't seem to be a Bayesian analysis that takes into account a skeptic's v. low prior probability (see e.g., Wagenmakers et al. 2011). I also think it could be made clearer that even if there is statistical support for M1, this cannot necessarily be directly interpreted as evidence for ESP. Here's a comment from Jeff Rouder on Andrew Gelman's blog to that effect: "Perhaps this is too simplistic, but I think it's important to separate truth and Bayes factors. I believe I can learn from comparing models without any of them be true, and, in that vein, use Bayes factor as a comparison metric. If I am judicious and wise, the BF comparison of select models may point me toward the next step in understanding the phenomenon at hand."

Response #7:

There are two things to consider here: (1) the prior probability distributions used in modeling M0 and M1 in the statistical model; (2) and the prior beliefs of the reader about the relative likelihood of the two models being true. We have gone into great lengths to use different probability distributions for M1, and the array of the priors we are using have been accepted by the consensus panel. On the other hand, our interpretation of the Bayes factors does not take into account the prior beliefs of the different readers or researchers about the relative likelihood of M0 and M1 to be true. We cannot anticipate the prior beliefs of all of our readers, so to be balanced, the interpretation of Bayes factors is currently assuming that the reader is undecided about the likelihood of the two models, and thus, finds both models (M0 and M1) equally likely to begin with. We don't see how we could change this without favoring one or the other model in our interpretation, and whatever unbalanced prior odds we would choose, it would probably not fit the actual prior beliefs of most readers anyway. To clarify this, we have added the following footnote to the text in the results section to clarify how to interpret Bayes factors:

"Note that Bayes factors are interpreted as the degree by which some prior beliefs about the relative odds of two models are to be updated. This means that the Bayes factors computed in our study should not be interpreted on their own, rather, they should be used to "update" the readers' prior beliefs about the relative odds of the two models. For example a person who found M0 to be a hundred times more likely to be true than M1 before the study, after observing a $BF_{01} = 1/25$ at the end of the study could update their beliefs accordingly, and still think that M0 is more likely, but only four times compared to M1. Because we do not know the prior odds of the readers, the interpretation of the Bayes factors in this paper are written assuming that the reader believed the two models to be equally likely before seeing our results. If this is not the case, the reader should update their beliefs accordingly."

Comment #8:

There are of course many other explanations for findings consistent with the M1 model other than ESP. Also, as argued by Fiedler & Krueger (2013), to assume a positive effect in this paradigm is evidence of ESP is to conflate the explanandum (the event to be explained) with the explanans (the argument used for

the explanation). This doesn't seem to be addressed in sufficient detail in the current draft of the discussion (though perhaps this doesn't need addressing until stage 2 review).

Response #8:

We made an effort to ascertain that our findings will not be over-interpreted, whatever the final results may turn out to be. We think this is clear from the pre-drafted conclusion section, and also, from the newly added "plain word summary" (see above in response # 3). At this point, we think it would be premature to draft the discussion section any further than it already includes, but we will definitely pay attention to provide a balanced and reasonably cautious interpretation of the findings at Stage 2. This is also aided by the fact that the team of authors include both sceptics and proponents of the ESP hypothesis.

Comment #9:

Regarding the pre-study questionnaires about belief in ESP and sensation seeking – it seems sub-optimal to take these measures before, rather than after, the core experiment. Pre-study delivery will presumably increase the risk of demand characteristics – specifically, participants working out what the study hypothesis is and changing their behavior. Admittedly, it is not clear to me exactly how their behavior would change in a way that might affect the results of the study. For example, assuming psi is not real, even if participants know the researchers are studying psi, this should not alter their performance on the task because there is no deliberate strategy they can adopt to achieve anything other than chance performance. Nevertheless, if there is no good reason to conduct these measures before the experiment, my recommendation would be to perform them afterwards as a principle of good experimental design. The same goes for any hypothesis-blind experimenters. [NB – its actually not clear to me if the participants and the experiments are supposed to be hypothesis blind or not, perhaps this could be made more explicit? Sorry if I've missed it]

Response #9:

The following section was added to the supplement:

“Rationale for not blinding participants or experimenters in the study regarding the nature of the study:

Both participants and experimenters in this study are aware of the hypothesis (although, participants are not made aware specifically that the experiment was designed to test precognition), just like in the original study (Bem, 2011). Participants were told by the experimenters in the original study that this is an experiment about ESP, and their attitude toward ESP was recorded exactly the same way as in our replication. It is unclear how the results would be altered by knowledge of the hypothesis in this study. If there is no ESP, the study outcome cannot be influenced by knowledge about the hypotheses. However, if there is ESP, belief about the possible existence of ESP (expectancy) could play a role in the outcome, or the exposure to the questionnaire could otherwise affect the end result. Both the ESP proponents and skeptics in the consensus panel urged us to keep as close to the original protocol as possible, even if some details seem unimportant for the main purpose of the study. We also see this as useful in averting some of the possible post-hoc criticism related to the study not being “exact replication” if the results turn out to support M0. The experimenters will also have to be aware of the hypotheses, because they will inform the participants about the nature of the study. These complications could be avoided if we fully computerized

the briefing just like the data collection process. However, this was rejected by some of the panel members in the consensus panel, because it would take the human contact out of the procedure, which may or may not have an important role in eliciting the effect.”

This section also explains why we would rather not change the timing of these assessments (aside from potentially influencing expectancy, exposure to the questionnaire might have other influences on the effect). To clarify this, we changed the text where we discuss these assessments in the methods section as follows: “We included these questionnaires to match the original protocol by Bem (2011) as closely as possible, because it is unclear whether exposure to these questionnaires would alter the outcome of the study. However, data from these questionnaires will not be used in hypothesis testing.”

Comment #10:

Whether the clarity and degree of methodological detail would be sufficient to replicate exactly the proposed experimental procedures and analysis pipeline

This is likely to be the most transparent research project I’ve ever reviewed. The methods are clearly explained and the authors have said they will share relevant materials, such as instructions to experimenters. Additionally, they will provide at least one video of the experiment being performed which will aid any replication efforts.

A few comments/suggestions:

(1) “Stimulus images used in the study can be obtained free of charge for researchers from the administrators of the NAPS and IAPS image sets.” – this is not ideal from a reproducibility perspective because the long-term preservation of and access to these materials relies on a third party. If the system maintaining these image sets were no longer operating in 10 years time for instance, then it may no longer be possible to conduct a high fidelity replication of the study. I understand that there may be rights issues that prevent this, but would it be possible for the researchers to share these materials directly themselves?

Response #10:

The user agreement of both the NAPS and IAPS images restrict the use of the images to the research study and forbids sharing them with those not involved in the research project. Since currently the images are readily available for other researchers to use from the right owners, instead of sharing them directly, we propose the following: We have now deposited the images in a private OSF repository to preserve them for the future. In case the original images will not be available anymore from the original sources (and provided that this would not involve violation of rights), we would share the images with other researchers upon request. We have made a note of this in the manuscript.

Comment #11:

(2) “These issues are discussed in the ‘Additional methodological considerations’ section in the Supplement.” – I don’t know where to obtain the supplement so I have not evaluated any part of it.

Response #11:

The Supplement was attached to the submission via the editorial system, but it is also available via OSF: <https://osf.io/dbprj/>

Comment #12:

(3) The analysis code is super clear and well annotated. It has been shared on the OSF and already successfully tested on the pilot data. We can even see it operating for ourselves on the provided website. This is beyond excellent. One suggestion would be to take steps to ensure that the software environment in which the analysis is performed can also be preserved. There are several tools available that can help with this, for example Binder or Code Ocean. I've found Code Ocean quite straightforward to use.

Response #12:

We will add the final analysis code to Code Ocean or equivalent once the code can be considered finalized (at Stage 1 IPA).

Comment #13:

Whether the authors provide a sufficiently clear and detailed description of the methods to prevent undisclosed flexibility in the experimental procedures or analysis pipeline

The descriptions in the paper seem very clear to me. Execution of the protocol is going to be filmed and audited by independent researchers. Assuming those procedures are robust, it's hard to see how meaningful protocol deviations could occur and remain undisclosed.

The authors are eagerly constraining their own analytic flexibility by providing the exact code they will use to perform the analysis – that's about as good as it gets.

Whether the authors have considered sufficient outcome-neutral conditions (e.g. positive controls) for ensuring that the results obtained are able to test the stated hypotheses

It is not clear what such a positive control would be in the present design as there is no intervention.

Response #13:

We are grateful for these positive comments.

Comment #14:

Miscellaneous minor comments

“replication is a safe but costly method for verifying credibility” – I'm not sure what is meant here by “safe”, could you clarify?

Response #14:

We changed the word “safe” to “reliable” better reflect what we meant.

Comment #15:

“The credibility enhancing methods described in this paper in combination with previously established transparency best practices can effectively negate or reveal all known sources of researcher and publication bias.” – that is a very strong statement and it does not seem warranted. It is very difficult to measure bias and demonstrate it has been reduced or negated. We know that existing measures, such as pre-registration, are far from 100% effective (e.g., Claesen et al. <https://psyarxiv.com/d8wex>). Some of the current measures (e.g., direct data deposition) have not been tested (as far as I know).

Response #15:

We have changed this sentence to: “The credibility enhancing methods described in this paper in combination with previously established transparency best practices will be used to negate or reveal known sources of researcher and publication bias.”

Comment #16:

“Adult participants with no psychiatric illness who have not participated in the study before and who are not under the influence of drugs or alcohol at the time of the session will be eligible to participate.” – perhaps state that these are inclusion criteria so they cannot be later misused as post-hoc exclusion criteria.

Response #16:

We have added: “These criteria will be checked only before the start of data collection. No participant can be excluded based on these criteria after data collection has started.”

Comment #17:

“After receiving a briefing about the experiment and its goals” – who provides this briefing? Relevant whether it’s a hypothesis-blind experiment or hypothesis-aware PI, for example.

Response #17:

The briefing is delivered by the experimenter present at the data collection session. As stated above, everyone is hypothesis-aware.

We clarified the text: “After receiving a briefing about the experiment and its goals from the experimenter”

Comment #18:

“not all participants will get the same compensation to facilitate accrual” – The meaning of this sentence, and the general procedures for compensating participants, was unclear to me.

Response #18:

We clarified the text: “compensation of participants may differ between data collection sites” (e.g., some sites might offer university credit for participation, whereas others might offer monetary compensation, or no compensation at all)”

Comment #19:

Regarding direct data deposition (DDD) and born open data – is it theoretically possible for the code to be modified to allow a form of ‘man in the middle attack’ – for example, temporarily holding the data and enabling manipulation prior to deposition. This of course seems extremely unlikely – but the whole reason for thinking DDD is necessary is the potential for fraudulent interaction with the data – thus it seems reasonable to consider just how robust this system would be against a determined bad actor.

Response #19:

The code cannot be modified without it being detectable. The server contents are synced with a GitLab repository, this way the code running on the server is verifiable at all times. Any modification to the code would be visible. The code was checked by the IT auditor, and was considered to be safe. If the editor and/or the reviewer thinks this is necessary, we can ask the IT auditor to make a specific comment about security against “man in the middle”-type threats. More generally: the goal when designing our procedures was to protect against reasonably foreseeable threats. We presume that a well trained team of hackers would probably be able to hack our system. However, this would require considerable planning and time investment to do it in a way that it is not detectable, since it would require feeding fake data into the system very slowly so that it does not raise suspicion. We do not think that the stakeholders involved here would have such resources or would want to use such resources on biasing this particular study.

Comment #20:

“Any manipulation of the data is clear and can be backtracked at any time because of the audit trail kept by GitHub” – this implies that manipulation of the data would be possible, but traceable. Could the authors elaborate on how the manipulation could occur?

Response #20:

Data could be manipulated if someone got access to the server and modified the version of the data stored there. These data would be pushed to GitHub. This modification would be clear by cross-checking previous versions of the data (all previous states of the data are preserved on GitHub).

We clarified the text: “Any manipulation of the data (for example by modifying the server-side copy of the data) is clear and can be backtracked at any time because of the audit trail kept by GitHub”

Comment #21:

A potential opportunity for bias – can the experimenters observe participants’ performance? If so, this could influence their completion of the lab logs which could lead to exclusions. Experimenters should ideally not be aware of the hypothesis and unable to monitor participants performance.

Response #21:

We added the following section to clarify:

“Inclusion of data in analysis

All data will be entered into the data analysis that is collected during the main study (not the pilot study), except for data generated during system tests. (The experimenter ID(s) of the test account(s) will be specified in the pre-registered analysis code. We do not anticipate the need for substantial testing in live experiment mode. We may use this to make sure that the system is available for data collection before the day of the first live data collection session, and after unanticipated server-side events such as server shutdown/restart if any.) The pilot data will not be combined with the data collected in the main experiment, and it will not be used in the confirmatory analyses. No other data will be excluded from analysis for any reason. Contents of laboratory logs will not be used to exclude data from the confirmatory statistical analyses.”

Comment #22:

Perhaps an explicit statement that the pilot data is not being combined with the main data is warranted?

Response #22:

We added: “The pilot data will not be combined with the data collected in the main experiment, and it will not be used in the confirmatory analyses.” (see also response #21)

Comment #23:

“The failure to replicate/successful replication of previous positive findings with this strict methodology indicates that it is likely/unlikely that the overall positive effect in the literature might be/would be only the result of recognized methodological biases rather than ESP/biases.” – I don’t think this statement is accurate for skeptics with a low prior belief in ESP. If the study does show an effect consistent with M1, then a skeptic with a very low prior belief will still not be convinced by this single study. The possibility of some unanticipated/unrecognized methodological artefact would remain a more likely explanation (for some).

Response #23:

That is why we say that “Successful replication of previous positive findings with this strict methodology indicates that it is unlikely that the overall positive effect in the literature would be only the result of ***recognized*** methodological biases rather than ESP.” Later we also state that “However, there could be methodological biases that have not yet been recognized”

Reviewer 2's comments:

Comment #24:

Review of "Raising the value of research studies in psychological science by increasing the credibility of research reports: The Transparent Psi Project."

Declaration of conflicts of interest: I have been a participant in the first round of the Delphi process as laid down in the paper, but have withdrawn from the project because of a lack of time.

The authors present a replication effort of the notorious studies by Bem (2011), in which it was claimed that research participants can 'feel the future'. Bem's original work was widely criticized because sceptics believed the effects to be impossible, the statistical analyses to be unsound, and, perhaps most importantly, because several sceptics failed to replicate the original results. Bem's paper is seen as one of the main catalysts in the so-called replication crisis in psychology. Since 2011, many psychologists have called for reforms in the way psychological science is carried out. The authors of the present paper have implemented several of these methods (and then some more) in order to test the hypothesis that positive results in the parapsychological literature are due to bad research practices and methodological issues.

The authors of the present paper have employed an impressive range of safeguards to rule out any alternative explanations that have to do with the practical aspects of research, research design, and data handling should the project result in a positive finding, i.e. a finding that would be supportive of ESP.

Given the massive scope of the project, there are several levels at which this paper needs to be evaluated. I have structured my review according to the levels I personally find most relevant.

In short, I believe this project to have enormous potential - to my knowledge, it is the most comprehensive implementation of transparency and openness practices in psychological science to date. As such, regardless of theoretical importance, about which one may argue, this project deserves a place in the literature.

Relevance to field of parapsychology The present paper aims to make a theoretical contribution to the field of parapsychology by testing the hypothesis that positive results in the literature are due to bad research practices. Although I believe the project can make a theoretical contribution to parapsychology, I do not believe the project can give an answer to the broad main question asked here, namely whether positive findings in the parapsychological literature are due to bad research practices.

During the Delphi process, only one single experiment has been considered, and an experiment that has several problematic issues with regard to methodology. The number of trials per participant is rather small, to name just one example. Moreover, in several replication studies, including a large project presented at this year's convention of the Parapsychological Association, it has been shown that the sought-after effect is very likely not to replicate - regardless of the meta-analysis presented by the authors.

Obviously, the most likely reason sceptics will present for this is that these effects cannot exist, but even within parapsychology there are several theories that claim that ESP-effects disappear the moment they can reliably be used as a signal, or when 'organizational closure' (whatever that may mean) of a system is insufficient. The 'born open' nature of the project would directly conflict with the idea that anomalous phenomena only arise within rather isolated systems, and are only expressed as 'excess correlations' that sometimes appear to violate the normal laws of physics (cf. Von Lucadou e.a.) If one takes this position, one would even explicitly predict a negative finding for this project. Although this idea may seem very alien and weird to mainstream psychologists, it is getting traction within the parapsychological community - not so much because it provides a convenient escape from explaining why ESP-effects are so hard to replicate, but because there is increasing empirical support using novel experimental paradigms and analysis techniques for this idea.

In sum, the issue here is that only a positive finding would be a theoretically meaningful result for the field of parapsychology. It would lend credibility to the psi hypothesis, and as an additional bonus, would make short work of the above mentioned 'model of pragmatic information'. However, in the much more likely event of a negative finding, there are two explanations - the mainstream explanation (psi does not exist), but also the explanation that psi cannot be measured in this particular manner, whether it is because of the paradigm, or because of the open nature of the project. In either case, the authors cannot make the claim that this project is able to answer their main research question, namely whether positive findings in the parapsychological literature are due to bad research practices. The project should at least be expanded with the above mentioned novel methods (they're called correlation matrix experiments - I'm quite sure the authors are familiar with such experiments) in order to increase the theoretical value of the study.

Response #24:

Our main question in this study is whether the positive results reported by Bem in his original experiment 1 (2011), and later in the meta-analysis for the same experimental paradigm (2016) would replicate in a protocol where we attempt to control all recognized sources of experimenter bias. This is testing a prediction of what we call the "pure bias theory". We agree that there are multiple explanations for both a positive and a negative findings, and we list some of the possible explanations in our pre-written conclusions. We do not agree that a negative result would be meaningless for the field of parapsychology. As you mentioned, one conclusion that we would have to draw in that case is that "psi cannot be measured in this particular manner", which would cast serious doubt to the positive results of the original

Bem 2011 experiment 1 findings, and also to the positive findings of the Bem et al. 2016 meta-analysis including 14 studies using the same paradigm.

We are aware of a research method in parapsychology exploring ‘excess correlations’. However, to our understanding, in order to implement such methodology in our study, we would have to change the study design dramatically, and at the very least this would require the re-initiation of the consensus design process. We do not see this warranted at this point, considering that the consensus panel approved of our research design, with half of the panel members being “ESP proponents”.

Comment #25:

Alternatively, at the very least the authors might want to consider to ask each involved experimenter and lab lead to (privately) note their expectations for their own contributions and for the project as a whole, to learn more about experimenter effects.

Response #25:

It would be relatively easy to ask site PIs to note their expectations for the outcome of the project as a whole and for the data collected in their lab in particular.

It may be interesting to see the expectations of the site PIs with relation to their overall belief in ESP measured by the ASGS. So this dataset would be interesting from a meta-science perspective as well, how much people with different beliefs about the existence of an effect in general are convinced about the replication of an effect in a specific study. We could even add a question about whether the site PIs think that the original study had the potential to detect an ESP effect, if it existed, to see whether the particular person believes that this is the right research paradigm to test the effect or not (even if they may believe in ESP).

However, using this dataset to analyze the correlation of the outcome and experimenter expectations in the main study may be problematic. During the consensus design process, we got serious pushback from the “ESP sceptic” side of the consensus panel whenever we proposed to include any additional analysis other than the confirmatory analyses listed in our study, even if we explicitly labeled them as exploratory. Any additional analysis is perceived by the “ESP sceptics” as a possibility that an ESP-like chance finding would occur. This is possible in this case as well, since there are only two good reasons for expectations to be correlated with the outcome of the study: ESP or fraud. Thus, we would rather not include this analysis in our study to stay within the agreed upon analyses with the consensus panel.

Nevertheless, we could collect this dataset as indicated above, and the reviewer or other researchers can pre-register an analysis based on this dataset.

However, it is not clear what the reviewer means by keeping these ratings private. We could collect this dataset from PIs at the same time when we ask them to fill out the ASGS, so that only the PIs of the project and the auditors would have access to this data until the end of the study, when we would make them available with the rest of the data. Would that be satisfactory to the reviewer? If the reviewer meant that this data should be kept private (only accessible for the person doing the rating) until the end of the

study, we are afraid that would leave too much room for bias, so it would not be in line with the principles this study is built on.

Comment #26:

In particular, when combined with a parallel replication without the all the stringent measures named in the paper (see below), this would make a very interesting and compelling theoretical contribution.

Response #26:

See responses below and above.

Comment #27:

Relevance to methodology Within this project, many of the innovations and reforms that have been proposed in the literature to improve the quality and reliability of the research process come together. I can only applaud the enormous effort the authors have made, and am deeply impressed with the result. In particular the shiny apps are absolutely fantastic and an absolute breakthrough in how to communicate science to a broader audience. This alone would warrant a publication on science communication.

That said, the experiences of the authors with this process are to me the most interesting part of this study - I feel these should be the main focus of the paper. A structured - quantitative and qualitative - evaluation of the entire process, with all relevant parties is indispensable. A very interesting and perhaps even necessary addition to such an evaluation would be a parallel project in which the transparency measures such as born open data are not implemented, save the independent audit at the end. This would allow for a direct empirical test of the effect and effectiveness of using open practices for increasing data and process quality. Suppose that the audit indeed finds deviations from protocol and/or irregularities in the data which would alter the conclusions of the parallel project, this would lend strong support to the idea that open practices are a necessity for reliable science.

This is a very important point, as many of the perceived benefits of increased transparency in science are based on assumptions. Although these assumptions are very plausible, I would strongly welcome a proper empirical evaluation, in particular when combined with an assessment of experimenter expectations and belief (see above).

Response #27:

To organize a replication attempt without the methodological tools we propose is a great idea, and would provide for an interesting contrast to the main study. However, this would only be meaningful if this parallel replication not using the credibility-enhancing tools would be also well powered. As shown in our power analysis, this would require thousands of participants, effectively doubling the sample requirement for our project. We don't see this as feasible within the scope of the limited timeline of our project dictated by the external funder. Also, it would be a great challenge on its own to figure out how to make the two replication projects comparable. Such a project would only be informative of the underlying biases and errors in the field, and the effects of the new credibility enhancing methodologies if participating labs and experimenters would be blinded to their "group allocation" and the purpose of the study. Such blinding would probably not be possible with our project, given that many of the stakeholders

are already aware of our project and the tools we propose. We have been developing another project where this might be feasible, but it would not be completed within the timeframe of the current project.

We also think that the experiences we gained during this project are very valuable. We expect to talk about these experiences in the discussion section of this paper to some extent, but this manuscript is already quite long, so we might write up a separate publication on this topic. We also intend to publish a separate piece on the consensus design process and a tutorial on how to implement the real-time solutions used in our project.

Comment #28:

In the light of this previous point, another important part of the evaluation should be a detailed cost-benefit analysis of all the measures the authors have implemented. It is vitally important to realize that setting up a Delphi process, implementing parallel replications and a ‘born open’ data infrastructure, and an independent data and procedure audit are not only very time consuming, but also very expensive, and potentially out of reach for many researchers, let alone that many paradigms may not be implemented in such a way. Finally, a lot of flexibility is lost as a result of incorporating a Delphi process or requiring parallel multilab replications. Of course this is a trade-off: the situation pre-replication crisis was one of too much flexibility for researchers. It is a very interesting and highly relevant question where to put the line between flexibility and accountability. Overall, I must admit that on a personal level I feel the measures implemented in this project to be ‘too much of a good thing’, and from my personal experience as Director of Research and Data Infrastructure, I believe many researchers feel the same way. I would be very interested in hearing the authors’ opinion on the matter whether the stringent measures in this project are truly necessary to guarantee scientific integrity or that they see the present project primarily as a proof of concept of several transparency reforms suggested by the open science community. This is not a fully empirical matter, obviously, but the authors of this paper and their co-workers are in the unique position to give a first-hand insight into how operating within very strict conditions works out. If not in this paper, I would strongly encourage the authors to give an account of their experiences beyond what we typically read in research papers.

Response #28:

Thank you for this suggestion. We plan to include such a cost-benefit analysis in the discussion section of the manuscript at Stage-2, since that is the point when we will have experience with the use of all of the proposed credibility-enhancing tools.

We now added to the aims of the project: “to assess the costs and benefits of implementing these methodological advances in a multi-lab project”

Comment #29:

As a side note, I would strongly recommend the editor of this journal to consider a call for opinion papers on this project, in particular to spark a debate within the community about the necessity for reforms towards openness and accountability, and how far such reforms should go.

In sum, I feel there is a great potential in this paper: regardless of the theoretical contribution, of which I have some reservations with regard to its value, these authors are pioneers in working within the most stringent conditions of transparency. A document of their experiences is extremely valuable for the scientific community.

Response #29:

We agree that a call for opinion papers is a good idea.

Comment #30:

Legal, and ethical issues With regard to the measures implemented in the project, I do have several concerns about the ethical and legal aspects, in particular in the light of the European privacy legislation (GDPR). Given that under most ethics codes, compliance with legal regulations is required I have grouped ethics and legal together here. With regard to the ethics of the experimental procedure, I have absolutely no problems. However, the paper does not give sufficient information to evaluate if all legal regulations with regard to data protection have been met. I would like to stress here that this is not the authors' fault - there are many unclearities regarding the GDPR, and in particular how the GDPR affects open science. However, given the pioneering role of the authors I am afraid this point needs to be brought up.

The goal of the GDPR is to give citizens (data subjects) more control over how their personal data is used. The GDPR has derogations for research, but researchers should be precise and very careful in documenting how they protect the legal rights of their subjects.

Personal data has a broad definition under the GDPR - any data that may be traced back to an individual is personal data, even if a person is not directly identifiable. The authors have clearly recognized this point - for example, they even take care that participants are not identifiable by means of a time stamp. Of course, if it is fundamentally impossible to re-identify participants, data is anonymous and the GDPR does not apply. However, in order to claim that data is truly anonymous, one should run an analysis on parameters such as dispersion and separability of variables and records, which will be very difficult within the born-open model, as the dataset continuously changes. From a legal perspective, the born-open dataset as presented here will by definition be a pseudonymized dataset.

In order to operate transparently within the GDPR, pseudonymization of personal data prior to publicly sharing is not sufficient. In particular, explicit informed consent is required in which it is very clearly explained that data will be made public, what the risks are to the data subject (I can imagine these are very limited in this particular case), and what measures have been taken to protect the data subject's identity, how the data will be used, and that the data subject voids her/his rights on data removal, data correction, etc. The data subject needs to explicitly agree on these points. It would be very helpful if the authors could share their informed consent forms so other researchers who want to use the 'born open' model could use these to write their own.

Response #30:

To our knowledge our project is perfectly GDPR compatible. The data we collect cannot be considered personal data, since they can not be traced back to any individual. There are multiple ways we make sure that this is really the case: Dataset is pushed to GitHub in bursts of 200 rows instead of a row at a time, to make data points untraceable based on knowledge about when the data were published. Time stamps and lab IDs are encrypted before the push to GitHub, and the key is not shared publicly. Data on age are collected in age ranges instead of exact age.

Nevertheless, we are transparent in our informed consent form that dataset collected in this project will be publicly accessible, and that it is not possible to withdraw data from the public dataset once it has been collected (partly because we ourselves would not be able to identify which rows in the dataset belong to which participants).

The informed consent form will be different from lab-to-lab. The informed consent form used in our home lab is accessible via this link: <https://osf.io/eh94q/>

Comment #31:

Another legal issue arises with the external audit. The auditing party will need access to personal data, not just of research participants, but also of the researchers. Have the authors documented this process, and how have they arranged the necessary processing agreements and obtained informed consent?

Response #31:

We do not collect any personal data from participants, since the data they provide cannot be linked to them personally. Even if auditors do have access to timestamps of data collection, they would need additional insider information to identify any data: specifically, this is only possible for research session where there was only one participant present at the session, and there was no other session going on at the same time at any other lab (since the study is designed for data collection in groups, even the occurrence of such a session is low). For such a session, they would further need information on who was the participant at this particular session. We do not keep records of who participated at which session, and thus, auditors do not have access to this information.

We do collect personal data on experimenters and site-PIs, specifically, we have their contact information, their affiliation, their full names, and their Australian Sheep-Goat Scale score, and from the experimenters we have video recordings of a mock research session. The experimenters and site-PIs will be informed of this when they provide this data and how they can remove personal data from our database. For more details, see response #31 below, our Self assessment in preparation for DPIA document at <https://osf.io/sv5xt/>, and the informed consent forms for the participants and experimenters at <https://osf.io/eh94q/>

Comment #32:

In sum, the most important question I have for the authors at this point is whether a data protection impact assessment (DPIA) has been carried out, as required by the GDPR for a project as this (following the guidelines from the Article 29 Working Party on data protection impact assessment, see

http://ec.europa.eu/newsroom/document.cfm?doc_id=47711) During a DPIA, data protection issues are systematically charted and evaluated from a multi-stakeholder perspective. The resulting document can be used to demonstrate compliance with the GDPR. It is critically important to realize that a DPIA does not result in a ‘yes’ or ‘no’ with regard to GDPR compliance, but rather to find the optimal way to carry out a research project in such a way that data subjects’ fundamental rights to data autonomy are safeguarded, and where this is not possible, this is clearly communicated to the data subjects.

If a DPIA has been carried out, this document would be extremely valuable for the open science community, and warrant a publication on its own, as other researchers who want to implement practices can refer to this document. If no DPIA has been carried out, can the authors specify why a DPIA has not been deemed necessary, and whether this was decided by an ethics committee, or a data protection officer of the host institute?

Response #32:

We do not have a formal DPIA yet (or a statement about its necessity). Nevertheless, we have started the process with our local data protection officer. At this point we share the detailed self assessment we created in preparation for the DPIA (see on OSF: <https://osf.io/sv5xr/>). In this document we describe in detail how we ensure the protection of personal data. As the reviewer notes, this is not so much an issue for research participants, as we do not collect any personal data from them. However, we do collect personal data from experimenters and site-PIs. We believe that our data management procedures are compliant with GDPR requirements, since all people involved are informed about data handling, and how those from whom we collect personal data (experimenters and site-PIs) can request the deletion of their personal data from our database.

Experimenters and site-PIs are also informed that they cannot remove their ASGS score from the database. However, without the personal data that they can delete, this data can not be directly linked to them personally. Nevertheless, a person with insider information about the lab members at the participating lab, and how their lab works, and access to the experimenter ID, laboratory ID, and session timestamps may be able to triangulate their personal identity, so they could eventually link the ASGS total scores (beliefs about ESP) to them. (Please note that experimenter ID, laboratory ID, and session timestamps are all encrypted in the public dataset, so only the core research team and auditors will have a chance to do this). We are transparent about this as well in the informed consent form. We think that the retention of this anonymized data may be useful for researcher for meta-science purposes and also for being able to track the progress of the study at different research sites. However, if for some reason the retention of this anonymized information is seen as problematic by the data protection officer, we are willing to modify our data collection procedures or even exclude this data entirely from our study, since it is not essential for answering the main research questions.

Response to Reviewer 3’s comments

Comment #33:

This is a review and comment on Kekecs et al. stage 1 RR entitled “Raising the value of research studies in psychological science by increasing the credibility of research reports: The Transparent Psi Project.”

As I understand the project, two main aims motivate the research team :

1- Showing a concrete way to carry out a highly rigorous protocol based on the recent progress in methodology.

2- Using this protocol to test a controversial hypothesis in psychology

The significance of the research question(s):

This project is of high importance for the field. It is crucial to provide examples on how to manage a scientific project with all the tools supposed to increase transparency and scientific quality. I also appreciate the idea of hypothesis testing applied to Bem’s work because Bem’s findings triggered a lot of questions in our field but also in the general population concerning the rigor of psychology and science in general. Hence, I can say that this work really deserves to be carried out.

The logic, rationale, and plausibility of the proposed hypotheses

The two main aims are clearly linked and this is a great idea to use a methodological interest to deal with the test of a (highly) controversial hypothesis. The plausibility of the hypothesis is one of the questions of the project because of the “meta-science” aspect of this work. All in all, the introduction is very clear and perfectly shows the relevance of the work to be done.

The soundness and feasibility of the methodology and analysis pipeline (including statistical power analysis where applicable):

There is no doubt for me that the project is feasible and that the methodology is adapted to the aim of the authors. The research team did everything to ensure the quality of the protocol and the scientific rigor proposed here is impressive.

Whether the clarity and degree of methodological detail would be sufficient to replicate exactly the proposed experimental procedures and analysis pipeline:

There is no doubt that this procedure could be replicated. The precision of information in the manuscript and online is far above the classical reports in the field. Often, the authors even propose several ways to ensure the precision of their descriptions (e.g. videos of experimenters giving the instructions).

Whether the authors provide a sufficiently clear and detailed description of the methods to prevent undisclosed flexibility in the experimental procedures or analysis pipeline:

Degrees of freedom are very limited because of the clear details provided in the methods and analysis section. Moreover, This project benefits from a committee composed of “pro ESP” and “against ESP” which limits flexibility even more drastically.

Whether the authors have considered sufficient outcome-neutral conditions (e.g. positive controls) for ensuring that the results obtained are able to test the stated hypotheses:

The authors use the original protocol for the replication. The protocol is well suited to test the hypothesis. There is a “control condition” with neutral images but this condition will not be used in data analysis. I understand that the authors want to keep this condition in order to match the original protocol. However I would like to know more about their motivation to keep this condition out of the analysis. Indeed, we could also consider the comparison between the “erotic” and the “neutral” condition as relevant. The “neutral condition” could serve as a choice “baseline” representing the “no effect condition”.

Response #33:

We aimed to conduct a replication as closely matching the original study as possible including the main confirmatory analysis. The original study contrasted the proportion of correct guesses in erotic trials to that expected by chance.

We agree that contrasting correct guess rate in the neutral and the erotic trials makes sense if we want to test the hypothesis that type of the stimulus is important in determining the effect. However, this is not the main focus of our study. Adding a hypothesis test related to this would increase the required sample size by hundreds of participants, and we do not see this as a worthwhile at this point without high quality evidence for the existence of the effect in the first place. During the consensus design process, we got serious pushback from the “ESP sceptic” side of the consensus panel whenever we proposed to include any additional analysis other than the confirmatory analyses listed in the current version of the manuscript, even if we explicitly labeled them as exploratory. Any additional analysis is perceived by the “ESP sceptics” as a possibility that an ESP-like chance finding would occur.

We encourage everyone who has a reasonable hypothesis to pre-register a statistical test themselves. The data will be openly available to test a hypothesis about the effect of erotic vs. neutral stimuli.

Comment #34:

I also have some others comments :

1) I do not understand why primary analysis are performed with Bayes Factors and then Bayesian parameter estimation is used for Robustness check. The procedures proposed by Kruschke (2018) could be used in the primary analysis for both parameter estimation and hypothesis testing.

Response #34:

There are several reasons for choosing the Bayes factor analysis approach instead of the Bayesian parameter estimation as our primary hypothesis testing method. First of all, the consensus panel predominantly was in favor of the Bayes factor approach instead of the parameter estimation approach. This was probably due to the role that the Bayes factor approach played in the statistical criticisms against the findings reported by Bem (2011). This approach lets us use the priors previously proposed in the literature in the commentaries by Wagenmakers and Bem (Bem, Utts & Johnson, 2011; Wagenmakers, Borsboom, Kievit & van der Maas, 2011). Our simulations demonstrate desirable operational characteristics for the Bayes factor approach. Also, our simulation analyses showed that the approach described by Kruschke & Liddell (2018) requires higher sample size targets to produce the same operational characteristics (inferential error rates) compared to the Bayes factor. This is an important consideration when the execution of the study already requires thousands of participants.

Nevertheless, as the reviewer is aware, we also apply the Kruschke & Liddell (2018) method in our study to demonstrate robustness to different statistical approaches, just not as a primary analysis tool. Furthermore, switching the primary and the robustness tests may require us to re-initiate the consensus panel process, delaying study execution significantly. So, given that the reviewer does not point out any specific criticism about the Bayes factor approach we are proposing, and given the potential high cost in applying the Bayesian parameter estimation as a primary analysis, we would rather stay with our current analysis plan.

We have now included the following section in the Supplement:

“Rationale for choosing Bayes factors for statistical inference instead of Bayesian parameter estimation:

There are several reasons for choosing the Bayes factor analysis approach instead of the Bayesian parameter estimation as our primary hypothesis testing method. First of all, the consensus panel predominantly was in favor of the Bayes factor approach instead of the parameter estimation approach. This was probably due to the role that the Bayes factor approach played in the statistical criticisms against the findings reported by Bem (2011). This approach lets us use the priors previously proposed in the literature in the commentaries by Wagenmakers and Bem. One of the few cases where the Bayes factor approach is thought to be appropriate is in fact in such research questions, where the null model is meaningful and plausible. Our simulations demonstrate desirable operational characteristics for the Bayes factor approach. Also, our simulation analyses showed that the Kruschke (2018) method requires higher sample size targets to produce the same operational characteristics (inferential error rates) compared to the Bayes factor. This is an important consideration when the execution of the study already requires thousands of participants.”

Comment #35:

2) I hope I did not miss anything in the manuscript but why 51%? What is the logical justification of this threshold?

Response #35:

We have now included the following section in the Supplement:

“Rationale for choosing +1% successful guess rate as the smallest effect size of interest:

The smaller the effect we want to be able to detect the larger the sample size required, thus, we needed to draw the line somewhere to ensure feasibility of study execution. Our goal in this study was not to prove or disprove the ESP model. Rather, we wanted to evaluate the likelihood that the results presented in the original study (53% (90% HDI: 51% - 55%)) were biased. The ESP proponents and ESP opponents agreed with using +1% successful guess rate as a minimal effect size of interest in light of the consensus derived conclusions for the study. Specifically, they all agreed that if the study yielded support for M0, the conclusions will include that:

‘...The failure to replicate previous positive findings with this strict methodology indicates that it is likely that the overall positive effect in the literature might be the result of methodological biases rather than ESP. However, the occurrence of ESP effects could depend on some unrecognized moderating variables that were not adequately controlled in this study, or ESP could be very rare or extremely small, and thus undetectable with this study design. Nevertheless, even if ESP would exist, our findings strongly indicate that this particular paradigm, utilized in the way we did, is unlikely to yield evidence for its existence...’

We believe that the +1% smallest effect size of interest (SESOI) threshold proposed in the manuscript is consistent with these conclusion (as shown in the quote, the conclusion allows for the existence of an extremely small effect even if M0 is supported, but also notes that if this would be the case, the current paradigm is very inefficient to provide evidence for it).”

Comment #36:

3) Why BF = 25 for decision making? What is more it is likely that more observations will be needed to reach BF = 0.04 than 25 (e.g. see Schönbrodt, Wagenmakers, Zehetleitner, & Perugini, 2017). Is the protocol “fair” and evenly designed for both hypotheses ?

Response #36:

We have now included the following section in the Supplement:

“Rationale for choosing the Bayes factor thresholds to be 25 and 1/25:

We have surveyed the consensus design panel about what would they consider as appropriate confidence or decision making criteria in our study, given the conclusions that we aim to draw from our study. After aggregating the responses of the consensus panel, the inference thresholds of BF 25 or 1/25 and $p < 0.005$ were proposed. These thresholds were deemed to be acceptable by the consensus panel members.”

When the chance of successful guesses in the population was 50%, the probability of:

- correctly supporting M0 was 0.962

- inconclusive study was 0.038
- falsely supporting M1 was lower than 0.0001 (no false support for M1 was found in the 10000 simulations)

When the chance of successful guesses in the population was 51%, the probability of:

- correctly supporting M1 was 0.999
- inconclusive study was 0.0009
- falsely supporting M0 was 0.0006

We believe that these are acceptable error rates for both eventualities, giving a fair chance for supporting both M0 and M1, and a very low chance for detection of false supporting findings.

Comment #37:

4) Why do you choose to report the 90% HDI and then to see if 95% of the posterior falls inside or outside the rope ?

Response #37:

Since we use a one-tailed test, the area falling to the right of the 90% HDI is 95% of the total posterior density. Similarly, the area falling to the left of the upper bound of the 90% HDI is the 95% of the total posterior density. So what we are proposing is to see whether the 90% HDI bounds fall to the left or to the right of the ROPE threshold, or whether the two boundaries contain the ROPE threshold.

Comment #38:

5) Why do you choose not to blind experimenters? You take a lot of precautions in your protocol. I agree that the influence of experimenters may not be the largest source of bias but still, don't you think that blinding the experimenters could improve your protocol?

Response #38:

(This response is the same as response #9):

The following section was added to the supplement:

“Rationale for not blinding participants or experimenters in the study regarding the nature of the study:

Both participants and experimenters in this study are aware of the hypothesis (although, participants are not made aware specifically that the experiment was designed to test precognition), just like in the original study (Bem, 2011). Participants were told by the experimenters in the original study that this is an experiment about ESP, and their attitude toward ESP was recorded exactly the same way as in our replication. It is unclear how the results would be altered by knowledge of the hypothesis in this study. If there is no ESP, the study outcome cannot be influenced by knowledge about the hypotheses. However, if there is ESP, belief about the possible existence of ESP (expectancy) could play a role in the outcome, or the exposure to the questionnaire could otherwise affect the end result. Both the ESP proponents and skeptics in the consensus panel urged us to keep as close to the original protocol as possible, even if some details seem unimportant for the main purpose of the study. We also see this as useful in averting some of

the possible post-hoc criticism related to the study not being “exact replication” if the results turn out to support M0. The experimenters will also have to be aware of the hypotheses, because they will inform the participants about the nature of the study. These complications could be avoided if we fully computerized the briefing just like the data collection process. However, this was rejected by some of the panel members in the consensus panel, because it would take the human contact out of the procedure, which may or may not have an important role in eliciting the effect.”

Comment #39:

Overall I find this project of high interest and I want to underscore the meticulous job done by the research team. I look forward to their response to my comments.

Minor comments :

Page 17, Line 19: I think that there could be a punctuation problem in the following sentence “We will build an intercept only mixed logistic regression model (using the glmer function in the lme4 package in R, Bates et al., 2019). The lme4 package. R package version, 2(1), 74.) allowing for a random intercept for participants, to predict the outcome (success or failure) of the guess in experimental trials.”

Response #39:

The sentence was revised:

“We will build an intercept-only mixed logistic regression model (using the glmer function in the lme4 package in R, Bates et al., 2019). This will allow for a random intercept for participants, to predict the outcome of the guess (success or failure) in the experimental trials.”

Comment #40:

Page 21, Line46: missing parenthesis: “(Bem et al., 2016; Bierman, Spottiswoode, & Bijl, 2016; Rouder & Morey, 2011”

Response #40:

Corrected.

References (added by reviewer #3)

Kruschke, J. K. (2018). Rejecting or Accepting Parameter Values in Bayesian Estimation. *Advances in Methods and Practices in Psychological Science*, 1(2), 270–280. <https://doi.org/10.1177/2515245918771304>

Schönbrodt, F. D., Wagenmakers, E.-J., Zehetleitner, M., & Perugini, M. (2017). Sequential hypothesis testing with Bayes factors: Efficiently testing mean differences. *Psychological Methods*, 22(2), 322–339. <https://doi.org/10.1037/met0000061>

Responses to Reviewer 4's comments:

Comment #41:

Comments to the Author(s)

Overall assessment:

Thank you for the opportunity to review the research proposal for the 'transparent Psi project'. I was very excited to read it. It is a carefully planned proposal that hits a soft spot at the right time – the debate around ESP following Bem's 2011 article and the subsequent replication attempts. The proposed research plan offers an impressive, extensive battery of tests and safeguards that will allow to unequivocally settle the debate. I am also a bit skeptic. My skepticism has nothing to do with the ability of the research team to execute the ambitious plan; on the contrary, I am convinced that the plan is adequate and the research team has the capacity to execute it. Rather, it is about the cost-benefit of the proposed safeguards and feasibility of implementation in future studies. I wonder if the time, effort, human labour, and dollar costs of such a study can be invested regularly in scientific investigations. My guess is that this type of concerted effort might be limited to a (very) small number of critical studies that attempt to resolve an ongoing dispute. Overall, this looks like a terrific project and I strongly recommend it goes ahead. I would like to see practical, smaller scale and cost-effective solutions to the credibility problem.

Response #41:

We plan to include such a cost-benefit analysis in the discussion section of the manuscript at Stage-2, since that is the point when we will have experiences with the use of all of the proposed credibility-enhancing tools.

We now added to the aims of the project: "to assess the costs and benefits of implementing these methodological advances in a multi-lab project"

Comment #42:

Significance:

Very high, given both the scientific and the public interest in ESP, and the ongoing controversy surrounding Bem's original study (2011) and subsequent replication or analysis attempts (eg Wagenmakers and colleagues, 2011, 2015).

Logic and rationale:

The hypothesis are broadly based on the original study, with M1 and M0 indicating support for better-than-random guessing, vs random. As such, they are adequate.

Methodology:

The proposed methods are sound and clearly explained. The proposal outlines a number of key features that serve to ensure the credibility of the findings. These include deposition of protocols and testing logs, born open data, other bias reduction, and increased transparency via third party observers. Each 'tool' has its own merit, and together they form an almost formidable wall to protect the integrity of the study (notwithstanding my concerns about the costs required for the implementation).

Other considerations:

- The proposal highlights the advantages of real time report, with which I wholeheartedly agree. In a nut shell, data is immediately deposited to a publicly open repository and the results of the study will be analysed as the data is accumulated (not at the end of data collection). This offers greater transparency. A potential risk of the real time report though is that spurious effects could look significant at some arbitrary point in time and members of the public or media might this snapshot in time to draw conclusions. Are there appropriate mechanisms to minimise that kind of risk?

Response #42:

We have now included a warning message in the real-time research report where appropriate noting that “Result not yet final!”

“*Data presented on this page represent the current trend calculated from the data. The results should not be over-interpreted! Random variations may cause the data to cross the decision thresholds. Statistical decisions will only be drawn at the pre-specified stopping points. The next stopping point will be at reaching X trials”

Comment #43:

- A potential risk of the born-open data is that sharing data in real time that compromise that anonymity of the participants. The research team had taken a number of measures (encrypted timestamps, group release) to minimise this risk and I have found the overall strategy satisfactory.
- The research team had offered a number of exit points for data collection, based on predetermined numbers of trials and subjects. Perhaps they would like to consider Bayesian Optional Stopping (e.g., Rouder, 2014, PB&R) or if not explain why it is not adequate.

Response #43:

The following section was added to the supplement:

“Rationale for not using Bayesian optional stopping:

There are multiple reasons for using a sequential analysis plan instead of optional stopping. First of all, we use a hybrid frequentist-Bayesian method for statistical inference to increase the robustness of our inference (see reasoning for the use of a mixed model logistic regression together with a Bayesian binomial test in the main text). Optional stopping would very soon decrease the p-value threshold in the mixed-model logistic regression to unattainable values. This could be overcome if we only did the mixed-model analysis at reaching the desired BF threshold. However, this has multiple disadvantages: our

simulations show that this way the inferential error rates would be too high in a scenario where there are systematic personal differences in correct guess rate. Furthermore it is not clear how should we proceed if the BF threshold is reached but the p -value threshold is not. Second, the benefit of Bayesian optional stopping is not as great in our multi-site study, where there is less central control over participant flow.”

References:

- Bem, D. J., Utts, J., & Johnson, W. O. (2011). Must psychologists change the way they analyze their data? *Journal of Personality and Social Psychology*, 101, 716-719.
- Wagenmakers, E.-J., Wetzels, R., Borsboom, D., Kievit, R. & van der Maas, H. L. J. (2011). Yes, psychologists must change the way they analyze their data: Clarifications for Bem, Utts, and Johnson (2011). doi: [10.31234/osf.io/tvarg](https://doi.org/10.31234/osf.io/tvarg)
- Kruschke, J. K. (2018). Rejecting or accepting parameter values in Bayesian estimation. *Advances in Methods and Practices in Psychological Science*, 1(2), 270-280.

Appendix D

This is a review and comment on Kekecs et al. update on stage 1 RR entitled “Raising the value of research studies in psychological science by increasing the credibility of research reports: The Transparent Psi Project.”

After reading the comments of the associate editor and the comments of the other reviewers I would like to add some precision about my position toward this work. I want to insist on some of their points because I think that the scientific issue of this RR is important. I agree with many of the points they highlighted and join them in their opinion overall. Like them, I also have a very strong belief that precognition does not exist. I also agree that psychological mechanisms can only be in line with physical mechanisms. In other words, knowing the current state of knowledge in physics, biology, and psychology, it would require extremely strong evidence to make me change my mind about precognition. Thus, my prediction is that a well designed study has a very low probability to bring evidence toward precognition.

As it has been already said by other reviewers, we can discuss the interest of such a study. Is it relevant to spend resources on pseudoscientific content when other important questions deserve attention in psychology? However, the precognition topic has already been addressed in the psychology literature and some studies claim to bring evidence toward precognition. This can have an impact on the beliefs toward precognition in the general population. This can also tarnish the “reputation” of psychology as a science. Hence, I believe that we need a clear and clean scientific work in order to provide a rigorous answer to previous studies on the subject.

I agree with the first reviewer’s comment on the impact of a potential positive result. It is important indeed to consider the dissemination of the results. We, as scientists, have a duty to ensure the good interpretation of the results. We also have a responsibility to help people understand Bayesian thinking and interpretation of the results knowing the current state of knowledge (and this is well done in a footnote added by the authors). This should be considered both in the scientific paper and other dissemination media. I also believe that there could also be an easy misunderstanding about the aim of this study when disseminated. The scientific team chooses to replicate the initial study as close as possible and they do not want to provide a better protocol. Hence, as the authors explain, the study is not necessarily well designed to test precognition. The study aims to understand how positive results could emerge from the initial study. In my opinion, this point will be very important to highlight when communicating to the general population. Besides, it will also be necessary to explain why a very rigorous procedure may not be sufficient to test precognition. Indeed, the scientific team chooses to improve some aspects as compared to the original study (e.g. data analysis, transparency...) but also chooses to keep some methodological points identical. I understand why because it has been well explained by the authors. But again, the justification of these choices may be of high importance when communicating to a wide audience.

I am overall satisfied with the responses to my previous comments. I think that some points could still be discussed, especially concerning data analysis. However the decisions of the scientific team are clearly justified. Hence, I think that the protocol is ready to be launched.

Appendix E

Cover letter for manuscript submission

Dear Professor Chambers,

We submit a Stage 2 Registered Report to Royal Society Open Science titled ‘Raising the value of research studies in psychological science by increasing the credibility of research reports: The Transparent Psi Project.’.

The primary aim of our project is to provide a textbook case of how to utilize methodological best practices to systematically eliminate bias from researcher degrees of freedom and other human factors, and thus, to produce research matching the highest credibility standards in psychological science. This is demonstrated through a fully transparent multi-site replication of Bem’s (2011) study 1, designed by a consensus panel of 29 experts.

We have completed data collection after collecting data from 2,220 participants according to the pre-specified stopping rules of our study. We found conclusive support for the null hypothesis, contrary to Bem’s original findings. We have successfully used the planned methodological tools to achieve full transparency and verifiable credibility of these findings. In the end of the manuscript we discuss our experiences with implementing these methodological tools, and we formulate recommendations based on these experiences.

The unique added value of our study lies in the practical demonstration of the methodological innovations as well as in carrying out a consensus-based research protocol on this highly polarized topic.

As per the Stage 2 registered report guidelines of RSOS:

- The Data Accessibility section on page 47 contains the URL for archived study data, digital materials/code and the laboratory log. Data and materials are made freely available via Open Science Framework.
- The Materials and Methods section on page 9 contains the URL for the approved Stage 1 protocol on the Open Science Framework.
- No data other than pilot data included at Stage 1 was collected prior to the date of IPA.
- The introduction and methods section was substantially unchanged compared to the approved Stage 1 RR, although minor grammatical and stylistic edits were made to help readability of the text and to change tense from future to past tense where this was relevant.
- We confirm the research was carried out in accordance with the approved research protocol, and there were no substantial changes to the preregistered protocol. Two minor changes occurred, which do not influence the outcome of the study, and are clearly described in the Changes compared to the preregistered protocol subsection on page 24.

- All registered analyses are reported. Exploratory and post-hoc analyses are clearly labeled as exploratory and are separated from pre-specified confirmatory tests. The results of the exploratory analyses do not influence the conclusions.

Let me know if you have any questions or if we need to submit additional materials.

Best wishes,

Zoltan Kekecs, PhD

Assistant Professor, ELTE, Department of Affective Psychology

kekecs.zoltan@ppk.elte.hu

Appendix F

This is a review and comment on Kekecs et al. stage 2 RR entitled “Raising the value of research studies in psychological science by increasing the credibility of research reports: The Transparent Psi Project.”

First of all, I would like to reiterate my enthusiasm for this project. This example of experimental procedure is of high importance for the research field. It is crucial to provide examples on how to manage a scientific project with all the tools supposed to increase transparency and scientific quality. The research team did everything to ensure the quality of the protocol and the methodological rigour proposed here is impressive. The precision of information in the manuscript and online is far above the classical reports in the field.

Below are some minor comments or suggestions. I consider them minor because whatever decision the authors will make at these levels, the quality of the initial work is already impressive.

A) Checks and controls

- I cannot access the experiment at <https://transparentpsi.com>. (Error message). The demonstration of the experiment (OpenSesame, <https://osf.io/ypwk3>) is working but a) there are minor display problems (sometimes, text sticks out of the screen), but I guess this was managed in the browser and b) Information form and consent form are not displayed (I guess this is normal since this is only a demonstration).
- I checked <https://github.com/kekecsz/transparent-psi-results/commits/master>. As expected, 0 deletion during data collection and after (from January 10, 2020 to November, 18, 2022 <https://github.com/kekecsz/transparent-psi-results/graphs/contributors?from=2020-01-09&to=2022-11-18&type=c>).

According to my checks and controls (unless I am mistaken) :

- data are able to test the authors’ proposed hypotheses
- the Introduction, rationale and stated hypotheses are the same as the approved Stage 1 submission
- the authors adhered to the registered experimental procedures. Minor deviations are explained and not significant
- Conclusions are justified given the data

B) Minor comments

- **p.12** “*Data that are made public as they are being collected are referred to as ‘born-open’ data (Rouder, 2016)*”. This is a small detail, but (as I understand it) according to Rouder (2016), born-open data are data created without any human approval or action, versioned, logged, time stamped, and uploaded/archived as they are being collected. Hence, we could imagine born-open data which are not made fully public e.g. like a) open data with protected access, when sensitive, personal data are available only from an approved third-party repository that manages access to data to qualified researchers through a documented process or b) open data hidden from experimenters during the experiment. More generally, I am not sure to clearly understand the difference between born-open data as defined by Rouder (2016) and direct data deposition as described in the paper. As this procedure is an important part of the project, it could be important/interesting to clearly explain the difference between these two approaches (maybe with a table ?), especially concerning data integrity. For instance, the authors state **p.12** that “*One difficulty with born-open data is that sharing data in real-time can result in some participants’ data becoming identifiable for those who know when a given person participated in the study.*” but again, as I understand Rouder (2016), “real time” is a flexible procedure. Indeed Rouder (2016) explains that in his lab, upload occurs nightly, that is, the data are available within 24 hours of their creation.

- **p.27** *“In total, 2,220 individuals participated in the study. Among these, 2,207 participants started the session before the study stopping rule was triggered. An additional 13 participants started the session after the stopping rule was met, but their data were not included in the analysis.”* In my opinion, these last 13 participants should still be included in the analyses. Analysing more data than expected can't hurt the conclusions, unless they are particularly noisy. I recognize that this is a small number of observations compared to the total number of observations. The conclusions would probably not change whether or not these data were included. But as a matter of principle, I consider that existing data should be used. I understand the aim of respecting the pre-registration as closely as possible, of course. But here, the deviation from the pre-registration would not harm the quality of the conclusions, on the contrary.
- **p.35** *“Our results cannot be interpreted as evidence against the existence of ESP itself”* Could the authors specify why? In particular, this appears to me to be in contradiction with the following sentence **p.37** *“The findings of this study are not consistent with the predictions of the ESP model in this particular paradigm”*
- Overall, if we want the field to move towards this type of protocol, it will probably be necessary to simplify the implementation of the procedures and/or train the researchers, especially regarding computer skills. The large amount of technical details to be considered might scare off some psychology researchers. In my opinion, the next step is to make the implementation of the protocols more accessible.
- Would it be worthwhile to add a README file in order to explain the function of R codes, how to use them and in what order to use them (as the authors already did for the OpenSesame demonstration)? I think this would help to reproduce the analyses.

C) Details

- I checked the box *“Should the paper be seen by a specialist statistical reviewer?”* but this is not a specific choice I make for this work. I think that any work should be seen by a specialist statistical reviewer.
- The authors may now cite (Beffara et al., 2018) / Beffara, B., Bret, A., & Nalborczyk, L. (2018). A fully automated, transparent, reproducible, and blind protocol for sequential analyses. PsyArXiv. <https://doi.org/10.31234/osf.io/v7xpg> **p.13 and p.50** as Beffara Bret, B., Beffara Bret, A., & Nalborczyk, L. (2021). A fully automated, transparent, reproducible, and blind protocol for sequential analyses. *Meta-Psychology*, 5. <https://doi.org/10.15626/MP.2018.869> for a more recent/developed version.

D) Thoughts about RRs, robust science and peer-review

I humbly submit this short paragraph as an element of reflection for the authors but also for the editor and other reviewers. The robust search method implemented by the authors is a concrete and comprehensive example of a search practice to be advocated. In this context, an important point to consider, in my opinion, is that this method greatly increases the amount of material/documents to review. I think this is a very good thing! However, I wonder if it is possible to review all of this material in a comprehensive and rigorous manner with a conventional number of reviewers. Moreover, I wonder if this type of research, which I hope will become more common, would not require to assign specific tasks to the different reviewers (results, scripts, code, procedure, respect of the pre-recording, data integrity...). Perhaps systematizing the comparison between stage 1 and stage 2 of the RR could also improve the quality of the evaluation process. I do not pretend to provide any perfect solution here. Rather, I wish to suggest that the peer-review process should perhaps evolve along with research practices.

Again, I want to underscore the meticulous job done by the research team and I would like to thank the authors for this work. I look forward to their response to my comments.

Appendix G

Review of Royal Society RSOS-191375.R2

Thank you for the opportunity to review the manuscript RSOS-191375.R2 ('transparent Psi project'). I reviewed the original preregistration proposal as well as R1. The authors had two major aims [e.g., p. 7, p. 39]: the primary aim was to develop and execute a rigorous testing protocol for psychological experiments. The second aim was to illustrate this protocol on a contested psychological study, Bem's psi experiment. I believe the study fully fulfils the second aim. In my previous review of the pre-registration I raised concerns about the cost-benefit of the proposed safeguards and feasibility of implementation in future studies. I wondered if "the time, effort, human labour, and dollar costs of such a study can be invested regularly in scientific investigations." I suspect this type of concerted effort might be limited to a small number of critical studies that attempt to resolve an ongoing dispute, so future users of the suite of tools would benefit from some cost-benefit analysis.

Overall, the paper is clear and well-written, and can probably be published in its present form. It offers what is possibly the most comprehensive treatment of a psychological phenomenon I have ever read. This is exactly why it provides, in my opinion, such a convincing answer to the debate around psi, and at the same time raises the question of whether such a mammoth effort can be routinely invested to explore psychological questions.

Given the time, effort, and costs associated with the proposed measures, it would be helpful to include some form of cost-benefit analysis, perhaps something as simple as a table with the benefits and crude cost estimates of each tool. The authors have done a good job in rank-ordering the measures based on subjective reports [p. 46-49], but what about formal estimates of, say, cost, personnel, and time? among other concerns, setting such a high threshold for expensive and time-consuming control measures might preclude many labs (and research students) from ever publishing psychological research again. I fully understand the conundrum: the Authors offer means for best-practice, why should we settle for less? yet at the same time it is clear very few lab have the necessary resources. A rudimentary table could allow labs with limited resources to identify the most cost-effective control measures.

Other considerations:

p. 14 and elsewhere: Bore open data and real-time research reports can lead to interpretation of partial results without awaiting the prescribed stopping rule. How can this risk be minimised? I believe the authors have addressed this in their response letter to R1 (by placing a warning message) and wonder if it should be included as part of the method's best-practice?

p. 15: I applaud the idea of accumulating evidence over time, and the constant update of belief. At the same time I wonder HOW this is done in different situations such as the example below. In the example figure, the x axis represents time and the y axis could be any measure of evidence strength. I present two hypothetical scenarios, one in which evidence is accrued monotonically, perhaps even linearly, over time (solid blue line), and the other in which it is noisy, going up and down (dotted red line). I hope it illustrates the challenges to interpretation. How can researchers evaluate this accumulation over time? Remember that at any point in time the researchers do not know what happens to evidence strength in point $t+1$. Perhaps they should update their beliefs differently depending on the observed pattern. Can the Authors provide guidelines for interpretation?

p. 28 (Table 1): please indicate in the caption the possible range of the ASGS scale. ASGS scores from Padova, and to a lesser extent Amsterdam vary quite drastically from those of other locations. Should this be taken into consideration when interpreting the results ? for example (assuming high ASGS implies stronger beliefs in para-normal), should evidence in favour of psi from these institutions be overweighted, and evidence against psi underweighted ?

On a somewhat related note, I do not know whether this is practical, but it would be interesting to know something about the beliefs of the Reviewers of this manuscript, and more broadly what are the prior beliefs of Reviewers on any topic they review, when they evaluate articles. For example, an 'anti-paranormal' Reviewer would require quite strong evidence to be persuaded otherwise. Would they be more likely to recommend 'accept' for this submission ?

Appendix H

This is a response letter to the editor's and reviewer's comments on submission RSOS-191375.R2

We are grateful for the helpful comments by the editor and the reviewers. Below we provide our response to each of the suggestions and comments.

Associate editor's comments:

Comment # 1:

The four reviewers from Stage 1 kindly returned to evaluate your Stage 2 manuscript, and I have also read it carefully myself. As I expected based on my reading, the reviews are very positive -- perhaps the most consistently glowing I have seen in the last ten years. As you know (and as I laid out in my very first action letter over 3 years ago), I approached your submission with some hesistance as editor given the topic area and my own biases about the existence of ESP. However, after consultation, I decided at the time that my initial instincts to desk reject your submission were wrong, and at the point of Stage 1 IPA I was convinced that I had made the correct decision to encourage you to proceed. Now seeing the completed Stage 2 manuscript, I am even more convinced that this is quite a special piece of work. It is difficult to estimate the impact of any one contribution to the science, but I believe your article may kickstart a transformation in the way RRs are used to tackle the most important and/or controversial questions in any field. For this, I want to congratulate the entire team for this remarkable accomplishment.

The reviews are very positive, but there are nevertheless some useful nuggets to consider in Stage 2 revision. One in particular that resonated with me was the comment by Reviewer 4 to beef up the cost/benefit analysis in the Discussion. Most of the other suggestions are minor. In revising, please avoid any further changes to the parts of the manuscript unless doing so is necessary to correct an error of fact or typo/grammatical error (Nb. this instruction overrules all reviewer requests that might suggest doing otherwise).

Response #1:

We are grateful for your positive feedback and your support.

We improved the 7. Discussion of the methodological tools section by including a new Cost-analysis section, exploring the costs (especially work hour costs) of the project and the specific research methods. We also made some other adjustments to the other sub-sections in this section to incorporate the results of this cost-analysis.

Reviewer 1's comments:

I appreciate the opportunity to provide a Stage 2 review for this Registered Report. I've read compared the Stage 1 and Stage 2 manuscripts and examined the changes/additions in detail. The discussion is informative, concise, and balanced. I have no concerns with the conduct, reporting, or interpretation of the study, and my review is therefore brief.

Comment #2:

I have one minor comment/question:

Perhaps I missed the explanation, but in Table 1, why are some details (e.g., the institution or PI name) about participating laboratories not disclosed? Perhaps the reason could be provided in the table caption.

Below I have responded directly to the journal's questions for reviewers:

1. Whether the data are able to test the authors' proposed hypotheses by passing the approved outcome-neutral criteria (such as absence of floor and ceiling effects or success of positive controls or other quality checks). Failure to pass these conditions may lead to manuscript rejection.

Yes.

2. Whether the Introduction, rationale and stated hypotheses are the same as the approved Stage1 submission (required).

Yes.

3. Whether the authors adhered precisely to the registered experimental procedures. Essentially yes. The authors report a few minor deviations from the registered protocol, which all seem completely appropriate to me.

4. Where applicable, whether any unregistered exploratory statistical analyses are justified, methodologically sound, and informative.

Yes.

5. Whether the authors' conclusions are justified given the data. Please note that editorial decisions will not be based on the perceived importance, novelty, or conclusiveness of the results.

Yes.

As a final word: this collaboration between researchers from both sides of a scientific debate, working together through a consensus design process is highly commendable. Moreover, I believe this is the most rigorous and transparent study I've ever read. For a group of researchers to voluntarily opt for this level of protection against their own biases is inspiring. A wonderful demonstration that the scientific spirit is alive and well. I congratulate the authors on their achievement and look forward to seeing this work published.

Response #2:

"Not disclosed" means that the lab members did not give permission for that data point to be disclosed in order to protect personal data such as the ASGS score of a certain individual or the identity of the experimenters on the trial session video recordings. We have added this to the table notes for Table 1.

We are thankful for your encouraging words, and your helpful comments throughout the review process.

Reviewer 2's comments:

Comment #3:

This is a review and comment on Kekecs et al. stage 2 RR entitled "Raising the value of

research studies in psychological science by increasing the credibility of research reports: The Transparent Psi Project.”

First of all, I would like to reiterate my enthusiasm for this project. This example of experimental procedure is of high importance for the research field. It is crucial to provide examples on how to manage a scientific project with all the tools supposed to increase transparency and scientific quality. The research team did everything to ensure the quality of the protocol and the methodological rigour proposed here is impressive. The precision of information in the manuscript and online is far above the classical reports in the field.

Below are some minor comments or suggestions. I consider them minor because whatever decision the authors will make at these levels, the quality of the initial work is already impressive.

A) Checks and controls

- I cannot access the experiment at <https://transparentpsi.com>. (Error message). The demonstration of the experiment (OpenSesame, <https://osf.io/ypwk3>) is working but a) there are minor display problems (sometimes, text sticks out of the screen), but I guess this was managed in the browser and b) Information form and consent form are not displayed (I guess this is normal since this is only a demonstration).
- I checked <https://github.com/kekecsz/transparent-psi-results/commits/master>. As expected, 0 deletion during data collection and after (from January 10, 2020 to November, 18, 2022 <https://github.com/kekecsz/transparent-psireresults/graphs/contributors?from=2020-01-09&to=2022-11-18&type=c>).

According to my checks and controls (unless I am mistaken) :

- data are able to test the authors' proposed hypotheses
- the Introduction, rationale and stated hypotheses are the same as the approved Stage 1 submission
- the authors adhered to the registered experimental procedures. Minor deviations are explained and not significant
- Conclusions are justified given the data

Response #3:

Thank you for checking these materials! <https://transparentpsi.com/> is accessible, however, we did not renew the https protocol certificate for the site after we stopped data collection. That is why most browsers will issue a warning message when trying to access the site. Now we renewed the certificate, and it will be valid for another 3 months, so the reviewer can access the data collection site. The data collection platform can only be accessed in test mode (accessed in the dropdown “session type”). That mode has a few differences compared to the live experiments, such as that all erotic images are replaced by a uniform green image to protect underaged people and those who want to see the working of the platform but do not want to see erotic content. Also, the waiting time of the first starry sky image is reduced to a few seconds, so that the platform can be tested faster. Also, data is pushed immediately to the github repository. This is different from the live mode, where data was only pushed to github after every 200 rows to protect personal data. Except for these differences, the test mode is identical to the live mode.

Comment #4:

B) Minor comments

– p.12 “Data that are made public as they are being collected are referred to as ‘bornopen’ data (Rouder, 2016)”. This is a small detail, but (as I understand it) according to Rouder (2016), born-open data are data created without any human approval or action, versioned, logged, time stamped, and uploaded/archived as they are being collected. Hence, we could imagine born-open data which are not made fully public e.g. like a) open data with protected access, when sensitive, personal data are available only from an approved third-party repository that manages access to data to qualified researchers through a documented process or b) open data hidden from experimenters during the experiment. More generally, I am not sure to clearly understand the difference between born-open data as defined by Rouder (2016) and direct data deposition as described in the paper. As this procedure is an important part of the project, it could be important/interesting to clearly explain the difference between these two approaches (maybe with a table ?), especially concerning data integrity. For instance, the authors state p.12 that “One difficulty with born-open data is that sharing data in real-time can result in some participants’ data becoming identifiable for those who know when a given person participated in the study.” but again, as I understand Rouder (2016), “real time” is a flexible procedure. Indeed Rouder (2016) explains that in his lab, upload occurs nightly, that is, the data are available within 24 hours of their creation.

Response #4:

Thank you for the reminder about Rouder’s implementation of born-open data. To be honest, the writing of this section was based on my memory of an earlier reading of his paper, and I did not realize that the process already incorporated direct data deposition (to GitHub) and that it also talks about some benefits of this process. Now re-reading this paper, I don’t see much difference, only in emphasis of the uses and benefits. So now, we reference Rouder 2016 in the Direct Data Deposition subsection: “For an earlier implementation of a form of direct data deposition, see (31)” Also, in the Born-open data subsection we explain why we find it useful to talk separately about direct data deposition and born-open data: “Rouder (31) described how their research lab set up an automatic periodic (daily) push of the contents of a shared local drive containing research data collected at their lab to a public GitHub repository. Rouder labeled this process as creating born-open data. However, we believe it is useful to explicitly distinguish direct data deposition (meaning immediate and direct saving of data into a version controlled third party repository) from born-open data (i.e., the data is made public immediately as it is being collected). Direct data deposition is meaningful in its own right without data sharing because it provides a tool through which the integrity of the raw data can be demonstrated. In some situations, researchers may not want or be able to share data; nevertheless, they can still use direct data deposition to substantiate claims about their data integrity (e.g., that no data was deleted, modified, or excluded during analysis). So here, we define born-open data a little differently from Rouder, referring only to the immediate data sharing aspect without direct data deposition.”

Comment #5:

– p.27 “In total, 2,220 individuals participated in the study. Among these, 2,207 participants started the session before the study stopping rule was triggered. An additional 13 participants started the session after the stopping rule was met, but their data were not included in the analysis.” In my opinion, these last 13 participants should

still be included in the analyses. Analysing more data than expected can't hurt the conclusions, unless they are particularly noisy. I recognize that this is a small number of observations compared to the total number of observations. The conclusions would probably not change whether or not these data were included. But as a matter of principle, I consider that existing data should be used. I understand the aim of respecting the pre-registration as closely as possible, of course. But here, the deviation from the pre-registration would not harm the quality of the conclusions, on the contrary.

Response #5:

The Transparent Psi Project is all about playing it by the book. We explicitly say in our protocol what will be the analysis points in our studies. Including data from these 13 participants will not change the conclusion substantially, on the other hand it would add a protocol deviation, which is something we go into great lengths to avoid in this project. Statistically it is also a bit problematic to deviate from the pre-set stopping points, because this would influence (ever so slightly) the operational characteristics of the study (type I and II error probability estimates). (Not to speak of the non-negligible effort of re-calculating all numbers included in the paper and reproducing all figures with these 13 participants added.) All in all, with respect, we would rather not include these 13 participants in our results in the paper. Of course, since all data and code is publicly available, anyone can calculate the results with and without these 13 participants to see what would be the difference in the results, if they wanted to.

Comment #6:

– p.35 “Our results cannot be interpreted as evidence against the existence of ESP itself”
Could the authors specify why ? in particular, this appears to me to be in contradiction with the following sentence p.37 “The findings of this study are not consistent with the predictions of the ESP model in this particular paradigm”

Response #6:

We rephrased the sentence to “Our results cannot rule out the existence of ESP itself, ...”

Comment #7:

– Overall, if we want the field to move towards this type of protocol, it will probably be necessary to simplify the implementation of the procedures and/or train the researchers, especially regarding computer skills. The large amount of technical details to be considered might scare off some psychology researchers. In my opinion, the next step is to make the implementation of the protocols more accessible.

Response #7:

We completely agree. We are working on solutions that would make some of these methodological tools more accessible, such as born open data, direct data deposition, real-time research reports, and some other credibility-enhancing features.

Comment #8:

– Would it be worthwhile to add a README file in order to explain the function of R codes, how to use them and in what order to use them (as the authors already did for the OpenSesame demonstration) ? I think this would help to reproduce the analyses.

Response #8:

The R code has extensive comments included in the code that in our opinion does allow for reproducibility. We have tested this during our code review process, which was done by an individual who was not closely familiar with our project. Could it be improved? Of course, but it would take significant work. This was one of the objectives we were working towards during the project that was not finished in time for the publication. Here is a package that Marton Kovacs was working on, in which he is re-implementing the TPP and the real-time research report code in a cleaner and more human-interpretable format:

<https://github.com/marton-balazs-kovacs/tppr>

Since it is not bug-fixed, we cannot include it in the current iteration of the paper.

<https://marton-balazs-kovacs.github.io/tppr/index.html>

Comment #9:

C) Details

– I checked the box “Should the paper be seen by a specialist statistical reviewer?” but this is not a specific choice I make for this work. I think that any work should be seen by a specialist statistical reviewer.

– The authors may now cite (Beffara et al., 2018) / Beffara, B., Bret, A., & Nalborczyk, L. (2018). A fully automated, transparent, reproducible, and blind protocol for sequential analyses. PsyArXiv. <https://doi.org/10.31234/osf.io/v7xpg> p.13 and p.50 as Beffara Bret, B., Beffara Bret, A., & Nalborczyk, L. (2021). A fully automated, transparent, reproducible, and blind protocol for sequential analyses. *Meta-Psychology*, 5. <https://doi.org/10.15626/MP.2018.869> for a more recent/developed version.

Response #9:

Thank you for these comments. We have changed the citation to the published paper.

Comment #10:

D) Thoughts about RRs, robust science and peer-review I humbly submit this short paragraph as an element of reflection for the authors but also for the editor and other reviewers. The robust search method implemented by the authors is a concrete and comprehensive example of a search practice to be advocated. In this context, an important point to consider, in my opinion, is that this method greatly increases the amount of material/documents to review. I think this is a very good thing! However, I wonder if it is possible to review all of this material in a comprehensive and rigorous manner with a conventional number of reviewers. Moreover, I wonder if this type of research, which I hope will become more common, would not require to assign specific tasks to the different reviewers (results, scripts, code, procedure, respect of the pre-recording, data integrity...). Perhaps systematizing the comparison between stage 1 and stage 2 of the RR could also improve the quality of the evaluation process. I do not pretend to provide any perfect solution here. Rather, I wish to suggest that the peer-review process should perhaps evolve along with research practices.

Again, I want to underscore the meticulous job done by the research team and I would like to thank the authors for this work. I look forward to their response to my comments.

Response #10:

We really appreciate your efforts at checking our materials and we fully realize that this project produces a massive amount of materials to check compared to other projects. This is one of the reasons we see it important to have paid auditors (or reviewers) who are contracted to do specific tasks in this review process. More importantly, that is why transparency will only get us (researchers) so far in terms of demonstrating integrity of our work, and we will have to rely more on automation, which can help us handle the ever-increasing amount of materials and data involved with high quality research.

Reviewer 3's comments:

Comment #11:

This paper is absolutely beautiful. It is the most comprehensive demonstration of 'open' and robust research practices yet, and deserves its place in the literature. The authors show 'how it is done' in a very convincing way. I applaud the authors for not only taking into account methodological rigor, but also research ethics and privacy. All in all, although one could (rightfully) argue that some of the measures the authors have taken are a bit too much of a good thing, the present project represents IMO the new gold standard for replications of high stake studies.

It also is the final nail in the coffin of the Bem paradigm. As such, this paper marks a very significant point in experimental parapsychology - after 11 years of discussion on the Bem findings, I would argue that even the most staunch believer will have to conclude that the alleged effect does not exist as a mechanistically replicable phenomenon. The importance of this cannot be overestimated - as the authors state, the paper does not disprove the psi hypothesis, but does provide very strong evidence against the psi-as-signal hypothesis.

The only minor point I would make is that the observed result, namely a null-effect for psi in a fully open setting, is completely in line with an alternative for the psi-as-signal hypothesis, namely the 'psi-as-correlation' hypothesis. In particular Walter Von Lucadou has argued that psi effects 'disappear' if one attempts to make direct measurements (or use psi to send signals from the future to the present, as in this experiment), and may only be manifested in anomalous correlation patterns within the data. Although I understand the skeptical tone in the present paper given the extremely convincing null result, it would be at least fair to address and acknowledge that even in the psi community, the ESP hypothesis is not uncontroversial.

Response #11:

We are grateful for your encouraging feedback and support.

We have added a sentence to the discussion right after we say that our results cannot rule out the existence of ESP: "In fact, there are some theories of ESP that are compatible with our finding (Walach et al., 2014)."

Reviewer 4's comments:

Comment #12:

Thank you for the opportunity to review the manuscript RSOS-191375.R2 ('transparent Psi project'). I reviewed the original preregistration proposal as well as R1. The authors had two

major aims [e.g., p. 7, p. 39]: the primary aim was to develop and execute a rigorous testing protocol for psychological experiments. The second aim was to illustrate this protocol on a contested psychological study, Bem's psi experiment. I believe the study fully fulfils the second aim. In my previous review of the pre-registration I raised concerns about the cost-benefit of the proposed safeguards and feasibility of implementation in future studies. I wondered if "the time, effort, human labour, and dollar costs of such a study can be invested regularly in scientific investigations." I suspect this type of concerted effort might be limited to a small number of critical studies that attempt to resolve an ongoing dispute, so future users of the suite of tools would benefit from some cost-benefit analysis.

Overall, the paper is clear and well-written, and can probably be published in its present form. It offers what is possibly the most comprehensive treatment of a psychological phenomenon I have ever read. This is exactly why it provides, in my opinion, such a convincing answer to the debate around psi, and at the same time raises the question of whether such a mammoth effort can be routinely invested to explore psychological questions.

Given the time, effort, and costs associated with the proposed measures, it would be helpful to include some form of cost-benefit analysis, perhaps something as simple as a table with the benefits and crude cost estimates of each tool. The authors have done a good job in rank-ordering the measures based on subjective reports [p. 46-49], but what about formal estimates of, say, cost, personnel, and time ? among other concerns, setting such a high threshold for expensive and time-consuming control measures might preclude many labs (and research students) from ever publishing psychological research again. I fully understand the conundrum: the Authors offer means for best-practice, why should we settle for less ? yet at the same time it is clear very few lab have the necessary resources. A rudimentary table could allow labs with limited resources to identify the most cost-effective control measures.

Response #12:

Thank you for your encouraging feedback.

We agree that such a cost-analysis could be useful to evaluate which of the credibility-enhancing methods could be implemented by a particular lab, and also help in seeking funding for high quality research projects. So we improved the 7. Discussion of the methodological tools section by including a new Cost-analysis subsection, exploring the costs (especially work hour costs) of the project and the specific research methods. We also made some other adjustments to the other subsections in this section to incorporate the results of this cost-analysis.

Comment #13:

Other considerations:

p. 14 and elsewhere: Bore open data and real-time research reports can lead to interpretation of partial results without awaiting the prescribed stopping rule. How can this risk be minimised ? I believe the authors have addressed this in their response letter to R1 (by placing a warning message) and wonder if it should be included as part of the method's best-practice ?

Response #13:

We have included a comment on this in the 7. Discussion of the methodological tools section when discussing the limitations of real-time data sharing and reporting: “There is also a risk of people following the study results via the real-time research reports drawing conclusions prematurely from interim results. In our study we tried to mitigate this risk by placing a warning on the real-time report site saying “Result not yet final! Data presented on this page represent the current trend calculated from the data. The results should not be over-interpreted! Random variations may cause the data to cross the decision thresholds. Statistical decisions will only be drawn at the pre-specified stopping points. The next stopping point will be at reaching X trials.” Researchers implementing real-time research reports might consider including similar warnings.”

Comment #14:

p. 15: I applaud the idea of accumulating evidence over time, and the constant update of belief. At the same time I wonder HOW this is done in different situations such as the example below. In the example figure, the x axis represents time and the y axis could be any measure of evidence strength. I present two hypothetical scenarios, one in which evidence is accrued monotonically, perhaps even linearly, over time (solid blue line), and the other in which it is noisy, going up and down (dotted red line). I hope it illustrates the challenges to interpretation. How can researchers evaluate this accumulation over time? Remember that at any point in time the researchers do not know what happens to evidence strength in point $t+1$. Perhaps they should update their beliefs differently depending on the observed pattern. Can the Authors provide guidelines for interpretation?

Response 14:

Updating of belief is not an integral part of the paper, and it would require a lot of background to be able to talk about this in sufficient depth in the paper. So we decided not to

change the manuscript in response to this comment. Nevertheless, we are happy to answer the question here: The two lines are realistically from two different time points of the study timeline. The red line represents shifts in evidence while we only have access to a small amount of data. the blue line is a better representation of the change in evidence when a lot of data is available. For example when using Bayes Factor, one often sees large changes in Bayes Factor throughout the first couple of data collection sessions, while after a while there is a stabilization of BF. This is often misinterpreted as the evidence is trending towards supporting one model or another, and as we get more data, we see more clearly. But this is not necessarily the case. It is simply a mathematical necessity, that in order to overthrow previous data, you need more and more contrary evidence, as the amount of data in the past grows larger. It also depends on the settings of the priors. Like in the case of Wagenmakers, E. J., Wetzels, R., Borsboom, D., van der Maas, H. L., & Kievit, R. A. (2012). An agenda for purely confirmatory research. *Perspectives on Psychological Science*, 7(6), 632-638, in Fig 2.

You can see that the BF changes erratically while later it seems to stabilize. Some might interpret this as meaning: we can be more and more confident that H0 is true. However, with the right prior settings and sufficiently low effect size, it is inevitable that the Bayes Factor will increase first, and only start decreasing after sufficient amount of deviations from the null model have been accumulated. So it is generally a mistake to stop data collection because the Bayes factor shows a particular value or trend. Interpretation of the evidence and study stopping should always be determined pre-data collection with the support of thorough analysis of the operational characteristics of the study (i.e. type I and II chance in various scenarios), especially in a sequential or “continuous study”. We hope this sheds some light on the complexity of the question at hand.

Comment #15:

p. 28 (Table 1): please indicate in the caption the possible range of the ASGS scale. ASGS scores from Padova, and to a lesser extent Amsterdam vary quite drastically from those of other locations.

Should this be taken into consideration when interpreting the results ? for example (assuming high ASGS implies stronger beliefs in para-normal), should evidence in favour of psi from these institutions be overweighted, and evidence against psi underweighted?

Response 15:

We have added information in the notes of Table 1 about the range of possible values for the ASGS scale.

As for weighting the results based on ASGS of PIs or experimenters, this analysis was not preregistered, and in our project we would like to keep post-hoc analyses to a minimum in order to decrease the risk of chance findings. Nevertheless, the data required to do this analysis is openly available to anyone who would like to do such re-analysis.

Comment #16:

On a somewhat related note, I do not know whether this is practical, but it would be interesting to know something about the beliefs of the Reviewers of this manuscript, and more broadly what are the prior beliefs of Reviewers on any topic they review, when they evaluate articles. For example, an 'anti-paranormal' Reviewer would require quite strong evidence to be persuaded otherwise. Would they be more likely to recommend 'accept' for this submission ?

Response 16:

We agree that this is an interesting question, but such an evaluation is outside of the scope of the present paper. We can only hope that the consensus design and the registered report publication format mitigates the effects of such biases, and our paper and other similar papers are evaluated mainly on the merit of their methodological quality.